# Enhancing Near OOD Detection in Prompt Learning: Maximum Gains, Minimal Costs

## Abstract

Prompt learning has shown to be an efficient and effective fine-tuning method for vision-language models like CLIP. While numerous studies have focused on the generalisation of these models in few-shot classification, their capability in near out-of-distribution (OOD) detection has been overlooked. A few recent works have highlighted the promising performance of prompt learning in far OOD detection. However, the more challenging task of few-shot near OOD detection has not yet been addressed. In this study, we investigate the near OOD detection capabilities of prompt learning models and observe that commonly used OOD scores have limited performance in near OOD detection. To enhance the performance, we propose a fast and simple post-hoc method that complements existing logit-based scores and can be easily applied to any prompt learning model without change in architecture or model re-training while keeping the same classification accuracy. Our method boosts existing prompt learning methods' near OOD detection performance in AUROC by up to 11.67% with minimal computational cost. Comprehensive empirical evaluations across 13 datasets and 8 models demonstrate the effectiveness and adaptability of our method[1].

## 1 Introduction

Pre-trained vision-language models such as ALIGN (Jia et al., 2021) and CLIP (Radford et al., 2021) have shown outstanding visual-text understanding by learning to align image features and textual features of a large-scale image-text dataset via contrastive learning. Consequently, CLIP naturally excels at zero-shot image classification, utilising a class name as natural language text instead of arbitrarily numbered category. For instance, cosine similarity between encoded image feature of an image and encoded textual feature of "a photo of a [CLASS]." with a class name as "[CLASS]" can perform zero-shot classification without requiring any additional classification head. Logits with different class names can then be transformed to probabilities by using the softmax function.

While CLIP's zero-shot capabilities are impressive, recent studies have highlighted its sensitivity to prompt wording. For instance, Zhou et al. (2022b) demonstrated that small variations in prompt structure (e.g., "a photo of a [CLASS]" vs. "a photo of [CLASS]") can lead to significant accuracy drops, sometimes exceeding 5% on standard benchmarks like Caltech101 (Fei-Fei et al., 2004). This observation has led to an emerging research direction of prompt learning for few-shot classification with vision-language models (Zhou et al., 2022b;a; Yao et al., 2023; Zhu et al., 2023; Khattak et al., 2023a;b), which optimises continuous context vectors in the word-embedding space, eliminating the need for handcrafting prompts.

Although existing methods have shown success in this area, the majority focus primarily on enhancing classification accuracy, leaving the equally important task of out-of-distribution (OOD) detection underexplored. OOD detection is crucial for real-world, safety-critical applications such as autonomous driving, healthcare, and industrial automation, where models must perform reliably under unfamiliar or unexpected conditions. In these applications, excelling in in-distribution (ID) classification is not enough. Models must also be capable of detecting and effectively handling OOD samples. Here, we focus on the more challenging task of *near OOD detection* (Yang et al., 2023;

---

[1]Codes are available at `https://anonymous.4open.science/r/near-OOD-prompt-learning-25D1`

2022; Zhang et al., 2023b; Fort et al., 2021; Ren et al., 2021; Winkens et al., 2020). Near OOD detection refers to a scenario where OOD samples share the same domain as ID samples but have different label space (e.g., both ID and near OOD samples are flower images with no overlapping classes).

In this paper, we focus on developing a post-hoc method to enhance near OOD detection performance for prompt learning methods in vision-language models without the need to retrain them. To the best of our knowledge, only a few existing works address related problems, and none of them address this specific problem directly. Specifically, Bai et al. (2023); Miyai et al. (2023) investigated far OOD detection, where OOD samples come from completely different domains. However, these methods are not post-hocs, which require to train the models from scratch. In addition, near OOD tasks are out of their consideration. Existing logit-based OOD scores, such as MaxLogit score (Hendrycks et al., 2022) or Energy score (Liu et al., 2020), which estimate a model's confidence in its predictions, can be applied to our case in a post-hoc manner. However, these scores are not specifically designed for vision-language models like CLIP where overlapping score distributions between ID and OOD samples often result in poor detection performance, especially in the near OOD tasks.

We address the above problem by proposing a simple yet novel and effective approach, which introduces a new logit-based score named Marginal Logit Score (MLS), computed from the output logits of an existing method. Tailored for vision-language models, the key idea of MLS is measuring the difference between the existing logit scores and a new complementary score named Context score. MLS creates clearer separation between ID and near OOD samples, leading to substantial performance gains. Notably, our method does not require any change to the model architecture and does not involve retraining, making it highly efficient and adaptable. Table 1 demonstrates the performance gain of MLS when used with existing methods.

Table 1: Improvement of near OOD AUROC using MaxLogit score with our method on 1-shot Caltech101 (Fei-Fei et al., 2004) with wall-clock time measured for the method. Refer to Section 5 for details.

| Model | △ AUROC | Computation Time (s) |
|---|---|---|
| CoOp (Zhou et al., 2022b) | +2.66 | 0.988 |
| CoCoOp (Zhou et al., 2022a) | +5.05 | 1.703 |
| IVLP (Khattak et al., 2023a) | +3.79 | 1.055 |
| KgCoOp (Yao et al., 2023) | +7.09 | 0.989 |
| ProGrad (Zhu et al., 2023) | +5.90 | 0.994 |
| MaPLe (Khattak et al., 2023a) | +5.74 | 0.974 |
| PromptSRC (Khattak et al., 2023b) | +6.09 | 1.005 |
| LoCoOp (Miyai et al., 2023) | +10.57 | 1.153 |

We validate our method across 13 diverse datasets and 8 state-of-the-art prompt learning models. Our experiments show that our framework improves near OOD detection performance by up to 11.67% in terms of AUROC, without affecting the classification accuracy of the underlying models. This demonstrates the versatility and effectiveness of our approach in real-world applications.

## 2 BACKGROUND

### 2.1 CONTRASTIVE LANGUAGE–IMAGE PRE-TRAINING (CLIP)

Contrastive Language–Image Pre-training (CLIP) (Radford et al., 2021), a pre-trained vision-language model learned to align 400 million image-text pairs, is renowned for its powerful zero-shot image classification performance. It measures cosine similarity between image feature of an unseen image and textual feature of a text prompt formatted as "a photo of a [CLASS]" where [CLASS] is the name of a class in a label space of interest. Formally, given an image $I \in \mathbb{R}^{H \times W \times 3}$ with $H$ being the height and $W$ being the width and a text prompt $T =$ "a photo of a [CLASS]", a classification logit is computed by $\langle \text{Enc}_I(I), \text{Enc}_T(T) \rangle$ where $\langle \cdot, \cdot \rangle$ is cosine similarity, $\text{Enc}_I(\cdot)$ is an image encoder, and $\text{Enc}_T(\cdot)$ is a text encoder. The image encoder can be either ResNet (He et al., 2016) or Vision Transformer (ViT) (Dosovitskiy et al., 2021), and the text encoder is Transformer (Vaswani et al., 2017). For the brevity of notation, we omit the notations of the encoders for the remainder of the paper.

### 2.2 PROMPT LEARNING OF CLIP

Prompt learning of CLIP was first introduced by Zhou et al. (2022b) through Context Optimization (CoOp), which adapts popular prompt learning techniques from the natural language pro-

cessing (NLP) field to CLIP. It addresses the issue of CLIP's classification being sensitive to the prompt's prefix (e.g., a large performance gap between when using "a photo of a [CLASS]" and "a [CLASS]") by optimising the prefix with few-shot samples. CoOp learns $M$ continuous context vectors $V = \{V_1, V_2, \cdots, V_M\}$ within word-embedding space where $V_i \in \mathbb{R}^D$ is the $i^{th}$ vector with $D$ being the word-embedding dimension. The learnable prompt is formalised as $P = \{V_1, V_2, \cdots, V_M, C\}$ where $C \in \mathbb{R}^D$ is the word-embedding of a class name appended to the context vectors. The classification logit of $i^{th}$ class is then computed by $\langle I, P_i \rangle$ where $P_i$ is the learnable prompt with the $i^{th}$ class name. The probability is estimated by the softmax function as $p(y = i | I, P_i) = \frac{\exp(\langle I, P_i \rangle / \tau)}{\sum_{k=1}^{K} \exp(\langle I, P_k \rangle / \tau)}$ where $K$ is the total number of classes and $\tau$ is the temperature scale. Cross-entropy loss is then minimised to learn the context vectors. Note that the only learnable parameters that are common among different prompt learning models are the $M$ context vectors. Since the introduction of CoOp, a number of subsequent works have aimed to improve its ID accuracy and generalisability with modifications in model architecture or additional loss terms (See Section 4). The aim of the paper is to develop a post-hoc approach that improves near OOD detection performance while being agnostic to base prompt learning models.

## 2.3 NEAR OUT-OF-DISTRIBUTION DETECTION

OOD detection is largely categorised as far OOD detection and near OOD detection based on the distribution shift between an ID test dataset and an OOD dataset along with difficulty of detection (Ren et al., 2021; Fort et al., 2021; Yang et al., 2021; 2022; Zhang et al., 2023b). Far OOD datasets have covariate shift in images (i.e., OOD samples are from domains that differ from the training set), and near OOD datasets which are more challenging to detect involve semantic shift (i.e., OOD samples are drawn from the same domain as the training set but belong to previously unseen label classes). Near OOD detection is also synonymous with fine-grained OOD detection (Zhang et al., 2023a) and hard OOD detection (Li et al., 2024; Ming et al., 2022).

In this paper, we study near OOD detection via prompt learning of CLIP, a new research problem to which no existing methods are tailored. Given a trained CLIP prompt learning method, we focus on post-hoc approaches that compute a score from the logits of the method to determine whether a given image is from ID or from OOD, which can be written as:

$$g\left(I; \{P_i\}_{i=1}^{K}, \alpha\right) = \begin{cases} 1 & S\left(I; \{P_i\}_{i=1}^{K}\right) \geq \alpha \\ 0 & S\left(I; \{P_i\}_{i=1}^{K}\right) < \alpha \end{cases} \tag{1}$$

where $g(\cdot)$ is a OOD detector, $\alpha$ is the threshold, and $S(\cdot)$ is a score function. By convention, the ground truth label is 1 for ID samples and 0 for OOD samples.

## 3 METHOD

### 3.1 PROBLEM SETTING

We focus on a near OOD detection problem for prompt learning models of CLIP, which is to detect whether a given image $I_{test}$ is from the ID test dataset $\mathcal{D}_{test}^{ID}$ of $(I_{test}^{ID}, y^{ID})$ pairs or a near OOD dataset $\mathcal{D}_{test}^{nearOOD}$ of $(I_{test}^{nearOOD}, y^{nearOOD})$ pairs where $y^{ID} \in \{1, \cdots, K\}$ is the ID label with $K$ classes and $y^{nearOOD} \in \{1, \cdots, L\}$ is the near OOD label with $L$ classes. The ID dataset and the near OOD dataset contain the same types of images (i.e., no covariate shift in images) but have no overlapping classes (i.e., $y^{ID} \cap y^{nearOOD} = \varnothing$). Without loss of generality, we assume that the context vectors $V$ have already been fine-tuned using a prompt learning model with a few-shot ID training dataset and only consider post-training stage in a post-hoc manner.

### 3.2 MOTIVATION

Energy and MaxLogit scores are two widely-used scores in the OOD detection literature (Liu et al., 2020; Hendrycks et al., 2022; Yang et al., 2021; Sun et al., 2022; Yang et al., 2022; Zhang et al., 2023b; Han et al., 2022; Sun et al., 2021). In our prompt learning context, these scores can be used to measure a model's confidence from the output logits (i.e., cosine similarity between image and textual features), based on the assumption that ID samples typically have higher scores than OOD

Figure 1: Density plots of Energy scores (left) and MaxLogit (right) computed with CoOp (Zhou et al., 2022b) on Flowers102 (Nilsback & Zisserman, 2008). Large regions of ID and near OOD samples overlap, which are highlighted by shaded boxes.

samples. Specifically, we have:

$$\textbf{Energy Score} \quad S_{\text{Energy}} = \tau \log \sum_{i=1}^{K} \exp\left(\langle I, P_i \rangle / \tau\right) \tag{2}$$

$$\textbf{MaxLogit Score} \quad S_{\text{MaxLogit}} = \max_{i} \langle I, P_i \rangle \tag{3}$$

Energy score can be viewed as an approximation of MaxLogit score when $\tau = 1$ as $\max_{i} \langle I, P_i \rangle \leq \log \sum_{i=1}^{K} \exp\left(\langle I, P_i \rangle\right) \leq \max_{i} \langle I, P_i \rangle + \log K$.

Although these logit-based scores have been commonly used for OOD detection, they perform sub-optimally in near OOD tasks. This is because near OOD samples often generate logit distributions that closely resemble those of ID samples, causing significant overlap between their score distributions. Figure 1 demonstrates this with a real-world dataset in the near OOD setting, where the overlap makes it difficult to distinguish between the two, leading to poor detection performance.

### 3.3 PROPOSED SCORE FUNCTION

Our objective is to develop a post-hoc method that improves MaxLogit and Energy scores to enhance near OOD detection for any CLIP-based prompt learning model with minimal learning cost.

#### 3.3.1 CONTEXT SCORE AND MARGINAL LOGIT SCORE

We first introduce the concept of **Context Score** $S_{\text{Context}} = \langle I, V \rangle$, which represents the cosine similarity between the image feature and the textual feature from the context vectors *without* any class name. The intuition is that the context vectors capture the general features of the model without associating images with any specific class and this score reflects how well the image aligns with these generic and non-class-specific features. The key insight is that for ID samples, the model should have a strong class-specific association, and therefore the Context score should be relatively low. Conversely, for OOD samples, the model's uncertainty should result in a higher Context score. Given a prompt learning method, Context score can be easily obtained by adding an additional label class with no class name (i.e., extending $K$ classes to $K + 1$ classes).

We argue that Context score is a valuable complement to MaxLogit or Energy score, especially when the difference between these scores is leveraged. Ideally, when a model is given an ID image, it confidently predicts that the image belongs to a specific ID class, resulting in high Energy or MaxLogit score which represents the model's confidence. In contrast, its Context score is much lower, as it is calculated using a prompt lacking the ground-truth class name. This creates a large gap between the original score and the Context score.

Conversely, when the model is given a near OOD image, MaxLogit or Energy score is small as the model is uncertain about the image belonging to any ID class. As a result, the gap between the original score and Context score is much smaller. This difference between Energy or MaxLogit

Figure 2: An example illustrating the effectiveness of MLS using ID and near OOD samples from Flowers102 (Nilsback & Zisserman, 2008) with the MaxLogit score shown in blue and the Context score shown in grey. While the MaxLogit score between the ID image and the near OOD image is not distinguishable (left bar plot), MLS which subtracts Context score from MaxLogit score is much more distinguishable (right bar plot).

score and Context score can be viewed as an alternative score, which we name *Marginal Logit Score (MLS)* defined as:

$$\textbf{Marginal Logit Score (Energy)} \quad S_{\text{MLS-E}} = S_{\text{Energy}} - S_{\text{Context}} \tag{4}$$

$$\textbf{Marginal Logit Score (MaxLogit)} \quad S_{\text{MLS-M}} = S_{\text{MaxLogit}} - S_{\text{Context}} \tag{5}$$

To demonstrate the effectiveness of MLS, we present an ID image and a near OOD image from Flowers102 (Nilsback & Zisserman, 2008) in Figure 2, with MaxLogit scores highlighted in blue and Context scores in grey. The MaxLogit score for the OOD sample is higher than that for the ID sample, indicating the model's failure to differentiate between the two. However, when the Context score is subtracted from the MaxLogit score, MLS is higher for the ID sample, showing that the model can successfully distinguish between the two samples.

We further illustrate its geometric interpretation in Figure 3a and Figure 3b where MaxLogit score and MLS-M are plotted at y-axis and Context score is plotted at x-axis. Geometrically, subtracting Context score from MaxLogit score is the same as applying vertical shearing transformation (Lax, 2007) to the samples, which essentially reduces the overlapping area highlighted by shaded boxes. With the MaxLogit score, the near OOD AUROC is 0.785 which is increased to 0.879 when evaluated with MLS-M.

### 3.3.2 MARGIN SCALE

While the simple subtraction of the Context score improves OOD performance, its fixed margin limits adaptability across different models and datasets. To address this, we introduce a coefficient $\beta$ termed *margin scale* that controls the amount of reduction of Context score from MaxLogit or Energy score in a more flexible manner:

$$\textbf{Marginal Logit Score (Energy)} \quad S_{\text{MLS-E}} = S_{\text{Energy}} - \beta \cdot S_{\text{Context}} \tag{6}$$

$$\textbf{Marginal Logit Score (MaxLogit)} \quad S_{\text{MLS-M}} = S_{\text{MaxLogit}} - \beta \cdot S_{\text{Context}} \tag{7}$$

In Figure 3c, we show that if the margin scale is applied, the near OOD AUROC is improved from 0.879 (equivalent to $\beta = 1$) to 0.942 ($\beta = 2.2$). To further demonstrate the importance of $\beta$ to the base prompt learning models, we show how the near OOD performance in AUROC of different models varies with different $\beta$ in Figure 4, where $\beta = 0$ represents the case of the original score without the context score and $\beta = 1$ is MLS without the margin scale. One can see that the value of $\beta$ significantly affects the performance with all the prompt learning models.

This naturally opens a question on how to set $\beta$ properly. If we were given the OOD samples, we could simply choose the value of $\beta$ that minimises the near OOD performance. However, such an approach is impractical for real-world applications where near OOD samples are unavailable before deployment. To address this, we propose to estimate the margin scale by only using *few-shot ID training samples*. Initially, MaxLogit score and Context score in Figure 3a exhibit positive correlation, leading to significant overlap in the density distributions of ID and near OOD samples. When the near OOD detection AUROC is maximised, as shown in Figure 3c, this correlation is

Figure 3: (a) MaxLogit score, (b) MLS-M without $\beta$, and (c) MLS-M with $\beta$ of test ID and near OOD samples with respect to Context scores. Areas where ID samples and near OOD samples overlap are highlighted with shaded boxes. All scores are computed using MaPLe (Khattak et al., 2023a) on Caltech101 (Fei-Fei et al., 2004). See Appendix A.4 for additional demonstrations with different models and datasets.

minimised, resulting in reduced overlap between the distributions. Thus, we formulate this problem as finding the margin scale that minimises the correlation between MaxLogit or Energy score and Context score.

We propose approximating this correlation using the covariance matrix of a bivariate normal distribution fitted with MLS and Context scores computed from the training samples. A key advantage of this approach is its computational simplicity and the availability of a closed-form solution via maximum likelihood estimation. By leveraging this, we find the margin scale that zeros out the off-diagonals of the covariance matrix of the fitted bivariate normal distribution.

**Lemma 3.1.** *Given $N$ scalar observations $\{\hat{x}_i\}_{i=1}^N$ and $\{\hat{y}_i\}_{i=1}^N$, we define two variables $x = \hat{x}$ and $y = \hat{y} - \beta \cdot \hat{x}$. The scale parameter $\beta$ that zeros out the covariance of two variables (i.e., the off-diagonals of a covariance matrix) which is approximated by maximum likelihood estimation is:*

$$\beta = \frac{\sum_{i=1}^N (\hat{x}_i - \mu_{\hat{x}})(\hat{y}_i - \mu_{\hat{y}})}{\sum_{i=1}^N (\hat{x}_i - \mu_{\hat{x}})^2} \tag{8}$$

*where $\mu_{\hat{x}} = \frac{1}{N} \sum_{i=1}^N \hat{x}_i$ and $\mu_{\hat{y}} = \frac{1}{N} \sum_{i=1}^N \hat{y}_i$.*

By using Lemma 3.1 with $\hat{y}$ being the MaxLogit score or Energy score and $\hat{x}$ being the Context score, the margin scale can be easily estimated with ID training samples (see Appendix A.1 for proof of the lemma). This scale estimation needs to be conducted only once after training finishes. Figure 4 shows the estimated margin scale in red dotted lines, demonstrating that our estimation is close to the value that results in the best performance. In addition to good accuracy, our method is a close-form estimation that only takes a small number of ID training samples with little computational cost.

## 4 RELATED WORK

**Vision-Language Models**   Vision-language models have significantly advanced in recent years, bridging the gap between visual and textual data. Early approaches, such as image captioning models (Karpathy & Fei-Fei, 2015; Wang et al., 2016; You et al., 2016), typically used convolutional neural networks (CNNs) to extract visual features and recurrent neural networks (RNNs) to generate descriptive text. The advent of transformers (Vaswani et al., 2017) handling long-range dependencies more effectively and contrastive learning (Oord et al., 2018) revolutionised this field. Notably, ALIGN (Jia et al., 2021), CLIP (Radford et al., 2021), and LiT (Zhai et al., 2022) leveraged a contrastive learning framework that aligns image and text embeddings in a multimodal space, allowing for zero-shot learning capabilities and impressive generalisation to unseen tasks and datasets. In this work, we leverage the powerful vision-language model CLIP and extend its near OOD capability.

**CLIP-based Prompt Learning**   Despite the remarkable zero-shot performance of CLIP, CLIP shows inherently unstable classification accuracy that varies by wording of prompt. To mitigate this

Figure 4: Near OOD detection AUROC using MLS-M vs. margin scale $\beta$ for CoOp (Zhou et al., 2022b), CoCoOp (Zhou et al., 2022a), IVLP (Khattak et al., 2023a), KgCoOp (Yao et al., 2023), ProGrad (Zhu et al., 2023), MaPLe (Khattak et al., 2023a), PromptSRC (Khattak et al., 2023b), and LoCoOp (Miyai et al., 2023) on 16-shots UCF101 (Soomro et al., 2012). The margin scale is approximated by Eq.(8), shown as red dotted lines.

issue, CoOp (Zhou et al., 2022b) was proposed to optimise a prompt in word embedding space, leveraging prompt learning from the NLP literature. CoCoOp (Zhou et al., 2022a) identified that CoOp has limited generalisation and proposed to condition image features to the learnable prompt. Subsequently, many studies have proposed different techniques to improve the generalisation (Yao et al., 2023; Zhu et al., 2023; Khattak et al., 2023a;b). While its generalisation has been largely improved, its OOD detection has been overlooked. LoCoOp (Miyai et al., 2023) proposed a OOD regularisation to improve OOD detection performance. Nevertheless, no study has addressed near OOD detection of prompt learning models.

**OOD Detection**   An early work of Hendrycks & Gimpel (2017) utilised the maximum softmax probability (MSP) as a score to identify OOD samples. Another notable work is Out-of-DIstribution detector for Neural networks (ODIN) (Liang et al., 2018) which extends MSP by introducing temperature scaling and input pre-processing to enhance separation of the scores from ID samples and OOD samples. Similar to ODIN, Mahalanobis (Lee et al., 2018) score also uses input pre-processing in addition to measuring distance in feature space. Delving into a more challenging task of near OOD detection, several studies analysed benchmarks of pre-trained networks in near OOD detection (Yang et al., 2023; 2022; Zhang et al., 2023b; Fort et al., 2021), and different training methods and score functions were proposed for near OOD detection (Ren et al., 2021; Winkens et al., 2020). Despite significant advancements in OOD detection for traditional classifier-based neural networks, many existing methods are not directly applicable to CLIP-based prompt learning models, which lack classifier heads. Furthermore, since these models do not update their image encoders during fine-tuning, many distance-based methods that rely on image features become ineffective.

## 5 EXPERIMENTS

### 5.1 EXPERIMENTAL SETTINGS

**Datasets**   Following previous works of CLIP-based prompt learning models (Zhou et al., 2022a; Khattak et al., 2023a; Yao et al., 2023; Zhu et al., 2023; Khattak et al., 2023a;b; Miyai et al., 2023), we use 11 publicly available datasets of ImageNet (Deng et al., 2009), Caltech101 (Fei-Fei et al., 2004), OxfordPets (Parkhi et al., 2012), StanfordCars (Krause et al., 2013), Flowers102 (Nilsback & Zisserman, 2008), Food101 (Bossard et al., 2014), FGVCAircraft (Maji et al., 2013), SUN397 (Xiao et al., 2010), DTD (Cimpoi et al., 2014), EuroSAT (Helber et al., 2019), and UCF101 (Soomro et al., 2012). A common task of these works involves training models on half of the label classes (e.g., base classes) and evaluating them on the other half classes (e.g., new classes) to measure base-to-new generalisation. We reframe this task as a near OOD detection problem. Specifically, the models trained on base classes are tested with a dataset where half of the samples belong to the base classes (ID) and the other half to new classes (near OOD). The task is to detect whether each test image

Table 2: Near OOD AUROC (↑) of prompt learning models over 13 datasets using the MaxLogit score and MLS-M.

(a) Average over 13 datasets.

|  | MaxLogit | MLS-M | △ |
|---|---|---|---|
| CoOp | 80.74 | 81.84 | +1.09 |
| CoCoOp | 81.09 | 82.74 | +1.65 |
| IVLP | 81.12 | 84.34 | +3.23 |
| KgCoOp | 80.84 | 83.12 | +2.28 |
| ProGrad | 79.77 | 82.35 | +2.58 |
| MaPLe | 81.06 | 83.94 | +2.88 |
| PromptSRC | 83.85 | 85.77 | +1.92 |
| LoCoOp | 77.55 | 81.74 | +4.18 |

(b) ImageNet.

|  | MaxLogit | MLS-M | △ |
|---|---|---|---|
| CoOp | 93.78 | 94.66 | +0.88 |
| CoCoOp | 94.85 | 95.14 | +0.29 |
| IVLP | 94.55 | 94.70 | +0.15 |
| KgCoOp | 94.21 | 94.21 | +0.01 |
| ProGrad | 93.62 | 94.67 | +1.04 |
| MaPLe | 94.20 | 94.35 | +0.16 |
| PromptSRC | 94.52 | 95.32 | +0.80 |
| LoCoOp | 93.10 | 94.56 | +1.46 |

(c) Caltech101.

|  | MaxLogit | MLS-M | △ |
|---|---|---|---|
| CoOp | 88.27 | 90.12 | +1.85 |
| CoCoOp | 85.80 | 89.02 | +3.22 |
| IVLP | 85.50 | 90.53 | +5.03 |
| KgCoOp | 83.64 | 90.06 | +6.42 |
| ProGrad | 82.96 | 88.85 | +5.89 |
| MaPLe | 85.91 | 91.53 | +5.62 |
| PromptSRC | 84.94 | 90.56 | +5.62 |
| LoCoOp | 76.08 | 87.75 | +11.67 |

(d) OxfordPets.

|  | MaxLogit | MLS-M | △ |
|---|---|---|---|
| CoOp | 86.22 | 88.73 | +2.52 |
| CoCoOp | 89.58 | 92.28 | +2.69 |
| IVLP | 88.84 | 91.94 | +3.10 |
| KgCoOp | 89.94 | 92.64 | +2.69 |
| ProGrad | 87.82 | 89.60 | +1.78 |
| MaPLe | 87.40 | 91.00 | +3.60 |
| PromptSRC | 90.80 | 93.45 | +2.65 |
| LoCoOp | 84.44 | 89.19 | +4.75 |

(e) StanfordCars.

|  | MaxLogit | MLS-M | △ |
|---|---|---|---|
| CoOp | 91.36 | 91.59 | +0.22 |
| CoCoOp | 92.43 | 92.99 | +0.57 |
| IVLP | 90.43 | 92.98 | +2.56 |
| KgCoOp | 92.77 | 93.27 | +0.51 |
| ProGrad | 91.52 | 92.63 | +1.11 |
| MaPLe | 91.39 | 92.85 | +1.47 |
| PromptSRC | 92.88 | 94.24 | +1.35 |
| LoCoOp | 88.24 | 91.94 | +3.70 |

(f) Flowers102.

|  | MaxLogit | MLS-M | △ |
|---|---|---|---|
| CoOp | 90.83 | 91.99 | +1.16 |
| CoCoOp | 87.93 | 89.41 | +1.48 |
| IVLP | 86.20 | 88.45 | +2.25 |
| KgCoOp | 87.61 | 91.12 | +3.52 |
| ProGrad | 89.27 | 91.41 | +2.14 |
| MaPLe | 86.05 | 88.34 | +2.29 |
| PromptSRC | 91.10 | 92.61 | +1.51 |
| LoCoOp | 86.17 | 88.59 | +2.42 |

(g) Food101

|  | MaxLogit | MLS-M | △ |
|---|---|---|---|
| CoOp | 86.70 | 87.91 | +1.21 |
| CoCoOp | 90.52 | 91.63 | +1.10 |
| IVLP | 89.70 | 91.87 | +2.18 |
| KgCoOp | 89.87 | 92.12 | +2.25 |
| ProGrad | 88.60 | 91.05 | +2.45 |
| MaPLe | 89.10 | 92.00 | +2.89 |
| PromptSRC | 90.94 | 92.11 | +1.17 |
| LoCoOp | 84.87 | 90.12 | +5.25 |

(h) FGVCAircraft.

|  | MaxLogit | MLS-M | △ |
|---|---|---|---|
| CoOp | 55.99 | 56.97 | +0.97 |
| CoCoOp | 52.60 | 55.04 | +2.45 |
| IVLP | 58.47 | 64.16 | +5.69 |
| KgCoOp | 57.82 | 57.46 | -0.36 |
| ProGrad | 53.69 | 55.67 | +1.97 |
| MaPLe | 52.18 | 56.93 | +4.74 |
| PromptSRC | 60.63 | 62.50 | +1.87 |
| LoCoOp | 50.99 | 56.12 | +5.13 |

(i) SUN397.

|  | MaxLogit | MLS-M | △ |
|---|---|---|---|
| CoOp | 75.78 | 76.75 | +0.97 |
| CoCoOp | 76.32 | 78.29 | +1.97 |
| IVLP | 77.13 | 79.60 | +2.46 |
| KgCoOp | 76.45 | 77.91 | +1.46 |
| ProGrad | 75.52 | 77.67 | +2.15 |
| MaPLe | 77.62 | 79.73 | +2.11 |
| PromptSRC | 78.51 | 80.70 | +2.19 |
| LoCoOp | 73.97 | 78.00 | +4.02 |

(j) DTD.

|  | MaxLogit | MLS-M | △ |
|---|---|---|---|
| CoOp | 68.90 | 69.60 | +0.69 |
| CoCoOp | 65.10 | 67.17 | +2.07 |
| IVLP | 64.99 | 67.93 | +2.94 |
| KgCoOp | 63.79 | 68.17 | +4.39 |
| ProGrad | 62.90 | 66.96 | +4.06 |
| MaPLe | 64.80 | 67.79 | +2.99 |
| PromptSRC | 69.09 | 70.38 | +1.29 |
| LoCoOp | 66.63 | 69.05 | +2.42 |

(k) EuroSAT.

|  | MaxLogit | MLS-M | △ |
|---|---|---|---|
| CoOp | 67.94 | 67.83 | -0.11 |
| CoCoOp | 66.87 | 66.76 | -0.10 |
| IVLP | 65.56 | 70.62 | +5.06 |
| KgCoOp | 62.41 | 65.66 | +3.25 |
| ProGrad | 68.96 | 69.71 | +0.75 |
| MaPLe | 71.18 | 72.28 | +1.09 |
| PromptSRC | 75.22 | 74.97 | -0.25 |
| LoCoOp | 66.72 | 67.85 | +1.13 |

(l) UCF101.

|  | MaxLogit | MLS-M | △ |
|---|---|---|---|
| CoOp | 82.17 | 83.65 | +1.48 |
| CoCoOp | 81.32 | 84.02 | +2.70 |
| IVLP | 80.26 | 84.55 | +4.29 |
| KgCoOp | 81.26 | 84.06 | +2.80 |
| ProGrad | 81.21 | 83.71 | +2.50 |
| MaPLe | 80.81 | 84.25 | +3.45 |
| PromptSRC | 83.19 | 85.43 | +2.24 |
| LoCoOp | 76.28 | 82.54 | +6.26 |

(m) CIFAR10.

|  | MaxLogit | MLS-M | △ |
|---|---|---|---|
| CoOp | 84.05 | 85.76 | +1.71 |
| CoCoOp | 89.91 | 92.49 | +2.58 |
| IVLP | 88.17 | 91.10 | +2.93 |
| KgCoOp | 90.26 | 92.77 | +2.52 |
| ProGrad | 83.94 | 89.08 | +5.13 |
| MaPLe | 87.79 | 91.81 | +4.02 |
| PromptSRC | 91.36 | 93.97 | +2.60 |
| LoCoOp | 84.69 | 90.22 | +5.53 |

(n) CIFAR100.

|  | MaxLogit | MLS-M | △ |
|---|---|---|---|
| CoOp | 77.67 | 78.30 | +0.63 |
| CoCoOp | 80.99 | 81.44 | +0.45 |
| IVLP | 84.69 | 88.01 | +3.32 |
| KgCoOp | 80.94 | 81.14 | +0.20 |
| ProGrad | 77.01 | 79.55 | +2.55 |
| MaPLe | 85.35 | 88.35 | +3.01 |
| PromptSRC | 86.82 | 88.78 | +1.96 |
| LoCoOp | 76.03 | 76.67 | +0.64 |

belongs to the ID dataset or the near OOD dataset. In addition, we include CIFAR10 (Krizhevsky et al., 2009) and CIFAR100 (Krizhevsky et al., 2009), which are standard near OOD detection benchmarks (Ren et al., 2021; Fort et al., 2021; Yang et al., 2021; 2022; Zhang et al., 2023b). For CIFAR10 and CIFAR100, we use all classes and evaluate with a test dataset consisting of both CIFAR10 test samples and CIAFR100 test samples, following the literature.

**Models** We use 8 prompt learning models: CoOp (Zhou et al., 2022b), CoCoOp (Zhou et al., 2022a), IVLP (Khattak et al., 2023a), KgCoOp (Yao et al., 2023), ProGrad (Zhu et al., 2023), MaPLe (Khattak et al., 2023a), PromptSRC (Khattak et al., 2023b), and LoCoOp (Miyai et al., 2023), all of which uses ViT-B/16 (Dosovitskiy et al., 2021) for the visual encoder and Transformer (Vaswani et al., 2017) for the text encoder. We follow their training details to train them with 16, 8, 4, 2, and 1-shot settings using 3 random seeds (see Appendix A.2 for implementation details). For each model, we use MaxLogit score, Energy score, and MCM (Ming et al., 2022) score as baselines where MCM is the state-of-the-art score for CLIP. As our focus is on the post-hoc method

Table 3: Near OOD AUROC (↑) and FPR95 (↓) of prompt learning models averaged over 13 datasets using MaxLogit score, Energy score, MLS, and MCM.

| | AUROC ↑ | | | | | FPR95 ↓ | | | | |
|---|---|---|---|---|---|---|---|---|---|---|
| | MaxLogit | Energy | MCM | MLS-M | MLS-E | MaxLogit | Energy | MCM | MLS-M | MLS-E |
| CoOp | 80.74 | 80.44 | 79.41 | **81.84** | 81.71 | 58.23 | 59.45 | 63.70 | **54.85** | 55.37 |
| CoCoOp | 81.09 | 80.53 | 79.28 | **82.74** | 82.74 | 55.78 | 57.49 | 63.76 | **51.67** | 51.90 |
| IVLP | 81.12 | 80.49 | 80.95 | 84.34 | **84.40** | 55.65 | 57.66 | 60.05 | **48.58** | 49.19 |
| KgCoOp | 80.84 | 80.14 | 79.82 | 83.12 | **83.23** | 57.16 | 59.69 | 62.20 | **51.52** | 51.90 |
| ProGrad | 79.77 | 78.79 | 79.91 | **82.35** | 81.93 | 60.07 | 62.74 | 62.89 | **54.02** | 55.34 |
| MaPLe | 81.06 | 80.39 | 80.54 | 83.94 | **83.99** | 55.58 | 57.82 | 60.68 | **48.36** | 48.76 |
| PromptSRC | 83.85 | 83.48 | 82.34 | 85.77 | **85.88** | 49.65 | 51.86 | 55.86 | **44.60** | 45.15 |
| LoCoOp | 77.55 | 75.94 | 79.22 | **81.74** | 81.25 | 64.89 | 69.07 | 63.37 | **55.76** | 57.80 |

directly applicable to fine-tuned prompt learning models without requiring near OOD samples, we exclude other scoring methods that require OOD samples, modifications of training procedures, or architecture changes in models.

## 5.2 EXPERIMENTAL RESULTS

We report average AUROC and false positive rate (FPR) of near OOD samples when true positive rate (TPR) of ID samples is at 95% with 16, 8, 4, 2, and 1-shot settings and 3 randoms seeds. Refer to Appendix A.3 for results of individual few-shot settings with standard deviations across the random seeds. In Table 2, we compare near OOD detection AUROC using the MaxLogit score and MLS-M score. Positive improvements in AUROC are observed in 100 out of 104 evaluations (i.e., 13 datasets×8 models) when MLS-M is used.

On average, the largest improvement was observed with LoCoOp, the recent prompt learning model for OOD detection. The same results for Energy score are shown in Table 12. Similar to MaxLogit score, AUROC is improved in 101 out of 104 evaluations when using Energy score. Refer to Table 6 and Table 13 for FPR95.

Table 3 shows a comparison between MaxLogit score, Energy score, MLS-M, MLS-E, and MCM, in terms of both AUROC and FPR95 averaged across 13 datasets. The MLS-M and Energy score outperform MCM in both AUROC and FPR95. When MLS-M and MLS-E are compared, MLS-M outperforms MLS-E for the half of the models in AUROC and all models in FPR95. This highlights that the proposed scores are better scores for near OOD detection tasks than MCM, and MLS-M is better than MLS-E. Refer to Table 21 and Table 22 for the results of all datasets.

## 6 DISCUSSION

**Comparison with MCM**  As discussed in Section 5.2, MLS outperforms MCM in AUROC and FPR95 averaged across 13 datasets. We notice that MCM outperforms these scores in a few datasets. We analyse this difference from the perspective of dataset distance. As MCM is the state-of-the-art OOD detection score for far OOD detection, MCM is expected to outperform as the dataset distance between an ID test dataset and a near OOD dataset increases. We empirically validate this assumption by measuring the dataset distance by Optimal Transport Dataset Distance (OTDD) (Alvarez-Melis & Fusi, 2020) which is shown in Figure 5. For each dataset, we measure the distance between the ID test dataset and the near OOD test dataset. The density is then plotted, with blue indicating areas where MLS performs better and grey indicating areas where MCM performs better. While MLS outperforms across a wide range of distance, it excels when the distance decreases. MCM, on the other hand, tends to outperform as the distance increases. Nevertheless, MLS outperforms in the majority of cases, resulting in superior average performance.

**Application to MCM**  A straightforward extension of our method is the application of the post-hoc process to the MCM score (i.e., $S_{\text{MLS-MCM}} = S_{\text{MCM}} - \beta \cdot S_{\text{Context}}$). The differences in AUROC and FPR95 with MCM and MLS-MCM are presented in Table 19 and Table 20. While it improves performance with MCM on most datasets, some datasets exhibit a decline in performance. This decline is due to the lack of a positive correlation between MCM and the Context score, which was ob-

served with logit-based scores in Figure 3. Figure 6 further demonstrates the differing relationships between the score and the Context score. This discrepancy primarily arises from the softmax normalisation, where MCM only considers the relative magnitude of logits between classes, allowing non-maximum logits to unexpectedly change the relationship.

**Application to Far OOD Detection** Although our method is not designed for far OOD detection, it can be potentially applied to the area. Following the OOD literature of CLIP (Ming et al., 2022; Miyai et al., 2023; Wang et al., 2023; Jiang et al., 2024), we use ImageNet as an ID dataset and iNaturalist (Van Horn et al., 2018), SUN (Xiao et al., 2010), Places (Zhou et al., 2018), and Texture (Cimpoi et al., 2014) as far OOD datasets. We leverage the fine-tuned models in our experiments and show AUROC results in Table 23 and FPR95 results in Table 24 in Appendix A.3.5. The post-hoc framework is effective for both MaxLogit score and Energy score, improving AUROC and FPR95 in 126 out of 128 evaluations where MLS outperforms MCM in a half of evaluations. This aligns with the near OOD results, being effective across various models and OOD datasets. The limited performance also aligns with our observation in Figure 5 that MCM can be more effective when the dataset distance between ID and OOD datasets increases. However, far OOD detection is beyond our focus.

Figure 5: Density plot of dataset distance between the ID test dataset and near OOD dataset measured with OTDD. Datasets where MLS outperforms are highlighted in blue, while those where MCM outperforms are highlighted in grey.

**Comparisons with More OOD Methods** In our experiments, we intentionally excluded OOD methods that require access to OOD samples during training, retraining the prompt learning models, or those incompatible with the fine-tuned CLIP-based prompt learning models. This is because our main contribution is a new score specifically designed for few-shot prompt learning models without these requirements. For example, ODIN (Liang et al., 2018) is not a suitable baseline as it requires access to OOD samples for hyperparameter tuning. Similarly, LogitNorm (Wei et al., 2022) needs to be trained with a specific loss function, requiring the retraining of prompt learning models with modified training objectives. Distance-based scores, such as Mahalanobis distance (Lee et al., 2018) and relative Mahalanobis distance (RMD) (Ren et al., 2021), were originally designed for traditional classifiers and are therefore incompatible with few-shot prompt learning models. For example, to compute RMD in our case, only image features can be used while the textual features are ignored. As the prompt learning models do not update their image encoders while fine-tuning, RMD will be the same regardless of prompt learning model used. Therefore, these scores are not suitable for our problem. However, we provide comparisons with LogitNorm and RMD in Appendix A.3.8 for readers interested in the comparisons.

# 7 CONCLUSION

In this work, we address few-shot near OOD detection of CLIP-based prompt learning models. To enhance existing logit-based scores, we propose a simple and fast post-hoc method applicable to any prompt learning model. Without changing training procedures of the existing models nor compromising classification accuracy, our method effectively enhances near OOD AUROC and lowers FPR95 for 8 recent prompt learning models across 13 real-world datasets. While our method is broadly applicable across various prompt learning models, the degree of improvement can vary depending on the underlying model characteristics. Some models may not see as substantial a benefit, particularly if their inherent logit distributions already exhibit strong separability between ID and OOD samples.

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

# A  APPENDIX

## A.1  PROOF OF LEMMA

We provide the proof of Lemma 3.1. For completeness of proof, we duplicate the lemma here.

**Lemma A.1.** *Given $N$ scalar observations $\{\hat{x}_i\}_{i=1}^N$ and $\{\hat{y}_i\}_{i=1}^N$, we define two variables $x = \hat{x}$ and $y = \hat{y} - \beta \cdot \hat{x}$. The scale parameter $\beta$ that zeros out the covariance of two variables (i.e., the off-diagonals of a covariance matrix) which is approximated by maximum likelihood estimation is:*

$$\beta = \frac{\sum_{i=1}^N (\hat{x}_i - \mu_{\hat{x}})(\hat{y}_i - \mu_{\hat{y}})}{\sum_{i=1}^N (\hat{x}_i - \mu_{\hat{x}})^2} \tag{9}$$

*where $\mu_{\hat{x}} = \frac{1}{N} \sum_{i=1}^N \hat{x}_i$ and $\mu_{\hat{y}} = \frac{1}{N} \sum_{i=1}^N \hat{y}_i$.*

*Proof.* It is well known that maximum likelihood estimation (MLE) of bivariate normal distribution for $N$ observations of variables $x$ and $y$ results in (Bishop, 2013):

$$\mu_x = \frac{1}{N} \sum_i x_i, \quad \mu_y = \frac{1}{N} \sum_i y_i \tag{10}$$

$$\Sigma = \begin{bmatrix} \sigma_{xx} & \sigma_{xy} \\ \sigma_{xy} & \sigma_{yy} \end{bmatrix} = \frac{1}{N} \sum_i \left( \begin{bmatrix} x_i \\ y_i \end{bmatrix} - \begin{bmatrix} \mu_x \\ \mu_y \end{bmatrix} \right) \left( \begin{bmatrix} x_i \\ y_i \end{bmatrix} - \begin{bmatrix} \mu_x \\ \mu_y \end{bmatrix} \right)^T \tag{11}$$

$$\sigma_{xy} = \frac{1}{N} \sum_i (x_i - \mu_x)(y_i - \mu_y) \tag{12}$$

where $\mu_x$ and $\mu_y$ are the means of $x$ and $y$, and $\Sigma$ is the covariance matrix. We let $x = \hat{x}$ and $y = \hat{y} - \beta \cdot \hat{x}$ and find $\beta$ that makes $\sigma_{xy} = 0$. By rewriting $\sigma_{xy}$ in terms of $\hat{x}$ and $\hat{y}$, we obtain $\beta$ as:

$$\sigma_{xy} = \frac{1}{N} \sum_i (\hat{x}_i - \mu_{\hat{x}})(\hat{y}_i - \beta \cdot \hat{x}_i - \mu_{\hat{y}} + \beta \cdot \mu_{\hat{x}}) = 0 \tag{13}$$

$$\beta = \frac{\sum_i (\hat{x}_i - \mu_{\hat{x}})(\hat{y}_i - \mu_{\hat{y}})}{\sum_i (\hat{x}_i - \mu_{\hat{x}})^2} \tag{14}$$

The resulting $\beta$ is the ratio of covariance of $\hat{x}$ and $\hat{y}$ to variance of $\hat{x}$.  $\square$

## A.2  IMPLEMENTATION DETAILS

We follow the officially released training guidelines for each prompt learning model using the same configuration files. The only additional line of code required is `beta=(((y-y.mean())*(x-x.mean())).sum())/(((x-x.mean())**2).sum())` to estimate the margin scale in Eq.(8). The overall algorithm is summarised in Algorithm 1. Table 4 shows common hyperparameters which are the number of epochs, batch size, and context vectors initialisation. Refer to their officially released codes for other model-specific hyperparameters. All models were trained on a single NVIDIA GeForce RTX 3090 GPU with PyTorch framework. The temperature scaling is 0.01 for the Energy score and 1 for the MCM score.

---

**Algorithm 1:** MLS Computation

---

**Input:** Few-shot training dataset of $N$ image-label pairs of $\{I_i, y_i\}_{i=1}^N$ where $y_i \in \{1, \cdots, K\}$ with $K$ classes, a test image $I_{test}$, a fine-tuned prompt learning model with learned context vectors $V$.

1 **for** $I_i, y_i$ **do**
2     Compute and store MaxLogit score $S_{\text{MaxLogit}}$ or Energy score $S_{\text{Energy}}$ by Eq.(3) and Eq.(2).
3     Compute and store Context score $S_{\text{Context}}$.
4 **end for**
5 Estimate margin scale $\beta$ by Eq.(8)
6 Compute MLS by Eq.(7) and Eq.(6)

---

Table 4: Training details of the prompt learning models..

|  | # Epochs | Batch Size | Context Vectors Initialisation |
|---|---|---|---|
| CoOp | 50 (ImageNet) 200 (Others) | 32 | |
| CoCoOp | 10 | 1 | |
| IVLP | 5 | 4 | |
| KgCoOp | 100 | 128 | "a photo of a" |
| ProGrad | 200 | 32 | |
| MaPLe | 5 | 4 | |
| PromptSRC | 20 | 4 | |
| LoCoOp | 50 | 32 | 16 vectors drawn from $\mathcal{N}(0, 0.02)$ |

### A.3 ADDITIONAL EXPERIMENTAL RESULTS

We provide additional experimental results other than the results in the main section.

### A.3.1 MAXLOGIT SCORE

We provide average AUROC and FPR95 across 13 datasets using MaxLogit in Table 5 and Table 6. Note that Table 5 is duplicated for completeness of this Section. From Table 7 to Table 11, average AUROC across 3 random seeds with standard deviations are reported for each few-shot setting. While overall improvements are observed across different few-shot settings, slight performance degradations are observed in EuroSAT and CIAFR10 at 1-shot. This is because of insufficient training samples used to find the optimal margin scale where EuroSAT and CIFAR10 have 5 classes and 10 classes in their training datasets respectively.

Table 5: Near OOD AUROC (↑) of prompt learning models over 13 datasets using the MaxLogit score and MLS-M.

(a) Average over 13 datasets.

|  | MaxLogit | MLS-M | △ |
|---|---|---|---|
| CoOp | 80.74 | 81.84 | +1.09 |
| CoCoOp | 81.09 | 82.74 | +1.65 |
| IVLP | 81.12 | 84.34 | +3.23 |
| KgCoOp | 80.84 | 83.12 | +2.28 |
| ProGrad | 79.77 | 82.35 | +2.58 |
| MaPLe | 81.06 | 83.94 | +2.88 |
| PromptSRC | 83.85 | 85.77 | +1.92 |
| LoCoOp | 77.55 | 81.74 | +4.18 |

(b) ImageNet.

|  | MaxLogit | MLS-M | △ |
|---|---|---|---|
| CoOp | 93.78 | 94.66 | +0.88 |
| CoCoOp | 94.85 | 95.14 | +0.29 |
| IVLP | 94.55 | 94.70 | +0.15 |
| KgCoOp | 94.21 | 94.21 | +0.01 |
| ProGrad | 93.62 | 94.67 | +1.04 |
| MaPLe | 94.20 | 94.35 | +0.16 |
| PromptSRC | 94.52 | 95.32 | +0.80 |
| LoCoOp | 93.10 | 94.56 | +1.46 |

(c) Caltech101.

|  | MaxLogit | MLS-M | △ |
|---|---|---|---|
| CoOp | 88.27 | 90.12 | +1.85 |
| CoCoOp | 85.80 | 89.02 | +3.22 |
| IVLP | 85.50 | 90.53 | +5.03 |
| KgCoOp | 83.64 | 90.06 | +6.42 |
| ProGrad | 82.96 | 88.85 | +5.89 |
| MaPLe | 85.91 | 91.53 | +5.62 |
| PromptSRC | 84.94 | 90.56 | +5.62 |
| LoCoOp | 76.08 | 87.75 | +11.67 |

(d) OxfordPets.

|  | MaxLogit | MLS-M | △ |
|---|---|---|---|
| CoOp | 86.22 | 88.73 | +2.52 |
| CoCoOp | 89.58 | 92.28 | +2.69 |
| IVLP | 88.84 | 91.94 | +3.10 |
| KgCoOp | 89.94 | 92.64 | +2.69 |
| ProGrad | 87.82 | 89.60 | +1.78 |
| MaPLe | 87.40 | 91.00 | +3.60 |
| PromptSRC | 90.80 | 93.45 | +2.65 |
| LoCoOp | 84.44 | 89.19 | +4.75 |

(e) StanfordCars.

|  | MaxLogit | MLS-M | △ |
|---|---|---|---|
| CoOp | 91.36 | 91.59 | +0.22 |
| CoCoOp | 92.43 | 92.99 | +0.57 |
| IVLP | 90.43 | 92.98 | +2.56 |
| KgCoOp | 92.77 | 93.27 | +0.51 |
| ProGrad | 91.52 | 92.63 | +1.11 |
| MaPLe | 91.39 | 92.85 | +1.47 |
| PromptSRC | 92.88 | 94.24 | +1.35 |
| LoCoOp | 88.24 | 91.94 | +3.70 |

(f) Flowers102.

|  | MaxLogit | MLS-M | △ |
|---|---|---|---|
| CoOp | 90.83 | 91.99 | +1.16 |
| CoCoOp | 87.93 | 89.41 | +1.48 |
| IVLP | 86.20 | 88.45 | +2.25 |
| KgCoOp | 87.61 | 91.12 | +3.52 |
| ProGrad | 89.27 | 91.41 | +2.14 |
| MaPLe | 86.05 | 88.34 | +2.29 |
| PromptSRC | 91.10 | 92.61 | +1.51 |
| LoCoOp | 86.17 | 88.59 | +2.42 |

(g) Food101

|  | MaxLogit | MLS-M | △ |
|---|---|---|---|
| CoOp | 86.70 | 87.91 | +1.21 |
| CoCoOp | 90.52 | 91.63 | +1.10 |
| IVLP | 89.70 | 91.87 | +2.18 |
| KgCoOp | 89.87 | 92.12 | +2.25 |
| ProGrad | 88.60 | 91.05 | +2.45 |
| MaPLe | 89.10 | 92.00 | +2.89 |
| PromptSRC | 90.94 | 92.11 | +1.17 |
| LoCoOp | 84.87 | 90.12 | +5.25 |

(h) FGVCAircraft.

|  | MaxLogit | MLS-M | △ |
|---|---|---|---|
| CoOp | 55.99 | 56.97 | +0.97 |
| CoCoOp | 52.60 | 55.04 | +2.45 |
| IVLP | 58.47 | 64.16 | +5.69 |
| KgCoOp | 57.82 | 57.46 | -0.36 |
| ProGrad | 53.69 | 55.67 | +1.97 |
| MaPLe | 52.18 | 56.93 | +4.74 |
| PromptSRC | 60.63 | 62.50 | +1.87 |
| LoCoOp | 50.99 | 56.12 | +5.13 |

(i) SUN397.

|  | MaxLogit | MLS-M | △ |
|---|---|---|---|
| CoOp | 75.78 | 76.75 | +0.97 |
| CoCoOp | 76.32 | 78.29 | +1.97 |
| IVLP | 77.13 | 79.60 | +2.46 |
| KgCoOp | 76.45 | 77.91 | +1.46 |
| ProGrad | 75.52 | 77.67 | +2.15 |
| MaPLe | 77.62 | 79.73 | +2.11 |
| PromptSRC | 78.51 | 80.70 | +2.19 |
| LoCoOp | 73.97 | 78.00 | +4.02 |

(j) DTD.

|  | MaxLogit | MLS-M | △ |
|---|---|---|---|
| CoOp | 68.90 | 69.60 | +0.69 |
| CoCoOp | 65.10 | 67.17 | +2.07 |
| IVLP | 64.99 | 67.93 | +2.94 |
| KgCoOp | 63.79 | 68.17 | +4.39 |
| ProGrad | 62.90 | 66.96 | +4.06 |
| MaPLe | 64.80 | 67.79 | +2.99 |
| PromptSRC | 69.09 | 70.38 | +1.29 |
| LoCoOp | 66.63 | 69.05 | +2.42 |

(k) EuroSAT.

|  | MaxLogit | MLS-M | △ |
|---|---|---|---|
| CoOp | 67.94 | 67.83 | -0.11 |
| CoCoOp | 66.87 | 66.76 | -0.10 |
| IVLP | 65.56 | 70.62 | +5.06 |
| KgCoOp | 62.41 | 65.66 | +3.25 |
| ProGrad | 68.96 | 69.71 | +0.75 |
| MaPLe | 71.18 | 72.28 | +1.09 |
| PromptSRC | 75.22 | 74.97 | -0.25 |
| LoCoOp | 66.72 | 67.85 | +1.13 |

(l) UCF101.

|  | MaxLogit | MLS-M | △ |
|---|---|---|---|
| CoOp | 82.17 | 83.65 | +1.48 |
| CoCoOp | 81.32 | 84.02 | +2.70 |
| IVLP | 80.26 | 84.55 | +4.29 |
| KgCoOp | 81.26 | 84.06 | +2.80 |
| ProGrad | 81.21 | 83.71 | +2.50 |
| MaPLe | 80.81 | 84.25 | +3.45 |
| PromptSRC | 83.19 | 85.43 | +2.24 |
| LoCoOp | 76.28 | 82.54 | +6.26 |

(m) CIFAR10.

|  | MaxLogit | MLS-M | △ |
|---|---|---|---|
| CoOp | 84.05 | 85.76 | +1.71 |
| CoCoOp | 89.91 | 92.49 | +2.58 |
| IVLP | 88.17 | 91.10 | +2.93 |
| KgCoOp | 90.26 | 92.77 | +2.52 |
| ProGrad | 83.94 | 89.08 | +5.13 |
| MaPLe | 87.79 | 91.81 | +4.02 |
| PromptSRC | 91.36 | 93.97 | +2.60 |
| LoCoOp | 84.69 | 90.22 | +5.53 |

(n) CIFAR100.

|  | MaxLogit | MLS-M | △ |
|---|---|---|---|
| CoOp | 77.67 | 78.30 | +0.63 |
| CoCoOp | 80.99 | 81.44 | +0.45 |
| IVLP | 84.69 | 88.01 | +3.32 |
| KgCoOp | 80.94 | 81.14 | +0.20 |
| ProGrad | 77.01 | 79.55 | +2.55 |
| MaPLe | 85.35 | 88.35 | +3.01 |
| PromptSRC | 86.82 | 88.78 | +1.96 |
| LoCoOp | 76.03 | 76.67 | +0.64 |

Table 6: Near OOD FPR95 (↓) of prompt learning models over 13 datasets using the MaxLogit score and MLS-M.

(a) Average over 13 datasets.

|  | MaxLogit | MLS-M | △ |
|---|---|---|---|
| CoOp | 58.23 | 54.85 | -3.38 |
| CoCoOp | 55.78 | 51.67 | -4.11 |
| IVLP | 55.65 | 48.58 | -7.07 |
| KgCoOp | 57.16 | 51.52 | -5.64 |
| ProGrad | 60.07 | 54.02 | -6.05 |
| MaPLe | 55.58 | 48.36 | -7.22 |
| PromptSRC | 49.65 | 44.60 | -5.05 |
| LoCoOp | 64.89 | 55.76 | -9.12 |

(b) ImageNet.

|  | MaxLogit | MLS-M | △ |
|---|---|---|---|
| CoOp | 31.02 | 26.40 | -4.62 |
| CoCoOp | 26.76 | 23.85 | -2.92 |
| IVLP | 27.23 | 24.54 | -2.70 |
| KgCoOp | 29.84 | 27.45 | -2.39 |
| ProGrad | 32.73 | 27.40 | -5.33 |
| MaPLe | 29.87 | 26.09 | -3.79 |
| PromptSRC | 27.89 | 22.82 | -5.07 |
| LoCoOp | 36.95 | 28.94 | -8.01 |

(c) Caltech101.

|  | MaxLogit | MLS-M | △ |
|---|---|---|---|
| CoOp | 38.42 | 29.90 | -8.52 |
| CoCoOp | 42.22 | 33.02 | -9.20 |
| IVLP | 45.94 | 30.97 | -14.97 |
| KgCoOp | 49.33 | 28.80 | -20.53 |
| ProGrad | 56.38 | 39.35 | -17.02 |
| MaPLe | 44.44 | 28.30 | -16.14 |
| PromptSRC | 43.79 | 25.83 | -17.96 |
| LoCoOp | 69.02 | 37.02 | -32.00 |

(d) OxfordPets.

|  | MaxLogit | MLS-M | △ |
|---|---|---|---|
| CoOp | 51.60 | 45.90 | -5.70 |
| CoCoOp | 42.47 | 35.89 | -6.58 |
| IVLP | 48.28 | 40.88 | -7.40 |
| KgCoOp | 47.06 | 35.52 | -11.54 |
| ProGrad | 50.75 | 51.94 | +1.19 |
| MaPLe | 50.86 | 42.87 | -7.99 |
| PromptSRC | 42.14 | 38.08 | -4.06 |
| LoCoOp | 54.57 | 48.65 | -5.92 |

(e) StanfordCars.

|  | MaxLogit | MLS-M | △ |
|---|---|---|---|
| CoOp | 32.58 | 31.62 | -0.97 |
| CoCoOp | 28.76 | 27.18 | -1.58 |
| IVLP | 33.89 | 26.07 | -7.82 |
| KgCoOp | 29.06 | 27.77 | -1.29 |
| ProGrad | 32.44 | 28.42 | -4.02 |
| MaPLe | 30.75 | 26.05 | -4.70 |
| PromptSRC | 26.27 | 22.88 | -3.38 |
| LoCoOp | 41.93 | 31.83 | -10.10 |

(f) Flowers102.

|  | MaxLogit | MLS-M | △ |
|---|---|---|---|
| CoOp | 40.02 | 36.14 | -3.87 |
| CoCoOp | 50.27 | 45.51 | -4.76 |
| IVLP | 54.62 | 49.00 | -5.62 |
| KgCoOp | 56.18 | 43.15 | -13.03 |
| ProGrad | 44.28 | 38.84 | -5.44 |
| MaPLe | 54.00 | 47.94 | -6.07 |
| PromptSRC | 38.96 | 34.33 | -4.63 |
| LoCoOp | 52.83 | 46.72 | -6.12 |

(g) Food101

|  | MaxLogit | MLS-M | △ |
|---|---|---|---|
| CoOp | 54.00 | 49.48 | -4.52 |
| CoCoOp | 41.85 | 37.57 | -4.28 |
| IVLP | 44.36 | 35.63 | -8.74 |
| KgCoOp | 44.42 | 35.98 | -8.44 |
| ProGrad | 48.58 | 40.02 | -8.57 |
| MaPLe | 45.70 | 35.26 | -10.44 |
| PromptSRC | 39.21 | 34.61 | -4.60 |
| LoCoOp | 58.93 | 42.34 | -16.59 |

(h) FGVCAircraft.

|  | MaxLogit | MLS-M | △ |
|---|---|---|---|
| CoOp | 83.67 | 82.75 | -0.92 |
| CoCoOp | 84.31 | 81.95 | -2.36 |
| IVLP | 79.19 | 75.35 | -3.84 |
| KgCoOp | 79.80 | 79.98 | +0.18 |
| ProGrad | 84.79 | 81.38 | -3.41 |
| MaPLe | 81.98 | 80.73 | -1.25 |
| PromptSRC | 76.62 | 75.53 | -1.08 |
| LoCoOp | 86.36 | 83.58 | -2.78 |

(i) SUN397.

|  | MaxLogit | MLS-M | △ |
|---|---|---|---|
| CoOp | 71.81 | 69.50 | -2.31 |
| CoCoOp | 70.74 | 65.62 | -5.12 |
| IVLP | 69.31 | 63.76 | -5.55 |
| KgCoOp | 70.46 | 67.17 | -3.29 |
| ProGrad | 73.83 | 68.12 | -5.71 |
| MaPLe | 68.72 | 63.34 | -5.38 |
| PromptSRC | 66.85 | 61.55 | -5.30 |
| LoCoOp | 76.47 | 67.24 | -9.24 |

(j) DTD.

|  | MaxLogit | MLS-M | △ |
|---|---|---|---|
| CoOp | 85.34 | 85.16 | -0.18 |
| CoCoOp | 88.48 | 87.50 | -0.98 |
| IVLP | 87.08 | 87.11 | +0.03 |
| KgCoOp | 88.52 | 86.93 | -1.59 |
| ProGrad | 87.78 | 87.55 | -0.23 |
| MaPLe | 88.85 | 86.81 | -2.04 |
| PromptSRC | 84.52 | 83.56 | -0.97 |
| LoCoOp | 84.51 | 83.94 | -0.57 |

(k) EuroSAT.

|  | MaxLogit | MLS-M | △ |
|---|---|---|---|
| CoOp | 85.12 | 85.20 | +0.08 |
| CoCoOp | 82.25 | 82.38 | +0.13 |
| IVLP | 81.16 | 73.84 | -7.32 |
| KgCoOp | 82.08 | 82.30 | +0.22 |
| ProGrad | 81.61 | 82.31 | +0.70 |
| MaPLe | 75.22 | 68.41 | -6.81 |
| PromptSRC | 65.95 | 65.83 | -0.12 |
| LoCoOp | 81.91 | 80.88 | -1.03 |

(l) UCF101.

|  | MaxLogit | MLS-M | △ |
|---|---|---|---|
| CoOp | 55.64 | 50.12 | -5.53 |
| CoCoOp | 57.04 | 50.37 | -6.67 |
| IVLP | 56.96 | 47.71 | -9.26 |
| KgCoOp | 54.77 | 50.16 | -4.61 |
| ProGrad | 57.39 | 48.84 | -8.55 |
| MaPLe | 57.16 | 49.27 | -7.89 |
| PromptSRC | 51.62 | 46.17 | -5.45 |
| LoCoOp | 67.38 | 55.38 | -11.99 |

(m) CIFAR10.

|  | MaxLogit | MLS-M | △ |
|---|---|---|---|
| CoOp | 53.49 | 49.45 | -4.04 |
| CoCoOp | 37.07 | 29.79 | -7.28 |
| IVLP | 39.00 | 30.87 | -8.13 |
| KgCoOp | 38.30 | 30.85 | -7.45 |
| ProGrad | 52.48 | 40.59 | -11.90 |
| MaPLe | 40.75 | 30.00 | -10.75 |
| PromptSRC | 30.54 | 22.74 | -7.80 |
| LoCoOp | 51.96 | 36.97 | -14.99 |

(n) CIFAR100.

|  | MaxLogit | MLS-M | △ |
|---|---|---|---|
| CoOp | 74.33 | 71.48 | -2.85 |
| CoCoOp | 72.96 | 71.09 | -1.88 |
| IVLP | 56.39 | 45.86 | -10.54 |
| KgCoOp | 73.27 | 73.66 | +0.39 |
| ProGrad | 77.89 | 67.49 | -10.40 |
| MaPLe | 54.19 | 43.60 | -10.59 |
| PromptSRC | 51.05 | 45.82 | -5.23 |
| LoCoOp | 80.71 | 81.44 | +0.73 |

Table 7: Near OOD AUROC (↑) of prompt learning models over 13 datasets using the MaxLogit score and MLS-M with 16-shots.

(a) Average over 13 datasets.

|           | MaxLogit    | MLS-M       | △           |
|-----------|-------------|-------------|-------------|
| CoOp      | 82.82±10.81 | 83.86±10.79 | +1.03±1.08  |
| CoCoOp    | 81.51±12.87 | 83.65±11.76 | +2.14±2.32  |
| IVLP      | 83.46±10.49 | 87.48±8.02  | +4.02±3.96  |
| KgCoOp    | 81.41±12.12 | 84.08±11.38 | +2.67±2.38  |
| ProGrad   | 81.76±11.56 | 83.83±11.66 | +2.07±2.43  |
| MaPLe     | 83.83±10.89 | 86.35±10.06 | +2.52±1.97  |
| PromptSRC | 85.76±9.23  | 87.52±9.07  | +1.76±1.55  |
| LoCoOp    | 79.26±11.23 | 82.30±12.45 | +3.05±3.61  |

(b) ImageNet.

|           | MaxLogit   | MLS-M      | △          |
|-----------|------------|------------|------------|
| CoOp      | 93.81±0.23 | 95.38±0.27 | +1.57±0.05 |
| CoCoOp    | 94.90±0.30 | 95.15±0.80 | +0.25±0.94 |
| IVLP      | 94.46±0.15 | 94.58±0.47 | +0.12±0.42 |
| KgCoOp    | 94.31±0.05 | 94.18±0.27 | -0.14±0.32 |
| ProGrad   | 94.14±0.18 | 94.97±0.06 | +0.82±0.15 |
| MaPLe     | 94.66±0.51 | 94.70±0.55 | +0.04±0.29 |
| PromptSRC | 94.66±0.21 | 95.60±0.18 | +0.93±0.25 |
| LoCoOp    | 93.70±0.10 | 94.96±0.36 | +1.26±0.45 |

(c) Caltech101.

|           | MaxLogit   | MLS-M      | △          |
|-----------|------------|------------|------------|
| CoOp      | 89.75±0.81 | 91.38±1.11 | +1.63±1.18 |
| CoCoOp    | 87.12±1.14 | 89.94±1.19 | +2.82±1.92 |
| IVLP      | 87.69±1.59 | 93.21±1.75 | +5.52±1.82 |
| KgCoOp    | 84.05±0.05 | 90.34±0.33 | +6.29±0.38 |
| ProGrad   | 82.24±2.23 | 89.63±0.49 | +7.40±2.15 |
| MaPLe     | 87.88±0.79 | 93.06±1.46 | +5.18±1.97 |
| PromptSRC | 84.68±0.22 | 90.55±0.15 | +5.86±0.14 |
| LoCoOp    | 78.82±1.57 | 88.71±0.60 | +9.89±1.44 |

(d) OxfordPets.

|           | MaxLogit   | MLS-M      | △          |
|-----------|------------|------------|------------|
| CoOp      | 85.22±1.19 | 86.66±1.64 | +1.45±0.46 |
| CoCoOp    | 90.93±0.76 | 93.54±0.23 | +2.61±0.69 |
| IVLP      | 91.15±1.82 | 93.74±1.33 | +2.59±0.73 |
| KgCoOp    | 90.28±0.28 | 92.16±0.97 | +1.88±0.80 |
| ProGrad   | 87.42±1.41 | 88.59±0.73 | +1.17±0.98 |
| MaPLe     | 90.02±0.45 | 94.16±1.27 | +4.14±1.28 |
| PromptSRC | 91.64±0.33 | 94.70±0.45 | +3.06±0.19 |
| LoCoOp    | 84.20±3.06 | 89.18±1.21 | +4.98±2.46 |

(e) StanfordCars.

|           | MaxLogit   | MLS-M      | △          |
|-----------|------------|------------|------------|
| CoOp      | 93.53±0.21 | 93.44±0.22 | -0.09±0.16 |
| CoCoOp    | 91.29±1.69 | 92.82±0.80 | +1.52±1.08 |
| IVLP      | 90.49±1.37 | 93.21±1.09 | +2.72±0.33 |
| KgCoOp    | 92.95±0.20 | 93.47±0.06 | +0.51±0.21 |
| ProGrad   | 93.25±0.06 | 94.04±0.13 | +0.80±0.08 |
| MaPLe     | 91.44±1.47 | 93.23±0.33 | +1.78±1.50 |
| PromptSRC | 93.84±0.18 | 95.35±0.18 | +1.51±0.13 |
| LoCoOp    | 89.08±1.19 | 92.10±1.80 | +3.02±0.94 |

(f) Flowers102.

|           | MaxLogit   | MLS-M      | △          |
|-----------|------------|------------|------------|
| CoOp      | 94.30±0.63 | 94.66±0.57 | +0.36±0.07 |
| CoCoOp    | 89.42±0.75 | 91.10±0.05 | +1.69±0.80 |
| IVLP      | 90.63±2.80 | 92.62±1.65 | +1.99±1.16 |
| KgCoOp    | 90.03±0.40 | 93.09±0.43 | +3.05±0.37 |
| ProGrad   | 90.72±0.89 | 93.03±0.23 | +2.32±0.68 |
| MaPLe     | 89.76±0.79 | 91.43±1.17 | +1.67±0.58 |
| PromptSRC | 94.40±0.18 | 95.69±0.07 | +1.29±0.24 |
| LoCoOp    | 87.05±1.89 | 89.98±0.29 | +2.92±2.19 |

(g) Food101

|           | MaxLogit   | MLS-M      | △          |
|-----------|------------|------------|------------|
| CoOp      | 88.67±0.17 | 90.13±0.18 | +1.46±0.13 |
| CoCoOp    | 90.54±0.55 | 92.43±0.76 | +1.89±0.24 |
| IVLP      | 90.40±0.78 | 92.81±0.52 | +2.41±1.29 |
| KgCoOp    | 90.26±0.03 | 92.80±0.05 | +2.54±0.04 |
| ProGrad   | 90.36±0.33 | 92.68±0.21 | +2.31±0.13 |
| MaPLe     | 90.79±0.89 | 93.10±0.49 | +2.30±0.51 |
| PromptSRC | 91.36±0.25 | 92.63±0.22 | +1.26±0.07 |
| LoCoOp    | 85.50±1.17 | 91.49±0.65 | +6.00±1.61 |

(h) FGVCAircraft.

|           | MaxLogit   | MLS-M      | △           |
|-----------|------------|------------|-------------|
| CoOp      | 55.40±1.81 | 56.30±1.88 | +0.90±1.39  |
| CoCoOp    | 49.94±2.22 | 57.15±3.04 | +7.21±4.00  |
| IVLP      | 59.68±1.37 | 71.67±1.42 | +11.99±0.11 |
| KgCoOp    | 57.08±0.46 | 57.12±1.48 | +0.04±1.10  |
| ProGrad   | 54.58±0.63 | 54.13±4.16 | -0.45±4.07  |
| MaPLe     | 56.37±4.94 | 61.55±5.18 | +5.18±0.32  |
| PromptSRC | 64.57±1.65 | 66.79±1.95 | +2.22±1.25  |
| LoCoOp    | 52.39±1.93 | 49.20±3.56 | -3.19±2.24  |

(i) SUN397.

|           | MaxLogit   | MLS-M      | △          |
|-----------|------------|------------|------------|
| CoOp      | 78.28±0.50 | 79.55±0.69 | +1.27±0.28 |
| CoCoOp    | 77.26±0.48 | 79.28±0.50 | +2.01±0.35 |
| IVLP      | 77.80±0.44 | 80.69±0.27 | +2.89±0.71 |
| KgCoOp    | 77.35±0.10 | 78.83±0.18 | +1.47±0.23 |
| ProGrad   | 77.06±1.03 | 79.49±0.85 | +2.42±0.28 |
| MaPLe     | 78.20±1.74 | 80.68±0.66 | +2.48±1.12 |
| PromptSRC | 79.60±0.10 | 82.05±0.47 | +2.45±0.37 |
| LoCoOp    | 75.80±0.33 | 79.85±0.46 | +4.05±0.72 |

(j) DTD.

|           | MaxLogit   | MLS-M      | △          |
|-----------|------------|------------|------------|
| CoOp      | 74.05±1.15 | 74.30±1.40 | +0.24±0.27 |
| CoCoOp    | 66.24±1.28 | 69.63±1.46 | +3.40±1.75 |
| IVLP      | 67.09±1.68 | 71.16±0.76 | +4.07±1.01 |
| KgCoOp    | 65.47±0.65 | 71.38±0.52 | +5.92±0.31 |
| ProGrad   | 63.89±1.41 | 68.36±1.45 | +4.47±1.16 |
| MaPLe     | 67.14±2.96 | 70.89±2.12 | +3.75±1.77 |
| PromptSRC | 71.98±0.74 | 73.59±0.71 | +1.62±0.61 |
| LoCoOp    | 68.15±0.66 | 72.60±0.68 | +4.45±0.74 |

(k) EuroSAT.

|           | MaxLogit   | MLS-M      | △           |
|-----------|------------|------------|-------------|
| CoOp      | 71.85±1.25 | 73.46±2.66 | +1.61±1.55  |
| CoCoOp    | 68.61±2.90 | 68.29±3.18 | -0.32±0.36  |
| IVLP      | 74.19±2.37 | 86.13±3.09 | +11.94±3.57 |
| KgCoOp    | 62.66±0.41 | 69.75±1.17 | +7.09±0.89  |
| ProGrad   | 75.98±1.94 | 76.16±1.80 | +0.18±0.19  |
| MaPLe     | 80.80±0.94 | 82.17±1.90 | +1.37±1.12  |
| PromptSRC | 80.02±2.66 | 79.75±2.54 | -0.27±0.26  |
| LoCoOp    | 68.62±5.75 | 69.66±5.80 | +1.04±0.55  |

(l) UCF101.

|           | MaxLogit   | MLS-M      | △          |
|-----------|------------|------------|------------|
| CoOp      | 83.64±0.49 | 85.37±0.51 | +1.73±0.95 |
| CoCoOp    | 82.76±1.14 | 85.17±0.81 | +2.41±0.56 |
| IVLP      | 82.48±2.21 | 86.65±0.37 | +4.16±1.93 |
| KgCoOp    | 82.08±0.13 | 85.01±0.62 | +2.92±0.51 |
| ProGrad   | 82.06±0.46 | 84.03±0.63 | +1.97±0.22 |
| MaPLe     | 83.00±0.94 | 84.81±1.10 | +1.81±0.27 |
| PromptSRC | 84.17±0.61 | 86.03±0.67 | +1.85±0.23 |
| LoCoOp    | 78.50±1.08 | 84.35±1.15 | +5.85±2.01 |

(m) CIFAR10.

|           | MaxLogit   | MLS-M      | △          |
|-----------|------------|------------|------------|
| CoOp      | 90.39±0.84 | 90.60±0.76 | +0.20±0.26 |
| CoCoOp    | 93.84±0.18 | 93.95±0.38 | +0.10±0.40 |
| IVLP      | 93.01±1.30 | 93.10±1.66 | +0.09±0.45 |
| KgCoOp    | 93.46±0.12 | 93.86±0.12 | +0.40±0.03 |
| ProGrad   | 92.94±0.40 | 93.30±0.28 | +0.36±0.16 |
| MaPLe     | 94.24±0.58 | 94.26±0.79 | +0.02±0.43 |
| PromptSRC | 95.13±0.09 | 94.88±0.35 | -0.24±0.29 |
| LoCoOp    | 92.34±0.70 | 91.13±0.46 | -1.21±0.59 |

(n) CIFAR100.

|           | MaxLogit   | MLS-M      | △          |
|-----------|------------|------------|------------|
| CoOp      | 77.81±0.91 | 78.93±1.06 | +1.12±1.78 |
| CoCoOp    | 76.76±1.06 | 79.06±1.14 | +2.31±0.94 |
| IVLP      | 85.87±0.90 | 87.68±2.39 | +1.81±1.55 |
| KgCoOp    | 78.28±0.43 | 81.07±0.32 | +2.79±0.51 |
| ProGrad   | 78.21±0.48 | 81.40±0.68 | +3.19±0.47 |
| MaPLe     | 85.48±0.77 | 88.50±2.08 | +3.01±1.34 |
| PromptSRC | 88.83±0.25 | 90.21±0.13 | +1.38±0.38 |
| LoCoOp    | 76.19±1.57 | 76.73±1.77 | +0.54±1.17 |

Table 8: Near OOD AUROC (↑) of prompt learning models over 13 datasets using the MaxLogit score and MLS-M with 8-shots.

### (a) Average over 13 datasets.

|  | MaxLogit | MLS-M | △ |
|---|---|---|---|
| CoOp | 81.60±11.10 | 82.84±11.08 | +1.24±1.01 |
| CoCoOp | 81.48±12.41 | 83.33±11.93 | +1.85±1.55 |
| IVLP | 82.20±11.80 | 85.11±10.87 | +2.91±3.80 |
| KgCoOp | 80.97±12.22 | 83.38±11.86 | +2.40±2.18 |
| ProGrad | 80.39±12.37 | 82.96±12.03 | +2.58±2.37 |
| MaPLe | 82.67±12.39 | 85.32±12.13 | +2.64±3.29 |
| PromptSRC | 85.44±9.45 | 87.36±8.88 | +1.92±1.70 |
| LoCoOp | 79.25±10.77 | 82.96±10.27 | +3.71±4.19 |

### (b) ImageNet.

|  | MaxLogit | MLS-M | △ |
|---|---|---|---|
| CoOp | 93.40±0.29 | 95.13±0.43 | +1.73±0.70 |
| CoCoOp | 94.87±0.55 | 95.74±0.18 | +0.87±0.42 |
| IVLP | 94.22±1.03 | 94.18±0.62 | -0.04±1.53 |
| KgCoOp | 94.29±0.04 | 94.04±0.38 | -0.25±0.36 |
| ProGrad | 93.82±0.61 | 94.99±0.22 | +1.17±0.60 |
| MaPLe | 93.76±1.36 | 94.51±1.07 | +0.76±0.40 |
| PromptSRC | 94.64±0.19 | 95.54±0.09 | +0.90±0.13 |
| LoCoOp | 92.60±0.52 | 94.22±0.53 | +1.61±0.44 |

### (c) Caltech101.

|  | MaxLogit | MLS-M | △ |
|---|---|---|---|
| CoOp | 88.88±0.91 | 90.54±1.21 | +1.66±0.85 |
| CoCoOp | 85.61±1.96 | 88.06±2.40 | +2.46±1.43 |
| IVLP | 88.03±1.47 | 92.33±1.21 | +4.31±0.55 |
| KgCoOp | 83.63±0.18 | 89.80±0.18 | +6.17±0.03 |
| ProGrad | 83.99±1.42 | 89.25±1.18 | +5.26±1.51 |
| MaPLe | 87.09±0.76 | 92.06±0.89 | +4.97±1.59 |
| PromptSRC | 85.04±0.34 | 90.68±0.54 | +5.64±0.49 |
| LoCoOp | 77.58±2.53 | 89.72±1.09 | +12.14±2.80 |

### (d) OxfordPets.

|  | MaxLogit | MLS-M | △ |
|---|---|---|---|
| CoOp | 86.69±1.96 | 89.06±1.86 | +2.37±0.13 |
| CoCoOp | 90.96±0.62 | 93.07±0.16 | +2.11±0.61 |
| IVLP | 90.40±0.59 | 92.21±0.56 | +1.81±0.86 |
| KgCoOp | 90.21±0.16 | 92.49±0.33 | +2.28±0.28 |
| ProGrad | 87.01±1.27 | 89.19±1.74 | +2.18±0.47 |
| MaPLe | 89.00±0.59 | 92.25±1.20 | +3.25±0.77 |
| PromptSRC | 91.60±0.19 | 93.92±0.35 | +2.32±0.21 |
| LoCoOp | 85.70±2.63 | 89.51±1.11 | +3.80±2.38 |

### (e) StanfordCars.

|  | MaxLogit | MLS-M | △ |
|---|---|---|---|
| CoOp | 92.30±0.62 | 93.03±1.10 | +0.73±0.56 |
| CoCoOp | 92.64±1.14 | 93.27±1.34 | +0.63±0.56 |
| IVLP | 89.63±1.84 | 93.38±1.22 | +3.75±2.47 |
| KgCoOp | 92.90±0.11 | 93.35±0.14 | +0.46±0.06 |
| ProGrad | 92.93±0.90 | 93.25±0.61 | +0.32±0.41 |
| MaPLe | 92.40±0.68 | 93.10±1.00 | +0.70±0.32 |
| PromptSRC | 92.94±0.24 | 94.43±0.85 | +1.49±0.68 |
| LoCoOp | 88.81±0.48 | 92.42±1.02 | +3.61±1.23 |

### (f) Flowers102.

|  | MaxLogit | MLS-M | △ |
|---|---|---|---|
| CoOp | 92.28±0.46 | 93.00±0.50 | +0.72±0.07 |
| CoCoOp | 88.33±0.08 | 89.32±0.22 | +0.99±0.30 |
| IVLP | 88.07±4.61 | 88.71±4.29 | +0.63±0.32 |
| KgCoOp | 88.36±0.20 | 92.26±0.37 | +3.90±0.37 |
| ProGrad | 90.46±0.19 | 92.43±0.46 | +1.98±0.50 |
| MaPLe | 87.04±0.98 | 88.53±0.79 | +1.49±0.72 |
| PromptSRC | 93.37±0.38 | 94.81±0.27 | +1.44±0.12 |
| LoCoOp | 88.52±2.42 | 90.72±1.72 | +2.20±0.71 |

### (g) Food101

|  | MaxLogit | MLS-M | △ |
|---|---|---|---|
| CoOp | 87.71±1.08 | 88.22±1.11 | +0.51±0.18 |
| CoCoOp | 91.19±0.48 | 91.74±0.62 | +0.55±0.24 |
| IVLP | 89.41±1.46 | 92.28±0.29 | +2.87±1.44 |
| KgCoOp | 89.85±0.10 | 92.00±0.29 | +2.15±0.32 |
| ProGrad | 89.28±0.37 | 91.03±0.45 | +1.75±0.36 |
| MaPLe | 90.50±0.81 | 92.68±0.24 | +2.17±0.94 |
| PromptSRC | 91.36±0.30 | 92.47±0.37 | +1.10±0.36 |
| LoCoOp | 86.12±1.73 | 89.93±1.12 | +3.80±0.75 |

### (h) FGVCAircraft.

|  | MaxLogit | MLS-M | △ |
|---|---|---|---|
| CoOp | 55.53±5.96 | 56.84±5.79 | +1.30±1.43 |
| CoCoOp | 53.41±1.06 | 55.94±2.55 | +2.54±2.25 |
| IVLP | 59.95±5.12 | 65.22±4.95 | +5.26±10.02 |
| KgCoOp | 57.14±1.06 | 57.48±0.42 | +0.33±1.41 |
| ProGrad | 55.17±1.98 | 55.92±6.71 | +0.75±5.07 |
| MaPLe | 48.85±8.17 | 53.45±14.82 | +4.60±9.94 |
| PromptSRC | 63.32±0.71 | 68.18±1.55 | +4.86±1.54 |
| LoCoOp | 56.10±7.22 | 62.22±9.93 | +6.11±3.34 |

### (i) SUN397.

|  | MaxLogit | MLS-M | △ |
|---|---|---|---|
| CoOp | 76.77±1.16 | 77.77±0.76 | +1.00±0.47 |
| CoCoOp | 76.74±0.43 | 79.67±1.02 | +2.93±1.39 |
| IVLP | 78.34±0.12 | 80.45±0.88 | +2.12±0.81 |
| KgCoOp | 76.89±0.17 | 78.52±0.31 | +1.63±0.35 |
| ProGrad | 76.17±0.28 | 79.14±0.62 | +2.97±0.77 |
| MaPLe | 79.09±1.04 | 80.51±0.98 | +1.42±0.17 |
| PromptSRC | 79.25±0.23 | 81.65±0.44 | +2.40±0.31 |
| LoCoOp | 75.00±1.06 | 78.96±0.98 | +3.96±1.14 |

### (j) DTD.

|  | MaxLogit | MLS-M | △ |
|---|---|---|---|
| CoOp | 71.45±0.53 | 72.10±1.04 | +0.64±0.55 |
| CoCoOp | 65.53±0.61 | 68.34±0.42 | +2.81±0.33 |
| IVLP | 67.23±0.52 | 69.55±0.50 | +2.33±0.07 |
| KgCoOp | 63.45±0.50 | 67.91±1.17 | +4.46±0.88 |
| ProGrad | 62.05±1.25 | 67.22±1.00 | +5.17±0.37 |
| MaPLe | 66.90±1.49 | 70.91±1.83 | +4.01±0.52 |
| PromptSRC | 70.78±1.16 | 71.82±1.05 | +1.05±0.10 |
| LoCoOp | 68.50±0.52 | 70.79±0.33 | +2.29±0.60 |

### (k) EuroSAT.

|  | MaxLogit | MLS-M | △ |
|---|---|---|---|
| CoOp | 68.76±3.30 | 69.45±3.03 | +0.69±0.28 |
| CoCoOp | 67.22±4.14 | 67.82±4.22 | +0.60±0.40 |
| IVLP | 62.81±11.25 | 70.59±14.90 | +7.78±3.79 |
| KgCoOp | 62.47±0.59 | 66.72±2.87 | +4.26±3.46 |
| ProGrad | 64.62±2.32 | 67.46±0.83 | +2.84±2.26 |
| MaPLe | 79.49±3.93 | 83.54±6.27 | +4.05±2.44 |
| PromptSRC | 81.33±5.10 | 81.35±5.33 | +0.02±0.25 |
| LoCoOp | 68.44±1.29 | 71.60±5.16 | +3.16±3.89 |

### (l) UCF101.

|  | MaxLogit | MLS-M | △ |
|---|---|---|---|
| CoOp | 83.55±0.94 | 85.42±0.73 | +1.87±0.73 |
| CoCoOp | 82.64±1.02 | 85.35±0.48 | +2.71±0.57 |
| IVLP | 81.35±1.72 | 86.23±1.34 | +4.88±1.19 |
| KgCoOp | 81.75±0.34 | 84.93±0.41 | +3.19±0.50 |
| ProGrad | 80.40±2.13 | 85.18±1.87 | +4.78±0.32 |
| MaPLe | 81.06±0.69 | 84.41±0.65 | +3.35±1.24 |
| PromptSRC | 84.54±0.84 | 86.81±0.95 | +2.27±0.17 |
| LoCoOp | 75.23±2.13 | 83.62±0.92 | +8.39±1.65 |

### (m) CIFAR10.

|  | MaxLogit | MLS-M | △ |
|---|---|---|---|
| CoOp | 88.11±2.48 | 88.12±2.73 | +0.01±0.26 |
| CoCoOp | 92.96±0.58 | 93.38±0.92 | +0.42±0.43 |
| IVLP | 92.27±1.15 | 93.34±0.78 | +1.07±0.43 |
| KgCoOp | 93.33±0.01 | 93.67±0.16 | +0.34±0.14 |
| ProGrad | 92.31±0.54 | 92.71±0.70 | +0.40±0.28 |
| MaPLe | 93.54±0.44 | 93.97±0.47 | +0.43±0.20 |
| PromptSRC | 94.92±0.47 | 94.75±0.45 | -0.16±0.31 |
| LoCoOp | 91.96±1.27 | 88.62±3.55 | -3.34±3.08 |

### (n) CIFAR100.

|  | MaxLogit | MLS-M | △ |
|---|---|---|---|
| CoOp | 75.41±0.95 | 78.30±0.46 | +2.89±0.57 |
| CoCoOp | 77.14±1.36 | 81.60±0.63 | +4.45±1.23 |
| IVLP | 86.85±0.73 | 87.94±0.63 | +1.09±0.45 |
| KgCoOp | 78.39±0.44 | 80.69±0.94 | +2.30±1.23 |
| ProGrad | 76.86±0.64 | 80.76±1.26 | +3.90±0.76 |
| MaPLe | 86.01±0.50 | 89.21±0.23 | +3.19±0.31 |
| PromptSRC | 87.67±0.69 | 89.33±0.83 | +1.66±0.18 |
| LoCoOp | 75.70±0.75 | 76.16±2.61 | +0.46±1.86 |

Table 9: Near OOD AUROC (↑) of prompt learning models over 13 datasets using the MaxLogit score and MLS-M with 4-shots.

(a) Average over 13 datasets.

| | MaxLogit | MLS-M | △ |
|---|---|---|---|
| CoOp | 81.06±11.11 | 82.10±11.35 | +1.04±1.56 |
| CoCoOp | 81.45±12.10 | 83.09±12.43 | +1.63±2.06 |
| IVLP | 81.14±12.63 | 84.08±12.27 | +2.94±2.39 |
| KgCoOp | 80.69±12.01 | 83.03±12.20 | +2.34±2.02 |
| ProGrad | 80.58±11.41 | 82.50±11.50 | +1.92±1.76 |
| MaPLe | 80.76±11.98 | 83.53±11.47 | +2.78±3.03 |
| PromptSRC | 84.27±9.89 | 86.10±9.92 | +1.83±1.54 |
| LoCoOp | 78.52±11.12 | 81.57±11.62 | +3.06±4.44 |

(b) ImageNet.

| | MaxLogit | MLS-M | △ |
|---|---|---|---|
| CoOp | 93.83±0.51 | 95.14±0.22 | +1.31±0.70 |
| CoCoOp | 95.32±0.02 | 95.73±0.05 | +0.41±0.04 |
| IVLP | 95.03±0.13 | 95.45±0.25 | +0.42±0.14 |
| KgCoOp | 94.20±0.08 | 94.29±0.08 | +0.10±0.03 |
| ProGrad | 93.84±0.40 | 94.63±0.31 | +0.79±0.19 |
| MaPLe | 93.94±1.47 | 93.77±0.69 | -0.16±0.86 |
| PromptSRC | 94.57±0.13 | 95.40±0.29 | +0.82±0.24 |
| LoCoOp | 93.24±0.26 | 94.86±0.15 | +1.62±0.35 |

(c) Caltech101.

| | MaxLogit | MLS-M | △ |
|---|---|---|---|
| CoOp | 90.33±1.75 | 91.76±2.61 | +1.42±1.39 |
| CoCoOp | 86.87±1.08 | 89.49±0.16 | +2.62±1.15 |
| IVLP | 84.15±0.90 | 91.65±0.68 | +7.51±1.56 |
| KgCoOp | 83.66±0.10 | 90.14±0.62 | +6.48±0.58 |
| ProGrad | 84.42±1.03 | 88.66±1.52 | +4.24±1.81 |
| MaPLe | 85.04±3.59 | 91.26±1.94 | +6.22±4.19 |
| PromptSRC | 85.28±0.44 | 90.60±0.34 | +5.32±0.27 |
| LoCoOp | 76.00±1.81 | 87.91±1.29 | +11.91±0.60 |

(d) OxfordPets.

| | MaxLogit | MLS-M | △ |
|---|---|---|---|
| CoOp | 87.50±1.91 | 89.59±1.72 | +2.09±0.20 |
| CoCoOp | 89.60±0.93 | 92.90±0.68 | +3.30±1.49 |
| IVLP | 89.53±1.52 | 93.28±0.55 | +3.75±1.13 |
| KgCoOp | 90.07±0.08 | 93.18±0.28 | +3.11±0.32 |
| ProGrad | 87.64±1.19 | 89.51±1.52 | +1.87±0.33 |
| MaPLe | 86.47±2.51 | 91.28±1.59 | +4.81±1.47 |
| PromptSRC | 90.75±1.02 | 93.93±1.02 | +3.18±0.32 |
| LoCoOp | 86.18±1.10 | 89.77±1.27 | +3.59±1.05 |

(e) StanfordCars.

| | MaxLogit | MLS-M | △ |
|---|---|---|---|
| CoOp | 91.17±1.22 | 91.43±1.03 | +0.26±0.36 |
| CoCoOp | 93.30±0.68 | 93.57±0.56 | +0.27±0.23 |
| IVLP | 92.00±1.03 | 92.85±1.03 | +0.85±0.01 |
| KgCoOp | 92.81±0.07 | 93.38±0.17 | +0.57±0.14 |
| ProGrad | 92.38±1.00 | 92.86±0.75 | +0.48±0.54 |
| MaPLe | 91.86±1.10 | 92.96±0.71 | +1.10±1.19 |
| PromptSRC | 92.77±0.19 | 93.82±0.28 | +1.05±0.39 |
| LoCoOp | 87.34±1.74 | 91.59±0.34 | +4.25±2.02 |

(f) Flowers102.

| | MaxLogit | MLS-M | △ |
|---|---|---|---|
| CoOp | 91.86±1.63 | 93.13±1.28 | +1.27±0.97 |
| CoCoOp | 87.75±1.53 | 89.30±0.74 | +1.55±1.04 |
| IVLP | 86.85±1.79 | 88.54±1.03 | +1.69±0.91 |
| KgCoOp | 86.91±0.26 | 90.44±0.01 | +3.53±0.27 |
| ProGrad | 88.87±1.85 | 91.37±1.52 | +2.50±0.87 |
| MaPLe | 85.57±2.40 | 87.96±1.68 | +2.38±1.30 |
| PromptSRC | 91.72±0.23 | 92.92±0.73 | +1.19±0.50 |
| LoCoOp | 87.15±0.57 | 89.05±0.72 | +1.90±0.65 |

(g) Food101

| | MaxLogit | MLS-M | △ |
|---|---|---|---|
| CoOp | 86.74±1.38 | 87.79±1.03 | +1.05±0.40 |
| CoCoOp | 90.61±0.90 | 91.88±0.41 | +1.27±0.50 |
| IVLP | 89.44±0.59 | 92.16±0.73 | +2.72±0.65 |
| KgCoOp | 89.57±0.21 | 92.25±0.17 | +2.68±0.27 |
| ProGrad | 87.78±0.58 | 90.72±0.16 | +2.94±0.50 |
| MaPLe | 88.95±1.02 | 92.03±0.69 | +3.08±1.55 |
| PromptSRC | 90.74±0.15 | 92.06±0.35 | +1.32±0.21 |
| LoCoOp | 84.36±2.01 | 90.44±1.29 | +6.08±1.32 |

(h) FGVCAircraft.

| | MaxLogit | MLS-M | △ |
|---|---|---|---|
| CoOp | 56.85±1.73 | 57.84±2.90 | +1.00±2.88 |
| CoCoOp | 54.94±2.06 | 53.40±1.26 | -1.54±2.98 |
| IVLP | 56.56±7.89 | 60.44±11.15 | +3.88±3.74 |
| KgCoOp | 57.62±1.51 | 56.08±1.55 | -1.54±0.51 |
| ProGrad | 54.14±2.94 | 54.24±3.66 | +0.10±0.96 |
| MaPLe | 51.02±0.95 | 57.51±4.12 | +6.49±5.03 |
| PromptSRC | 61.21±2.47 | 62.69±3.58 | +1.48±2.30 |
| LoCoOp | 52.51±3.97 | 54.96±8.63 | +2.45±9.18 |

(i) SUN397.

| | MaxLogit | MLS-M | △ |
|---|---|---|---|
| CoOp | 76.73±0.94 | 77.27±0.56 | +0.54±0.38 |
| CoCoOp | 76.24±0.84 | 78.52±0.33 | +2.28±0.99 |
| IVLP | 77.36±1.27 | 79.48±0.67 | +2.11±0.90 |
| KgCoOp | 76.60±0.44 | 77.76±0.54 | +1.16±0.10 |
| ProGrad | 75.84±1.03 | 77.43±1.27 | +1.58±0.86 |
| MaPLe | 78.33±0.74 | 79.93±0.40 | +1.61±0.95 |
| PromptSRC | 78.59±0.36 | 80.44±0.29 | +1.86±0.61 |
| LoCoOp | 73.59±1.03 | 78.12±0.62 | +4.53±0.64 |

(j) DTD.

| | MaxLogit | MLS-M | △ |
|---|---|---|---|
| CoOp | 68.35±0.34 | 69.74±0.64 | +1.39±0.70 |
| CoCoOp | 64.22±1.22 | 66.23±0.16 | +2.01±1.06 |
| IVLP | 65.12±0.45 | 67.49±0.99 | +2.37±0.68 |
| KgCoOp | 63.09±0.23 | 66.45±0.68 | +3.36±0.46 |
| ProGrad | 63.40±1.51 | 67.36±1.72 | +3.96±0.77 |
| MaPLe | 64.63±0.99 | 66.06±0.55 | +1.43±1.03 |
| PromptSRC | 68.39±1.39 | 70.29±1.02 | +1.90±0.69 |
| LoCoOp | 67.92±1.34 | 69.50±0.85 | +1.58±0.58 |

(k) EuroSAT.

| | MaxLogit | MLS-M | △ |
|---|---|---|---|
| CoOp | 67.51±2.56 | 66.19±2.93 | -1.32±1.80 |
| CoCoOp | 68.85±2.31 | 69.49±2.06 | +0.64±0.96 |
| IVLP | 60.33±7.70 | 65.49±9.60 | +5.16±2.58 |
| KgCoOp | 62.92±0.27 | 66.84±0.41 | +3.92±0.20 |
| ProGrad | 74.47±1.09 | 74.59±2.97 | +0.12±1.89 |
| MaPLe | 69.89±4.69 | 69.99±4.37 | +0.10±0.58 |
| PromptSRC | 77.62±0.65 | 78.29±1.08 | +0.66±0.45 |
| LoCoOp | 70.17±5.93 | 69.65±5.67 | -0.52±0.63 |

(l) UCF101.

| | MaxLogit | MLS-M | △ |
|---|---|---|---|
| CoOp | 81.09±1.53 | 82.52±1.91 | +1.43±0.95 |
| CoCoOp | 81.34±1.17 | 84.59±1.22 | +3.25±1.91 |
| IVLP | 80.78±1.59 | 84.87±1.46 | +4.10±0.84 |
| KgCoOp | 80.78±0.22 | 83.99±0.30 | +3.22±0.27 |
| ProGrad | 81.07±1.02 | 83.00±0.40 | +1.93±1.29 |
| MaPLe | 81.35±0.32 | 86.21±1.05 | +4.85±1.21 |
| PromptSRC | 84.07±0.51 | 86.77±0.99 | +2.70±1.15 |
| LoCoOp | 78.33±0.78 | 83.19±1.89 | +4.86±1.34 |

(m) CIFAR10.

| | MaxLogit | MLS-M | △ |
|---|---|---|---|
| CoOp | 87.12±2.56 | 87.29±2.73 | +0.17±0.24 |
| CoCoOp | 92.67±0.76 | 93.13±0.69 | +0.45±0.17 |
| IVLP | 91.92±1.84 | 92.81±1.31 | +0.89±0.54 |
| KgCoOp | 92.78±0.29 | 93.39±0.39 | +0.61±0.11 |
| ProGrad | 89.36±0.43 | 90.10±1.12 | +0.75±0.76 |
| MaPLe | 87.53±6.17 | 88.19±4.69 | +0.66±1.82 |
| PromptSRC | 92.74±1.69 | 92.86±1.83 | +0.13±0.16 |
| LoCoOp | 90.93±2.17 | 87.70±2.18 | -3.23±1.33 |

(n) CIFAR100.

| | MaxLogit | MLS-M | △ |
|---|---|---|---|
| CoOp | 74.64±1.74 | 77.60±2.69 | +2.96±1.48 |
| CoCoOp | 77.19±1.06 | 81.91±1.30 | +4.72±1.41 |
| IVLP | 85.72±0.61 | 88.55±0.76 | +2.83±0.29 |
| KgCoOp | 77.99±0.45 | 81.21±0.38 | +3.22±0.28 |
| ProGrad | 74.28±0.12 | 77.97±1.46 | +3.69±1.52 |
| MaPLe | 85.28±0.74 | 88.79±0.58 | +3.51±0.33 |
| PromptSRC | 87.04±0.65 | 89.17±1.09 | +2.13±0.94 |
| LoCoOp | 73.01±1.23 | 73.71±2.24 | +0.71±1.11 |

Table 10: Near OOD AUROC (↑) of prompt learning models over 13 datasets using the MaxLogit score and MLS-M with 2-shots.

**(a) Average over 13 datasets.**

|  | MaxLogit | MLS-M | △ |
|---|---|---|---|
| CoOp | 80.05±11.11 | 81.19±11.05 | +1.14±1.59 |
| CoCoOp | 80.76±12.16 | 82.12±12.60 | +1.35±1.70 |
| IVLP | 80.12±11.76 | 82.87±10.97 | +2.75±2.35 |
| KgCoOp | 80.83±11.54 | 83.13±11.49 | +2.30±2.01 |
| ProGrad | 79.43±11.87 | 82.07±11.48 | +2.64±2.91 |
| MaPLe | 80.49±11.63 | 83.42±11.27 | +2.92±2.35 |
| PromptSRC | 83.18±10.43 | 84.99±10.53 | +1.82±1.58 |
| LoCoOp | 76.28±12.95 | 80.62±11.92 | +4.34±4.81 |

**(b) ImageNet.**

|  | MaxLogit | MLS-M | △ |
|---|---|---|---|
| CoOp | 94.19±0.39 | 94.39±0.14 | +0.20±0.25 |
| CoCoOp | 94.31±0.34 | 94.52±0.27 | +0.21±0.08 |
| IVLP | 94.86±0.67 | 94.65±0.24 | -0.21±0.50 |
| KgCoOp | 94.15±0.17 | 94.32±0.35 | +0.17±0.18 |
| ProGrad | 93.22±0.29 | 94.76±0.25 | +1.54±0.26 |
| MaPLe | 94.50±0.71 | 94.45±0.32 | -0.05±0.55 |
| PromptSRC | 94.48±0.20 | 94.95±0.19 | +0.48±0.20 |
| LoCoOp | 92.89±0.56 | 94.31±0.07 | +1.42±0.54 |

**(c) Caltech101.**

|  | MaxLogit | MLS-M | △ |
|---|---|---|---|
| CoOp | 88.02±1.73 | 89.90±1.19 | +1.88±1.20 |
| CoCoOp | 85.33±0.73 | 88.48±0.99 | +3.15±0.28 |
| IVLP | 84.50±1.75 | 88.55±2.19 | +4.04±2.52 |
| KgCoOp | 83.72±0.41 | 89.78±0.36 | +6.06±0.06 |
| ProGrad | 83.67±2.53 | 90.32±2.56 | +6.64±2.43 |
| MaPLe | 85.89±1.69 | 91.87±0.61 | +5.98±1.71 |
| PromptSRC | 85.33±0.27 | 90.50±1.32 | +5.17±1.50 |
| LoCoOp | 72.52±4.07 | 86.38±1.65 | +13.85±3.56 |

**(d) OxfordPets.**

|  | MaxLogit | MLS-M | △ |
|---|---|---|---|
| CoOp | 86.15±3.09 | 89.19±3.57 | +3.05±0.49 |
| CoCoOp | 87.40±1.25 | 91.23±0.94 | +3.83±0.41 |
| IVLP | 84.75±1.68 | 89.58±1.17 | +4.83±1.67 |
| KgCoOp | 89.65±0.42 | 92.68±0.53 | +3.04±0.22 |
| ProGrad | 88.70±0.71 | 89.82±0.46 | +1.12±1.11 |
| MaPLe | 86.04±1.41 | 89.19±2.52 | +3.15±2.74 |
| PromptSRC | 89.91±0.38 | 92.48±0.71 | +2.57±0.42 |
| LoCoOp | 82.02±2.93 | 89.14±1.05 | +7.12±2.67 |

**(e) StanfordCars.**

|  | MaxLogit | MLS-M | △ |
|---|---|---|---|
| CoOp | 90.78±0.94 | 91.06±1.11 | +0.28±0.71 |
| CoCoOp | 92.91±0.91 | 92.77±1.00 | -0.14±0.14 |
| IVLP | 90.53±1.63 | 92.85±0.46 | +2.32±1.23 |
| KgCoOp | 92.55±0.05 | 93.08±0.39 | +0.53±0.44 |
| ProGrad | 89.29±1.35 | 91.70±1.04 | +2.41±0.61 |
| MaPLe | 90.47±1.41 | 93.26±0.38 | +2.78±1.69 |
| PromptSRC | 92.49±0.37 | 93.75±0.45 | +1.26±0.59 |
| LoCoOp | 87.43±2.72 | 91.69±1.86 | +4.26±0.96 |

**(f) Flowers102.**

|  | MaxLogit | MLS-M | △ |
|---|---|---|---|
| CoOp | 90.03±0.97 | 91.44±0.84 | +1.42±0.29 |
| CoCoOp | 87.50±0.44 | 89.01±0.96 | +1.51±1.01 |
| IVLP | 82.42±0.56 | 85.94±0.68 | +3.52±0.21 |
| KgCoOp | 86.51±0.58 | 90.60±0.44 | +4.08±0.17 |
| ProGrad | 84.55±0.54 | 87.98±0.22 | +3.43±0.46 |
| MaPLe | 89.06±0.30 | 90.76±0.41 | +1.70±0.24 |
| PromptSRC |  |  |  |
| LoCoOp | 86.04±1.21 | 87.79±0.65 | +1.75±0.64 |

**(g) Food101**

|  | MaxLogit | MLS-M | △ |
|---|---|---|---|
| CoOp | 84.32±0.55 | 85.60±1.02 | +1.28±0.63 |
| CoCoOp | 90.19±1.03 | 91.17±1.08 | +0.98±0.34 |
| IVLP | 90.08±0.26 | 91.09±0.45 | +1.01±0.66 |
| KgCoOp | 89.77±0.31 | 91.94±0.33 | +2.17±0.11 |
| ProGrad | 87.31±2.18 | 90.08±0.53 | +2.77±1.69 |
| MaPLe | 86.87±0.33 | 91.34±0.36 | +4.48±0.64 |
| PromptSRC | 90.62±0.54 | 91.64±0.59 | +1.02±0.22 |
| LoCoOp | 84.15±0.34 | 89.87±0.39 | +5.71±0.72 |

**(h) FGVCAircraft.**

|  | MaxLogit | MLS-M | △ |
|---|---|---|---|
| CoOp | 55.74±1.10 | 57.73±2.73 | +1.99±3.42 |
| CoCoOp | 52.84±1.55 | 51.58±2.72 | -1.26±2.25 |
| IVLP | 56.78±3.46 | 61.06±2.52 | +4.28±1.69 |
| KgCoOp | 60.16±2.21 | 59.39±2.44 | -0.77±0.44 |
| ProGrad | 52.61±1.01 | 56.25±5.57 | +3.64±6.10 |
| MaPLe | 55.94±1.28 | 60.17±3.89 | +4.22±3.82 |
| PromptSRC | 60.38±3.63 | 62.18±4.82 | +1.80±2.49 |
| LoCoOp | 46.36±1.48 | 55.36±6.99 | +9.00±5.61 |

**(i) SUN397.**

|  | MaxLogit | MLS-M | △ |
|---|---|---|---|
| CoOp | 74.08±0.32 | 75.16±0.62 | +1.08±0.55 |
| CoCoOp | 75.78±0.47 | 77.19±0.08 | +1.41±0.46 |
| IVLP | 76.86±0.69 | 78.73±0.81 | +1.87±0.71 |
| KgCoOp | 76.06±0.10 | 77.26±0.53 | +1.20±0.61 |
| ProGrad | 75.05±0.82 | 76.26±0.48 | +1.22±0.41 |
| MaPLe | 76.87±0.15 | 78.95±0.71 | +2.08±0.56 |
| PromptSRC | 77.91±0.20 | 80.19±0.31 | +2.28±0.27 |
| LoCoOp | 72.91±0.71 | 77.06±0.87 | +4.15±0.18 |

**(j) DTD.**

|  | MaxLogit | MLS-M | △ |
|---|---|---|---|
| CoOp | 66.73±1.35 | 67.52±1.39 | +0.79±0.53 |
| CoCoOp | 65.31±0.34 | 66.43±0.86 | +1.12±0.79 |
| IVLP | 63.97±0.99 | 66.74±0.74 | +2.77±1.15 |
| KgCoOp | 63.98±0.51 | 67.32±1.14 | +3.34±1.20 |
| ProGrad | 62.75±0.94 | 66.94±0.43 | +4.19±0.67 |
| MaPLe | 63.76±2.01 | 66.17±1.20 | +2.41±1.25 |
| PromptSRC | 67.63±0.24 | 68.20±0.30 | +0.57±0.20 |
| LoCoOp | 63.60±0.41 | 65.62±1.23 | +2.02±0.82 |

**(k) EuroSAT.**

|  | MaxLogit | MLS-M | △ |
|---|---|---|---|
| CoOp | 69.45±3.44 | 68.88±3.68 | -0.58±0.38 |
| CoCoOp | 68.72±4.97 | 69.27±5.04 | +0.55±0.89 |
| IVLP | 62.65±1.90 | 66.29±3.59 | +3.63±4.77 |
| KgCoOp | 63.15±0.24 | 67.37±1.54 | +4.22±1.70 |
| ProGrad | 68.21±5.43 | 69.67±3.82 | +1.46±2.18 |
| MaPLe | 65.05±5.56 | 68.44±6.25 | +3.38±1.21 |
| PromptSRC | 71.63±0.73 | 72.92±0.91 | +1.29±1.61 |
| LoCoOp | 63.36±2.81 | 66.60±4.99 | +3.24±2.39 |

**(l) UCF101.**

|  | MaxLogit | MLS-M | △ |
|---|---|---|---|
| CoOp | 82.56±1.23 | 83.75±0.47 | +1.19±0.77 |
| CoCoOp | 79.81±0.89 | 82.60±0.71 | +2.80±0.38 |
| IVLP | 78.07±0.27 | 82.26±0.57 | +4.18±0.40 |
| KgCoOp | 80.67±0.15 | 82.96±1.30 | +2.30±1.19 |
| ProGrad | 81.13±0.67 | 84.09±0.61 | +2.96±0.21 |
| MaPLe | 80.17±0.71 | 83.26±1.24 | +3.09±0.54 |
| PromptSRC | 82.14±0.69 | 84.13±0.19 | +1.99±0.65 |
| LoCoOp | 74.83±3.00 | 82.08±2.14 | +7.25±0.87 |

**(m) CIFAR10.**

|  | MaxLogit | MLS-M | △ |
|---|---|---|---|
| CoOp | 86.90±0.17 | 86.61±0.59 | -0.29±0.45 |
| CoCoOp | 93.11±0.38 | 93.36±0.41 | +0.24±0.08 |
| IVLP | 92.90±0.41 | 92.94±0.13 | +0.04±0.53 |
| KgCoOp | 93.17±0.07 | 93.54±0.06 | +0.37±0.05 |
| ProGrad | 88.24±0.35 | 87.03±1.63 | -1.21±1.41 |
| MaPLe | 93.02±0.44 | 92.72±0.42 | -0.30±0.83 |
| PromptSRC | 93.88±0.11 | 94.33±0.28 | +0.45±0.19 |
| LoCoOp | 92.70±0.80 | 88.96±1.85 | -3.73±2.64 |

**(n) CIFAR100.**

|  | MaxLogit | MLS-M | △ |
|---|---|---|---|
| CoOp | 71.71±1.18 | 74.21±0.75 | +2.50±1.84 |
| CoCoOp | 76.71±0.95 | 79.91±2.15 | +3.20±1.41 |
| IVLP | 83.15±1.72 | 86.63±1.02 | +3.48±0.71 |
| KgCoOp | 77.28±0.43 | 80.48±0.31 | +3.20±0.50 |
| ProGrad | 74.20±1.65 | 78.77±1.67 | +4.56±2.70 |
| MaPLe | 83.27±1.28 | 86.61±0.90 | +3.34±2.09 |
| PromptSRC | 85.83±0.39 | 88.86±0.45 | +3.03±0.19 |
| LoCoOp | 72.81±1.34 | 73.16±1.43 | +0.36±0.82 |

Table 11: Near OOD AUROC (↑) of prompt learning models over 13 datasets using the MaxLogit score and MLS-M with 1-shot.

### (a) Average over 13 datasets.

|  | MaxLogit | MLS-M | △ |
|---|---|---|---|
| CoOp | 77.78±11.44 | 78.69±11.97 | +0.92±2.15 |
| CoCoOp | 79.70±13.00 | 81.26±12.66 | +1.56±2.54 |
| IVLP | 80.00±10.91 | 82.27±11.08 | +2.26±3.19 |
| KgCoOp | 80.19±12.35 | 81.98±12.90 | +1.79±2.74 |
| ProGrad | 78.26±12.72 | 80.32±12.20 | +2.06±2.87 |
| MaPLe | 78.86±13.76 | 81.13±13.97 | +2.27±4.69 |
| PromptSRC | 81.56±12.17 | 82.81±13.06 | +1.25±2.45 |
| LoCoOp | 76.37±12.38 | 79.56±11.73 | +3.19±5.99 |

### (b) ImageNet.

|  | MaxLogit | MLS-M | △ |
|---|---|---|---|
| CoOp | 93.65±0.21 | 93.26±0.76 | -0.39±0.96 |
| CoCoOp | 94.85±0.39 | 94.57±0.57 | -0.28±0.92 |
| IVLP | 94.18±0.49 | 94.64±0.57 | +0.45±0.89 |
| KgCoOp | 94.07±0.12 | 94.24±0.29 | +0.16±0.23 |
| ProGrad | 93.09±0.63 | 93.99±0.40 | +0.90±0.24 |
| MaPLe | 94.12±0.62 | 94.32±0.40 | +0.19±0.44 |
| PromptSRC | 94.23±0.49 | 95.10±0.29 | +0.87±0.22 |
| LoCoOp | 93.09±0.52 | 94.46±0.28 | +1.37±0.47 |

### (c) Caltech101.

|  | MaxLogit | MLS-M | △ |
|---|---|---|---|
| CoOp | 84.37±2.48 | 87.03±1.50 | +2.66±1.47 |
| CoCoOp | 84.06±0.95 | 89.11±1.24 | +5.05±1.93 |
| IVLP | 83.13±2.65 | 86.92±2.13 | +3.79±2.12 |
| KgCoOp | 83.14±0.02 | 90.23±0.24 | +7.09±0.25 |
| ProGrad | 80.48±0.51 | 86.38±3.52 | +5.90±3.09 |
| MaPLe | 83.66±2.91 | 89.39±1.00 | +5.74±3.28 |
| PromptSRC | 84.36±0.32 | 90.45±0.51 | +6.09±0.81 |
| LoCoOp | 75.46±2.62 | 86.03±2.08 | +10.57±0.86 |

### (d) OxfordPets.

|  | MaxLogit | MLS-M | △ |
|---|---|---|---|
| CoOp | 85.54±1.64 | 89.17±2.00 | +3.63±1.51 |
| CoCoOp | 89.03±1.15 | 90.64±0.09 | +1.61±1.10 |
| IVLP | 88.39±1.71 | 90.89±0.15 | +2.50±1.58 |
| KgCoOp | 89.51±0.24 | 92.67±1.14 | +3.16±0.93 |
| ProGrad | 88.35±2.29 | 90.89±1.62 | +2.54±0.88 |
| MaPLe | 85.46±4.88 | 88.10±2.63 | +2.64±2.25 |
| PromptSRC | 90.11±0.68 | 92.23±0.50 | +2.12±1.16 |
| LoCoOp | 84.09±2.60 | 88.34±1.97 | +4.25±3.66 |

### (e) StanfordCars.

|  | MaxLogit | MLS-M | △ |
|---|---|---|---|
| CoOp | 89.04±1.69 | 88.99±1.70 | -0.05±0.54 |
| CoCoOp | 91.98±0.63 | 92.54±0.46 | +0.56±0.75 |
| IVLP | 89.48±0.68 | 92.62±1.57 | +3.14±0.89 |
| KgCoOp | 92.62±0.12 | 93.08±0.21 | +0.46±0.13 |
| ProGrad | 89.73±2.12 | 91.28±0.91 | +1.54±1.24 |
| MaPLe | 90.75±0.79 | 91.73±0.13 | +0.98±0.69 |
| PromptSRC | 92.38±0.60 | 93.83±0.44 | +1.45±0.27 |
| LoCoOp | 88.53±2.24 | 91.90±0.25 | +3.37±2.39 |

### (f) Flowers102.

|  | MaxLogit | MLS-M | △ |
|---|---|---|---|
| CoOp | 85.67±1.94 | 87.73±2.37 | +2.06±0.69 |
| CoCoOp | 86.67±0.55 | 88.32±0.52 | +1.65±0.14 |
| IVLP | 83.04±2.91 | 86.47±1.49 | +3.42±1.61 |
| KgCoOp | 86.21±0.08 | 89.22±0.80 | +3.01±0.84 |
| ProGrad | 83.11±0.73 | 89.02±0.97 | +0.91±0.44 |
| MaPLe | 83.34±1.97 | 85.81±2.41 | +2.46±0.47 |
| PromptSRC | 86.95±0.70 | 88.89±1.25 | +1.94±0.55 |
| LoCoOp | 82.06±1.53 | 85.38±2.01 | +3.32±0.84 |

### (g) Food101

|  | MaxLogit | MLS-M | △ |
|---|---|---|---|
| CoOp | 86.05±1.60 | 87.82±2.12 | +1.77±0.54 |
| CoCoOp | 90.07±0.75 | 90.91±0.37 | +0.84±0.73 |
| IVLP | 89.15±1.82 | 91.01±0.78 | +1.86±1.17 |
| KgCoOp | 89.88±0.14 | 91.58±0.47 | +1.70±0.50 |
| ProGrad | 88.27±1.64 | 90.75±1.42 | +2.48±1.10 |
| MaPLe | 88.40±0.68 | 90.83±0.75 | +2.43±0.59 |
| PromptSRC | 90.59±0.12 | 91.76±0.16 | +1.17±0.08 |
| LoCoOp | 84.22±3.05 | 88.87±1.10 | +4.66±2.67 |

### (h) FGVCAircraft.

|  | MaxLogit | MLS-M | △ |
|---|---|---|---|
| CoOp | 56.46±1.55 | 56.13±4.61 | -0.33±3.69 |
| CoCoOp | 51.86±2.42 | 57.14±5.02 | +5.29±3.96 |
| IVLP | 59.36±1.26 | 62.40±4.31 | +3.04±3.42 |
| KgCoOp | 57.11±1.22 | 57.23±3.26 | +0.12±2.04 |
| ProGrad | 51.97±1.64 | 57.80±3.36 | +5.83±3.69 |
| MaPLe | 48.73±7.71 | 51.97±5.93 | +3.24±13.20 |
| PromptSRC | 53.69±2.55 | 52.67±1.32 | -1.02±3.51 |
| LoCoOp | 47.61±4.93 | 58.87±6.44 | +11.26±9.05 |

### (i) SUN397.

|  | MaxLogit | MLS-M | △ |
|---|---|---|---|
| CoOp | 73.03±1.27 | 73.99±2.01 | +0.95±0.77 |
| CoCoOp | 75.58±0.29 | 76.79±0.55 | +1.21±0.26 |
| IVLP | 75.31±0.70 | 78.63±0.77 | +3.32±1.46 |
| KgCoOp | 75.35±0.08 | 77.19±0.14 | +1.84±0.22 |
| ProGrad | 73.48±0.56 | 76.04±0.67 | +2.57±0.20 |
| MaPLe | 75.59±1.31 | 78.56±0.92 | +2.97±0.44 |
| PromptSRC | 77.19±0.99 | 79.17±0.71 | +1.98±0.32 |
| LoCoOp | 72.56±1.04 | 75.99±0.86 | +3.43±0.26 |

### (j) DTD.

|  | MaxLogit | MLS-M | △ |
|---|---|---|---|
| CoOp | 63.93±1.09 | 64.32±0.70 | +0.40±0.79 |
| CoCoOp | 64.23±0.39 | 65.22±1.19 | +0.99±0.87 |
| IVLP | 61.56±1.52 | 64.71±1.91 | +3.16±0.85 |
| KgCoOp | 62.95±1.40 | 67.80±0.89 | +4.85±0.51 |
| ProGrad | 62.44±1.61 | 64.92±1.94 | +2.48±0.50 |
| MaPLe | 61.56±0.70 | 64.89±0.47 | +3.33±0.97 |
| PromptSRC | 66.69±1.81 | 67.99±2.73 | +1.30±0.98 |
| LoCoOp | 65.00±1.05 | 66.74±0.67 | +1.74±0.62 |

### (k) EuroSAT.

|  | MaxLogit | MLS-M | △ |
|---|---|---|---|
| CoOp | 62.12±4.42 | 61.19±4.11 | -0.93±0.61 |
| CoCoOp | 60.92±3.46 | 58.94±3.27 | -1.98±1.72 |
| IVLP | 67.81±4.12 | 64.61±5.13 | -3.21±5.92 |
| KgCoOp | 60.86±1.13 | 57.60±2.80 | -3.26±1.67 |
| ProGrad | 61.51±1.14 | 60.65±3.08 | -0.86±4.18 |
| MaPLe | 60.69±7.74 | 57.24±10.24 | -3.45±4.26 |
| PromptSRC | 65.51±4.26 | 62.56±2.38 | -2.95±1.87 |
| LoCoOp | 63.02±4.28 | 61.75±6.38 | -1.27±2.61 |

### (l) UCF101.

|  | MaxLogit | MLS-M | △ |
|---|---|---|---|
| CoOp | 79.98±1.30 | 81.17±0.71 | +1.19±0.69 |
| CoCoOp | 80.07±1.01 | 82.40±0.35 | +2.32±1.18 |
| IVLP | 78.63±1.72 | 82.75±0.78 | +4.12±2.11 |
| KgCoOp | 81.02±0.88 | 83.41±1.66 | +2.39±0.84 |
| ProGrad | 81.39±1.95 | 82.26±1.92 | +0.87±0.13 |
| MaPLe | 78.44±0.58 | 82.58±0.49 | +4.14±0.29 |
| PromptSRC | 81.02±0.85 | 83.40±0.99 | +2.38±0.14 |
| LoCoOp | 74.53±1.82 | 79.48±2.15 | +4.95±0.33 |

### (m) CIFAR10.

|  | MaxLogit | MLS-M | △ |
|---|---|---|---|
| CoOp | 83.06±1.34 | 81.77±0.90 | -1.29±1.60 |
| CoCoOp | 90.85±2.48 | 90.01±2.16 | -0.84±0.59 |
| IVLP | 87.01±6.32 | 86.81±3.85 | -0.20±3.51 |
| KgCoOp | 93.01±0.38 | 91.78±1.72 | -1.23±1.61 |
| ProGrad | 87.62±4.43 | 86.07±4.47 | -1.55±2.03 |
| MaPLe | 92.41±1.38 | 92.43±2.35 | +0.02±1.21 |
| PromptSRC | 93.62±0.39 | 92.59±2.13 | -1.02±1.78 |
| LoCoOp | 89.29±1.17 | 82.42±10.55 | -6.87±9.46 |

### (n) CIFAR100.

|  | MaxLogit | MLS-M | △ |
|---|---|---|---|
| CoOp | 68.18±1.23 | 70.43±3.00 | +2.24±2.90 |
| CoCoOp | 76.01±1.44 | 79.86±1.91 | +3.85±0.47 |
| IVLP | 83.00±0.54 | 87.04±0.30 | +4.04±0.72 |
| KgCoOp | 76.77±0.66 | 79.69±1.55 | +2.92±0.91 |
| ProGrad | 70.90±2.52 | 74.08±1.77 | +3.18±1.24 |
| MaPLe | 82.03±2.21 | 86.88±0.88 | +4.85±2.34 |
| PromptSRC | 83.99±0.11 | 85.91±0.89 | +1.92±0.82 |
| LoCoOp | 73.40±1.57 | 74.07±1.22 | +0.67±0.54 |

A.3.2  ENERGY SCORE

We provide the same results of Table 5 to Table 11 using Energy score in Table 12 to Table 18.

Table 12: Near OOD AUROC (↑) of prompt learning models over 13 datasets using the Energy score and MLS-E.

(a) Average over 13 datasets.

|  | Energy | MLS-E | △ |
|---|---|---|---|
| CoOp | 80.44 | 81.71 | +1.27 |
| CoCoOp | 80.53 | 82.74 | +2.21 |
| IVLP | 80.49 | 84.40 | +3.91 |
| KgCoOp | 80.14 | 83.23 | +3.09 |
| ProGrad | 78.79 | 81.93 | +3.14 |
| MaPLe | 80.39 | 83.99 | +3.60 |
| PromptSRC | 83.48 | 85.88 | +2.40 |
| LoCoOp | 75.94 | 81.25 | +5.31 |

(b) ImageNet.

|  | Energy | MLS-E | △ |
|---|---|---|---|
| CoOp | 93.73 | 94.83 | +1.10 |
| CoCoOp | 94.76 | 95.30 | +0.54 |
| IVLP | 94.50 | 94.94 | +0.44 |
| KgCoOp | 94.05 | 94.41 | +0.36 |
| ProGrad | 93.46 | 94.79 | +1.33 |
| MaPLe | 94.07 | 94.58 | +0.51 |
| PromptSRC | 94.46 | 95.56 | +1.10 |
| LoCoOp | 92.25 | 94.53 | +2.28 |

(c) Caltech101.

|  | Energy | MLS-E | △ |
|---|---|---|---|
| CoOp | 87.31 | 89.48 | +2.17 |
| CoCoOp | 84.19 | 88.04 | +3.85 |
| IVLP | 84.08 | 89.85 | +5.77 |
| KgCoOp | 80.39 | 88.79 | +8.39 |
| ProGrad | 80.60 | 87.60 | +7.00 |
| MaPLe | 84.52 | 90.95 | +6.43 |
| PromptSRC | 82.59 | 89.70 | +7.11 |
| LoCoOp | 71.45 | 86.09 | +14.65 |

(d) OxfordPets.

|  | Energy | MLS-E | △ |
|---|---|---|---|
| CoOp | 85.91 | 88.62 | +2.71 |
| CoCoOp | 88.93 | 92.01 | +3.08 |
| IVLP | 88.27 | 91.67 | +3.40 |
| KgCoOp | 89.25 | 92.28 | +3.03 |
| ProGrad | 87.54 | 89.51 | +1.97 |
| MaPLe | 86.66 | 90.64 | +3.98 |
| PromptSRC | 90.38 | 93.29 | +2.90 |
| LoCoOp | 82.90 | 88.44 | +5.54 |

(e) StanfordCars.

|  | Energy | MLS-E | △ |
|---|---|---|---|
| CoOp | 91.37 | 91.66 | +0.29 |
| CoCoOp | 92.56 | 93.17 | +0.62 |
| IVLP | 90.45 | 93.24 | +2.79 |
| KgCoOp | 92.85 | 93.43 | +0.59 |
| ProGrad | 91.34 | 92.55 | +1.21 |
| MaPLe | 91.53 | 93.14 | +1.62 |
| PromptSRC | 92.97 | 94.47 | +1.51 |
| LoCoOp | 87.93 | 92.16 | +4.23 |

(f) Flowers102.

|  | Energy | MLS-E | △ |
|---|---|---|---|
| CoOp | 90.12 | 91.51 | +1.40 |
| CoCoOp | 86.76 | 88.89 | +2.13 |
| IVLP | 84.51 | 87.44 | +2.93 |
| KgCoOp | 85.78 | 90.51 | +4.74 |
| ProGrad | 87.51 | 90.37 | +2.86 |
| MaPLe | 84.45 | 87.40 | +2.95 |
| PromptSRC | 89.92 | 91.86 | +1.93 |
| LoCoOp | 83.73 | 87.14 | +3.41 |

(g) Food101.

|  | Energy | MLS-E | △ |
|---|---|---|---|
| CoOp | 86.32 | 87.70 | +1.38 |
| CoCoOp | 90.09 | 91.44 | +1.35 |
| IVLP | 89.22 | 91.73 | +2.51 |
| KgCoOp | 89.29 | 91.96 | +2.67 |
| ProGrad | 87.93 | 90.90 | +2.98 |
| MaPLe | 88.43 | 91.84 | +3.41 |
| PromptSRC | 90.59 | 91.97 | +1.38 |
| LoCoOp | 82.51 | 89.45 | +6.94 |

(h) FGVCAircraft.

|  | Energy | MLS-E | △ |
|---|---|---|---|
| CoOp | 58.92 | 59.98 | +1.06 |
| CoCoOp | 57.28 | 61.61 | +4.33 |
| IVLP | 63.66 | 71.01 | +7.35 |
| KgCoOp | 66.07 | 66.68 | +0.61 |
| ProGrad | 57.15 | 59.58 | +2.43 |
| MaPLe | 57.03 | 63.37 | +6.33 |
| PromptSRC | 68.80 | 70.34 | +1.54 |
| LoCoOp | 55.31 | 61.06 | +5.75 |

(i) SUN397.

|  | Energy | MLS-E | △ |
|---|---|---|---|
| CoOp | 75.10 | 76.19 | +1.09 |
| CoCoOp | 75.10 | 77.36 | +2.27 |
| IVLP | 76.17 | 79.03 | +2.86 |
| KgCoOp | 74.80 | 76.54 | +1.74 |
| ProGrad | 74.00 | 76.48 | +2.48 |
| MaPLe | 76.65 | 79.09 | +2.44 |
| PromptSRC | 77.26 | 79.91 | +2.65 |
| LoCoOp | 71.27 | 76.23 | +4.96 |

(j) DTD.

|  | Energy | MLS-E | △ |
|---|---|---|---|
| CoOp | 68.35 | 69.16 | +0.81 |
| CoCoOp | 63.75 | 66.27 | +2.52 |
| IVLP | 63.94 | 67.25 | +3.31 |
| KgCoOp | 61.31 | 66.69 | +5.38 |
| ProGrad | 60.84 | 65.22 | +4.38 |
| MaPLe | 63.54 | 66.92 | +3.38 |
| PromptSRC | 67.56 | 69.27 | +1.71 |
| LoCoOp | 65.29 | 68.07 | +2.78 |

(k) EuroSAT.

|  | Energy | MLS-E | △ |
|---|---|---|---|
| CoOp | 67.54 | 67.49 | -0.06 |
| CoCoOp | 66.12 | 65.99 | -0.13 |
| IVLP | 63.89 | 69.72 | +5.83 |
| KgCoOp | 62.36 | 66.24 | +3.88 |
| ProGrad | 68.19 | 69.10 | +0.90 |
| MaPLe | 70.02 | 71.94 | +1.93 |
| PromptSRC | 74.40 | 74.30 | -0.10 |
| LoCoOp | 66.04 | 67.80 | +1.77 |

(l) UCF101.

|  | Energy | MLS-E | △ |
|---|---|---|---|
| CoOp | 81.31 | 82.96 | +1.65 |
| CoCoOp | 79.65 | 82.92 | +3.27 |
| IVLP | 78.96 | 83.84 | +4.88 |
| KgCoOp | 79.04 | 82.76 | +3.72 |
| ProGrad | 79.39 | 82.30 | +2.91 |
| MaPLe | 79.38 | 83.45 | +4.07 |
| PromptSRC | 81.77 | 84.56 | +2.79 |
| LoCoOp | 73.63 | 81.13 | +7.50 |

(m) CIFAR10.

|  | Energy | MLS-E | △ |
|---|---|---|---|
| CoOp | 82.84 | 84.86 | +2.02 |
| CoCoOp | 88.08 | 91.91 | +3.83 |
| IVLP | 86.40 | 90.40 | +3.99 |
| KgCoOp | 88.09 | 91.93 | +3.84 |
| ProGrad | 81.61 | 87.96 | +6.35 |
| MaPLe | 85.68 | 91.07 | +5.38 |
| PromptSRC | 89.76 | 93.41 | +3.65 |
| LoCoOp | 81.33 | 89.26 | +7.93 |

(n) CIFAR100.

|  | Energy | MLS-E | △ |
|---|---|---|---|
| CoOp | 76.84 | 77.75 | +0.91 |
| CoCoOp | 79.64 | 80.73 | +1.09 |
| IVLP | 82.35 | 87.05 | +4.70 |
| KgCoOp | 78.54 | 79.78 | +1.23 |
| ProGrad | 74.71 | 78.76 | +4.05 |
| MaPLe | 83.17 | 87.49 | +4.32 |
| PromptSRC | 84.76 | 87.83 | +3.07 |
| LoCoOp | 73.54 | 74.83 | +1.28 |

Table 13: Near OOD FPR95 (↓) of prompt learning models over 13 datasets using the Energy score and MLS-E.

(a) Average over 13 datasets.

|          | Energy | MLS-E | △      |
|----------|--------|-------|--------|
| CoOp     | 59.45  | 55.37 | -4.08  |
| CoCoOp   | 57.49  | 51.90 | -5.59  |
| IVLP     | 57.66  | 49.19 | -8.47  |
| KgCoOp   | 59.69  | 51.90 | -7.80  |
| ProGrad  | 62.74  | 55.34 | -7.41  |
| MaPLe    | 57.82  | 48.76 | -9.07  |
| PromptSRC| 51.86  | 45.15 | -6.71  |
| LoCoOp   | 69.07  | 57.80 | -11.27 |

(b) ImageNet.

|          | Energy | MLS-E | △      |
|----------|--------|-------|--------|
| CoOp     | 33.04  | 26.96 | -6.08  |
| CoCoOp   | 29.16  | 24.69 | -4.47  |
| IVLP     | 29.40  | 24.78 | -4.62  |
| KgCoOp   | 33.09  | 28.35 | -4.74  |
| ProGrad  | 35.87  | 28.41 | -7.46  |
| MaPLe    | 32.34  | 26.52 | -5.82  |
| PromptSRC| 30.06  | 22.97 | -7.09  |
| LoCoOp   | 45.33  | 32.02 | -13.31 |

(c) Caltech101.

|          | Energy | MLS-E | △      |
|----------|--------|-------|--------|
| CoOp     | 42.21  | 32.07 | -10.15 |
| CoCoOp   | 48.28  | 36.22 | -12.06 |
| IVLP     | 51.35  | 34.09 | -17.26 |
| KgCoOp   | 61.27  | 34.10 | -27.17 |
| ProGrad  | 65.31  | 45.11 | -20.20 |
| MaPLe    | 49.42  | 31.23 | -18.19 |
| PromptSRC| 52.12  | 28.81 | -23.30 |
| LoCoOp   | 80.74  | 44.34 | -36.40 |

(d) OxfordPets.

|          | Energy | MLS-E | △      |
|----------|--------|-------|--------|
| CoOp     | 52.87  | 46.91 | -5.95  |
| CoCoOp   | 45.06  | 37.87 | -7.18  |
| IVLP     | 50.67  | 43.75 | -6.92  |
| KgCoOp   | 49.07  | 38.01 | -11.06 |
| ProGrad  | 51.49  | 52.73 | +1.23  |
| MaPLe    | 53.88  | 45.57 | -8.30  |
| PromptSRC| 44.32  | 39.70 | -4.62  |
| LoCoOp   | 58.27  | 52.35 | -5.92  |

(e) StanfordCars.

|          | Energy | MLS-E | △      |
|----------|--------|-------|--------|
| CoOp     | 32.23  | 30.96 | -1.27  |
| CoCoOp   | 28.37  | 26.83 | -1.54  |
| IVLP     | 33.75  | 25.40 | -8.34  |
| KgCoOp   | 28.62  | 27.05 | -1.57  |
| ProGrad  | 32.69  | 28.19 | -4.49  |
| MaPLe    | 30.20  | 25.29 | -4.91  |
| PromptSRC| 26.11  | 22.29 | -3.82  |
| LoCoOp   | 42.86  | 31.34 | -11.52 |

(f) Flowers102.

|          | Energy | MLS-E | △      |
|----------|--------|-------|--------|
| CoOp     | 42.18  | 37.55 | -4.63  |
| CoCoOp   | 53.82  | 46.99 | -6.83  |
| IVLP     | 59.10  | 51.31 | -7.79  |
| KgCoOp   | 62.44  | 45.23 | -17.21 |
| ProGrad  | 49.71  | 43.54 | -6.17  |
| MaPLe    | 58.30  | 50.79 | -7.52  |
| PromptSRC| 42.84  | 36.78 | -6.06  |
| LoCoOp   | 59.87  | 51.74 | -8.14  |

(g) Food101

|          | Energy | MLS-E | △      |
|----------|--------|-------|--------|
| CoOp     | 55.28  | 50.05 | -5.23  |
| CoCoOp   | 43.18  | 37.95 | -5.23  |
| IVLP     | 45.66  | 35.66 | -10.00 |
| KgCoOp   | 46.71  | 36.35 | -10.36 |
| ProGrad  | 50.78  | 39.99 | -10.79 |
| MaPLe    | 47.53  | 35.38 | -12.15 |
| PromptSRC| 40.21  | 34.84 | -5.36  |
| LoCoOp   | 65.29  | 44.22 | -21.06 |

(h) FGVCAircraft.

|          | Energy | MLS-E | △      |
|----------|--------|-------|--------|
| CoOp     | 81.31  | 80.09 | -1.22  |
| CoCoOp   | 81.38  | 76.55 | -4.83  |
| IVLP     | 75.61  | 69.78 | -5.83  |
| KgCoOp   | 72.11  | 72.30 | +0.18  |
| ProGrad  | 82.41  | 78.20 | -4.21  |
| MaPLe    | 78.68  | 75.01 | -3.67  |
| PromptSRC| 69.47  | 69.00 | -0.48  |
| LoCoOp   | 83.64  | 79.53 | -4.11  |

(i) SUN397.

|          | Energy | MLS-E | △      |
|----------|--------|-------|--------|
| CoOp     | 73.24  | 70.54 | -2.69  |
| CoCoOp   | 73.55  | 67.30 | -6.25  |
| IVLP     | 71.54  | 65.08 | -6.46  |
| KgCoOp   | 74.01  | 70.02 | -3.99  |
| ProGrad  | 76.75  | 70.20 | -6.56  |
| MaPLe    | 71.20  | 64.64 | -6.56  |
| PromptSRC| 69.99  | 63.56 | -6.43  |
| LoCoOp   | 81.12  | 71.02 | -10.10 |

(j) DTD.

|          | Energy | MLS-E | △      |
|----------|--------|-------|--------|
| CoOp     | 86.00  | 85.21 | -0.79  |
| CoCoOp   | 88.92  | 87.58 | -1.34  |
| IVLP     | 87.86  | 87.82 | -0.04  |
| KgCoOp   | 88.49  | 86.42 | -2.08  |
| ProGrad  | 88.90  | 88.31 | -0.59  |
| MaPLe    | 89.53  | 87.31 | -2.22  |
| PromptSRC| 86.43  | 84.32 | -2.11  |
| LoCoOp   | 85.52  | 85.52 | +0.01  |

(k) EuroSAT.

|          | Energy | MLS-E | △      |
|----------|--------|-------|--------|
| CoOp     | 85.49  | 85.25 | -0.24  |
| CoCoOp   | 82.05  | 81.58 | -0.47  |
| IVLP     | 81.29  | 74.34 | -6.96  |
| KgCoOp   | 82.60  | 79.15 | -3.45  |
| ProGrad  | 82.91  | 88.90 | +0.09  |
| MaPLe    | 76.75  | 67.28 | -9.47  |
| PromptSRC| 67.18  | 66.27 | -0.91  |
| LoCoOp   | 82.06  | 79.99 | -2.07  |

(l) UCF101.

|          | Energy | MLS-E | △      |
|----------|--------|-------|--------|
| CoOp     | 56.84  | 51.16 | -5.68  |
| CoCoOp   | 58.31  | 50.90 | -7.41  |
| IVLP     | 58.72  | 48.52 | -10.20 |
| KgCoOp   | 56.41  | 50.89 | -5.52  |
| ProGrad  | 59.53  | 50.75 | -8.78  |
| MaPLe    | 58.65  | 49.51 | -9.14  |
| PromptSRC| 53.91  | 47.33 | -6.58  |
| LoCoOp   | 70.46  | 57.05 | -13.41 |

(m) CIFAR10.

|          | Energy | MLS-E | △      |
|----------|--------|-------|--------|
| CoOp     | 56.82  | 51.62 | -5.20  |
| CoCoOp   | 41.65  | 30.48 | -11.17 |
| IVLP     | 43.38  | 32.73 | -10.65 |
| KgCoOp   | 45.90  | 33.20 | -12.70 |
| ProGrad  | 58.65  | 43.18 | -15.47 |
| MaPLe    | 46.10  | 31.57 | -14.53 |
| PromptSRC| 35.64  | 24.13 | -11.51 |
| LoCoOp   | 60.82  | 39.74 | -21.08 |

(n) CIFAR100.

|          | Energy | MLS-E | △      |
|----------|--------|-------|--------|
| CoOp     | 75.30  | 71.41 | -3.89  |
| CoCoOp   | 73.60  | 69.70 | -3.90  |
| IVLP     | 61.31  | 46.22 | -15.08 |
| KgCoOp   | 75.27  | 73.58 | -1.69  |
| ProGrad  | 80.70  | 67.79 | -12.91 |
| MaPLe    | 59.11  | 43.73 | -15.38 |
| PromptSRC| 55.95  | 47.00 | -8.95  |
| LoCoOp   | 81.91  | 82.49 | +0.59  |

Table 14: Near OOD AUROC (↑) of prompt learning models over 13 datasets using the Energy score and MLS-E with 16-shots.

### (a) Average over 13 datasets.

|  | Energy | MLS-E | △ |
|---|---|---|---|
| CoOp | 82.67±10.20 | 83.84±10.18 | +1.17±1.20 |
| CoCoOp | 81.21±12.20 | 83.70±10.99 | +2.49±2.54 |
| IVLP | 83.25±9.90 | 87.55±7.60 | +4.30±4.05 |
| KgCoOp | 81.02±11.18 | 84.36±9.84 | +3.34±2.72 |
| ProGrad | 80.90±11.25 | 83.32±11.27 | +2.42±2.81 |
| MaPLe | 83.60±10.11 | 86.54±9.03 | +2.94±2.28 |
| PromptSRC | 85.53±8.47 | 87.67±8.19 | +2.13±1.90 |
| LoCoOp | 78.03±10.52 | 81.70±11.89 | +3.67±4.56 |

### (b) ImageNet.

|  | Energy | MLS-E | △ |
|---|---|---|---|
| CoOp | 93.74±0.33 | 95.56±0.27 | +1.82±0.06 |
| CoCoOp | 94.85±0.32 | 95.38±0.75 | +0.53±0.90 |
| IVLP | 94.49±0.16 | 94.84±0.50 | +0.36±0.48 |
| KgCoOp | 94.15±0.05 | 94.39±0.25 | +0.24±0.30 |
| ProGrad | 94.03±0.13 | 95.15±0.09 | +1.12±0.14 |
| MaPLe | 94.64±0.55 | 95.00±0.51 | +0.36±0.42 |
| PromptSRC | 94.64±0.21 | 95.88±0.16 | +1.25±0.22 |
| LoCoOp | 93.06±0.25 | 95.10±0.29 | +2.04±0.54 |

### (c) Caltech101.

|  | Energy | MLS-E | △ |
|---|---|---|---|
| CoOp | 89.02±0.89 | 90.88±1.16 | +1.87±1.31 |
| CoCoOp | 85.89±1.40 | 89.16±1.35 | +3.27±2.20 |
| IVLP | 86.56±1.70 | 92.84±1.90 | +6.27±2.02 |
| KgCoOp | 80.89±0.06 | 89.13±0.43 | +8.24±0.49 |
| ProGrad | 79.45±2.82 | 88.37±0.45 | +8.92±2.73 |
| MaPLe | 86.60±0.87 | 92.58±1.67 | +5.98±2.16 |
| PromptSRC | 82.29±0.19 | 89.74±0.16 | +7.44±0.11 |
| LoCoOp | 74.46±1.75 | 87.26±0.73 | +12.80±1.62 |

### (d) OxfordPets.

|  | Energy | MLS-E | △ |
|---|---|---|---|
| CoOp | 84.87±1.44 | 86.40±1.96 | +1.52±0.54 |
| CoCoOp | 90.51±0.79 | 93.43±0.31 | +2.92±0.70 |
| IVLP | 90.80±1.97 | 93.60±1.41 | +2.80±0.80 |
| KgCoOp | 89.58±0.21 | 91.70±1.01 | +2.12±0.89 |
| ProGrad | 87.13±1.50 | 88.41±0.68 | +1.28±1.05 |
| MaPLe | 89.43±0.65 | 93.96±1.35 | +4.53±1.38 |
| PromptSRC | 91.23±0.29 | 94.58±0.45 | +3.36±0.20 |
| LoCoOp | 82.84±3.33 | 88.48±1.24 | +5.64±2.74 |

### (e) StanfordCars.

|  | Energy | MLS-E | △ |
|---|---|---|---|
| CoOp | 93.48±0.19 | 93.41±0.20 | -0.07±0.18 |
| CoCoOp | 91.39±1.69 | 92.97±0.86 | +1.58±1.07 |
| IVLP | 90.53±1.54 | 93.44±1.29 | +2.90±0.30 |
| KgCoOp | 93.03±0.23 | 93.61±0.08 | +0.59±0.25 |
| ProGrad | 93.03±0.10 | 93.90±0.13 | +0.86±0.06 |
| MaPLe | 91.51±1.54 | 93.46±0.20 | +1.95±1.50 |
| PromptSRC | 93.86±0.16 | 95.49±0.01 | +1.63±0.18 |
| LoCoOp | 88.72±1.18 | 92.35±1.87 | +3.63±0.92 |

### (f) Flowers102.

|  | Energy | MLS-E | △ |
|---|---|---|---|
| CoOp | 93.69±0.72 | 94.14±0.65 | +0.45±0.08 |
| CoCoOp | 88.15±1.05 | 90.26±0.13 | +2.11±0.99 |
| IVLP | 89.67±3.16 | 91.96±1.95 | +2.28±1.22 |
| KgCoOp | 87.91±0.45 | 92.03±0.53 | +4.12±0.43 |
| ProGrad | 88.36±1.15 | 91.54±0.28 | +3.18±0.87 |
| MaPLe | 88.72±0.83 | 90.62±1.40 | +1.90±0.67 |
| PromptSRC | 93.36±0.19 | 94.96±0.13 | +1.59±0.30 |
| LoCoOp | 84.27±2.11 | 87.83±0.61 | +3.56±2.63 |

### (g) Food101

|  | Energy | MLS-E | △ |
|---|---|---|---|
| CoOp | 88.26±0.17 | 89.91±0.19 | +1.65±0.13 |
| CoCoOp | 90.04±0.56 | 92.29±0.80 | +2.25±0.30 |
| IVLP | 89.87±1.04 | 92.63±0.48 | +2.76±1.48 |
| KgCoOp | 89.71±0.06 | 92.76±0.03 | +3.04±0.06 |
| ProGrad | 89.78±0.36 | 92.64±0.27 | +2.87±0.10 |
| MaPLe | 90.26±1.07 | 93.01±0.63 | +2.74±0.50 |
| PromptSRC | 91.05±0.24 | 92.49±0.21 | +1.44±0.07 |
| LoCoOp | 83.07±1.58 | 90.78±0.93 | +7.71±2.00 |

### (h) FGVCAircraft.

|  | Energy | MLS-E | △ |
|---|---|---|---|
| CoOp | 58.67±1.68 | 59.64±1.78 | +0.98±1.53 |
| CoCoOp | 54.18±2.15 | 62.75±2.43 | +8.57±3.29 |
| IVLP | 64.80±2.11 | 75.75±0.99 | +10.95±1.23 |
| KgCoOp | 65.66±0.55 | 67.11±1.28 | +1.45±0.74 |
| ProGrad | 58.30±0.53 | 58.03±4.57 | -0.28±4.51 |
| MaPLe | 61.47±5.81 | 68.17±4.95 | +6.71±1.30 |
| PromptSRC | 71.61±1.30 | 74.28±2.01 | +2.67±0.91 |
| LoCoOp | 56.94±2.18 | 52.02±3.01 | -4.92±2.79 |

### (i) SUN397.

|  | Energy | MLS-E | △ |
|---|---|---|---|
| CoOp | 77.47±0.49 | 78.91±0.71 | +1.45±0.29 |
| CoCoOp | 76.14±0.51 | 78.46±0.49 | +2.31±0.37 |
| IVLP | 76.77±0.53 | 80.07±0.26 | +3.30±0.78 |
| KgCoOp | 75.66±0.09 | 77.43±0.22 | +1.76±0.27 |
| ProGrad | 75.33±1.06 | 78.14±0.99 | +2.81±0.32 |
| MaPLe | 77.24±1.89 | 80.02±0.76 | +2.79±1.18 |
| PromptSRC | 78.24±0.09 | 81.28±0.58 | +3.04±0.49 |
| LoCoOp | 73.04±0.30 | 77.96±0.65 | +4.92±0.89 |

### (j) DTD.

|  | Energy | MLS-E | △ |
|---|---|---|---|
| CoOp | 73.46±1.11 | 73.78±1.37 | +0.32±0.30 |
| CoCoOp | 64.88±1.32 | 68.81±1.39 | +3.93±2.01 |
| IVLP | 66.08±1.73 | 70.45±0.85 | +4.37±1.00 |
| KgCoOp | 62.52±0.75 | 69.68±0.49 | +7.15±0.46 |
| ProGrad | 61.18±1.21 | 65.78±1.57 | +4.61±1.22 |
| MaPLe | 66.07±3.15 | 70.26±2.28 | +4.19±2.02 |
| PromptSRC | 70.23±0.69 | 72.32±0.79 | +2.09±0.69 |
| LoCoOp | 66.54±0.84 | 71.23±0.91 | +4.69±0.82 |

### (k) EuroSAT.

|  | Energy | MLS-E | △ |
|---|---|---|---|
| CoOp | 71.11±0.80 | 72.98±2.62 | +1.87±1.90 |
| CoCoOp | 67.77±3.30 | 67.38±3.57 | -0.39±0.30 |
| IVLP | 72.49±2.75 | 85.82±2.86 | +13.34±3.57 |
| KgCoOp | 62.59±0.31 | 70.32±1.20 | +7.73±0.99 |
| ProGrad | 74.04±2.13 | 74.19±2.13 | +0.15±0.09 |
| MaPLe | 79.57±1.14 | 81.41±2.37 | +1.84±1.48 |
| PromptSRC | 78.87±2.45 | 78.58±2.27 | -0.29±0.27 |
| LoCoOp | 67.22±5.88 | 68.64±5.90 | +1.41±0.55 |

### (l) UCF101.

|  | Energy | MLS-E | △ |
|---|---|---|---|
| CoOp | 82.77±0.52 | 84.74±0.62 | +1.98±1.08 |
| CoCoOp | 81.32±1.02 | 84.19±0.56 | +2.87±0.64 |
| IVLP | 81.30±2.34 | 85.95±0.24 | +4.64±2.18 |
| KgCoOp | 79.79±0.09 | 83.61±0.82 | +3.82±0.73 |
| ProGrad | 79.94±0.49 | 82.33±0.97 | +2.39±0.49 |
| MaPLe | 81.63±1.04 | 83.84±1.25 | +2.21±0.32 |
| PromptSRC | 82.61±0.58 | 84.96±0.67 | +2.35±0.25 |
| LoCoOp | 75.72±1.15 | 82.57±1.31 | +6.85±2.26 |

### (m) CIFAR10.

|  | Energy | MLS-E | △ |
|---|---|---|---|
| CoOp | 90.39±0.84 | 90.60±0.76 | +0.20±0.26 |
| CoCoOp | 93.84±0.18 | 93.95±0.38 | +0.10±0.40 |
| IVLP | 93.01±1.30 | 93.10±1.66 | +0.09±0.45 |
| KgCoOp | 93.46±0.12 | 93.86±0.12 | +0.40±0.03 |
| ProGrad | 92.94±0.40 | 93.30±0.28 | +0.36±0.16 |
| MaPLe | 94.24±0.58 | 94.26±0.79 | +0.02±0.43 |
| PromptSRC | 95.13±0.09 | 94.88±0.35 | -0.24±0.29 |
| LoCoOp | 92.34±0.70 | 91.13±0.46 | -1.21±0.59 |

### (n) CIFAR100.

|  | Energy | MLS-E | △ |
|---|---|---|---|
| CoOp | 77.81±0.91 | 78.93±1.06 | +1.12±1.78 |
| CoCoOp | 76.76±1.06 | 79.06±1.14 | +2.31±0.94 |
| IVLP | 85.87±0.90 | 87.68±2.39 | +1.81±1.55 |
| KgCoOp | 78.28±0.43 | 81.07±0.32 | +2.79±0.51 |
| ProGrad | 78.21±0.48 | 81.40±0.68 | +3.19±0.47 |
| MaPLe | 85.48±0.77 | 88.50±2.08 | +3.01±1.34 |
| PromptSRC | 88.83±0.25 | 90.21±0.13 | +1.38±0.38 |
| LoCoOp | 76.19±1.57 | 76.73±1.77 | +0.54±1.17 |

Table 15: Near OOD AUROC (↑) of prompt learning models over 13 datasets using the Energy score and MLS-E with 8-shots.

### (a) Average over 13 datasets.

|  | Energy | MLS-E | △ |
|---|---|---|---|
| CoOp | 81.38±10.64 | 82.77±10.65 | +1.39±1.17 |
| CoCoOp | 81.16±11.79 | 83.44±10.96 | +2.28±1.87 |
| IVLP | 81.92±11.35 | 85.27±10.31 | +3.34±3.78 |
| KgCoOp | 80.61±11.35 | 83.71±10.47 | +3.10±2.60 |
| ProGrad | 79.71±12.08 | 82.65±11.65 | +2.93±2.57 |
| MaPLe | 82.29±11.58 | 85.45±11.08 | +3.16±3.24 |
| PromptSRC | 85.28±8.51 | 87.53±8.05 | +2.26±1.90 |
| LoCoOp | 78.06±10.24 | 82.63±9.46 | +4.57±4.78 |

### (b) ImageNet.

|  | Energy | MLS-E | △ |
|---|---|---|---|
| CoOp | 93.29±0.22 | 95.34±0.45 | +2.05±0.64 |
| CoCoOp | 94.78±0.62 | 95.89±0.09 | +1.11±0.56 |
| IVLP | 94.18±1.20 | 94.51±0.61 | +0.34±1.68 |
| KgCoOp | 94.16±0.03 | 94.22±0.43 | +0.05±0.41 |
| ProGrad | 93.68±0.72 | 95.14±0.25 | +1.47±0.71 |
| MaPLe | 93.56±1.53 | 94.72±0.96 | +1.16±0.63 |
| PromptSRC | 94.57±0.20 | 95.81±0.08 | +1.24±0.14 |
| LoCoOp | 91.81±0.61 | 94.26±0.56 | +2.45±0.52 |

### (c) Caltech101.

|  | Energy | MLS-E | △ |
|---|---|---|---|
| CoOp | 87.86±1.05 | 89.96±1.31 | +2.09±1.00 |
| CoCoOp | 84.16±2.29 | 87.06±2.75 | +2.90±1.66 |
| IVLP | 86.96±1.63 | 91.78±1.38 | +4.82±0.57 |
| KgCoOp | 80.33±0.24 | 88.50±0.27 | +8.17±0.15 |
| ProGrad | 81.78±1.73 | 88.09±1.39 | +6.31±1.76 |
| MaPLe | 85.75±1.21 | 91.54±1.09 | +5.79±2.12 |
| PromptSRC | 82.61±0.45 | 89.88±0.67 | +7.27±0.55 |
| LoCoOp | 73.11±2.77 | 88.20±1.08 | +15.10±2.96 |

### (d) OxfordPets.

|  | Energy | MLS-E | △ |
|---|---|---|---|
| CoOp | 86.31±2.02 | 88.83±1.92 | +2.53±0.15 |
| CoCoOp | 90.57±0.75 | 92.98±0.16 | +2.41±0.70 |
| IVLP | 89.92±0.68 | 91.94±0.65 | +2.02±0.89 |
| KgCoOp | 89.48±0.16 | 92.12±0.35 | +2.64±0.35 |
| ProGrad | 86.58±1.15 | 89.05±1.69 | +2.47±0.54 |
| MaPLe | 88.42±0.71 | 92.02±1.36 | +3.60±0.83 |
| PromptSRC | 91.19±0.27 | 93.77±0.38 | +2.58±0.24 |
| LoCoOp | 84.20±3.03 | 88.73±1.24 | +4.53±2.83 |

### (e) StanfordCars.

|  | Energy | MLS-E | △ |
|---|---|---|---|
| CoOp | 92.24±0.57 | 93.06±1.08 | +0.82±0.58 |
| CoCoOp | 92.83±1.12 | 93.50±1.37 | +0.67±0.58 |
| IVLP | 89.48±2.02 | 93.57±1.33 | +4.09±2.67 |
| KgCoOp | 92.97±0.10 | 93.51±0.13 | +0.54±0.06 |
| ProGrad | 92.76±0.97 | 93.15±0.67 | +0.39±0.44 |
| MaPLe | 92.47±0.69 | 93.24±1.05 | +0.77±0.36 |
| PromptSRC | 93.02±0.22 | 94.64±0.83 | +1.62±0.68 |
| LoCoOp | 88.46±0.57 | 92.62±1.03 | +4.17±1.25 |

### (f) Flowers102.

|  | Energy | MLS-E | △ |
|---|---|---|---|
| CoOp | 91.48±0.50 | 92.37±0.52 | +0.89±0.11 |
| CoCoOp | 87.02±0.02 | 88.32±0.43 | +1.30±0.45 |
| IVLP | 86.79±4.78 | 87.59±4.46 | +0.80±0.32 |
| KgCoOp | 86.43±0.09 | 91.65±0.48 | +5.22±0.49 |
| ProGrad | 88.67±0.30 | 91.35±0.56 | +2.68±0.67 |
| MaPLe | 85.83±0.77 | 87.63±0.67 | +1.81±0.82 |
| PromptSRC | 92.24±0.37 | 94.05±0.23 | +1.81±0.14 |
| LoCoOp | 86.09±2.83 | 88.96±2.00 | +2.87±0.84 |

### (g) Food101

|  | Energy | MLS-E | △ |
|---|---|---|---|
| CoOp | 87.40±1.19 | 87.98±1.22 | +0.58±0.21 |
| CoCoOp | 90.81±0.62 | 91.47±0.75 | +0.66±0.30 |
| IVLP | 88.77±1.69 | 92.09±0.36 | +3.32±1.67 |
| KgCoOp | 89.28±0.13 | 91.85±0.42 | +2.57±0.43 |
| ProGrad | 88.63±0.47 | 90.80±0.54 | +2.18±0.45 |
| MaPLe | 89.99±1.07 | 92.60±0.29 | +2.61±1.17 |
| PromptSRC | 91.03±0.27 | 92.31±0.43 | +1.28±0.45 |
| LoCoOp | 83.90±2.11 | 88.93±1.64 | +5.03±0.63 |

### (h) FGVCAircraft.

|  | Energy | MLS-E | △ |
|---|---|---|---|
| CoOp | 58.51±6.36 | 59.92±6.08 | +1.41±2.28 |
| CoCoOp | 58.61±1.00 | 63.19±3.58 | +4.58±3.21 |
| IVLP | 65.14±5.72 | 71.74±2.69 | +6.60±8.26 |
| KgCoOp | 65.74±0.96 | 67.61±0.96 | +1.87±1.73 |
| ProGrad | 58.66±2.45 | 60.02±7.36 | +1.36±5.42 |
| MaPLe | 53.69±8.20 | 59.43±14.57 | +5.74±8.67 |
| PromptSRC | 71.14±0.43 | 75.76±0.58 | +4.61±0.83 |
| LoCoOp | 60.78±7.50 | 67.63±9.05 | +6.85±3.06 |

### (i) SUN397.

|  | Energy | MLS-E | △ |
|---|---|---|---|
| CoOp | 76.02±1.27 | 77.14±0.85 | +1.12±0.50 |
| CoCoOp | 75.58±0.45 | 78.93±1.13 | +3.35±1.56 |
| IVLP | 77.43±0.18 | 79.94±0.97 | +2.51±0.94 |
| KgCoOp | 75.27±0.18 | 77.23±0.45 | +1.96±0.44 |
| ProGrad | 74.63±0.37 | 78.07±0.63 | +3.45±0.86 |
| MaPLe | 78.23±1.20 | 79.91±1.13 | +1.69±0.20 |
| PromptSRC | 77.94±0.24 | 80.88±0.55 | +2.93±0.36 |
| LoCoOp | 72.28±1.21 | 77.06±1.26 | +4.77±1.50 |

### (j) DTD.

|  | Energy | MLS-E | △ |
|---|---|---|---|
| CoOp | 70.79±0.49 | 71.54±1.12 | +0.74±0.63 |
| CoCoOp | 63.93±0.38 | 67.29±0.37 | +3.36±0.19 |
| IVLP | 66.04±0.36 | 68.68±0.39 | +2.64±0.10 |
| KgCoOp | 60.85±0.26 | 66.50±1.25 | +5.65±1.13 |
| ProGrad | 59.71±1.04 | 64.97±0.88 | +5.26±0.60 |
| MaPLe | 65.47±1.47 | 69.84±1.95 | +4.37±0.60 |
| PromptSRC | 69.00±1.03 | 70.36±1.04 | +1.36±0.05 |
| LoCoOp | 66.91±0.80 | 69.47±0.47 | +2.57±0.48 |

### (k) EuroSAT.

|  | Energy | MLS-E | △ |
|---|---|---|---|
| CoOp | 67.67±3.77 | 68.53±3.37 | +0.86±0.45 |
| CoCoOp | 65.60±3.79 | 66.79±3.74 | +1.19±0.73 |
| IVLP | 61.18±10.79 | 69.72±15.85 | +8.54±5.16 |
| KgCoOp | 62.19±0.52 | 67.08±3.15 | +4.89±3.56 |
| ProGrad | 63.62±2.46 | 66.39±0.81 | +2.78±2.66 |
| MaPLe | 77.01±4.54 | 83.09±6.48 | +6.07±2.26 |
| PromptSRC | 80.14±4.58 | 80.46±4.80 | +0.32±0.21 |
| LoCoOp | 66.85±0.86 | 71.13±4.97 | +4.28±4.19 |

### (l) UCF101.

|  | Energy | MLS-E | △ |
|---|---|---|---|
| CoOp | 82.79±1.00 | 84.92±0.78 | +2.14±0.76 |
| CoCoOp | 81.08±1.02 | 84.27±0.40 | +3.19±0.76 |
| IVLP | 79.97±1.70 | 85.60±1.40 | +5.63±0.99 |
| KgCoOp | 79.53±0.41 | 83.58±0.46 | +4.05±0.62 |
| ProGrad | 78.41±2.14 | 83.91±2.00 | +5.50±0.14 |
| MaPLe | 79.75±0.93 | 83.65±0.55 | +3.90±1.41 |
| PromptSRC | 83.12±0.90 | 85.94±0.98 | +2.82±0.27 |
| LoCoOp | 72.75±2.31 | 82.37±1.28 | +9.62±1.61 |

### (m) CIFAR10.

|  | Energy | MLS-E | △ |
|---|---|---|---|
| CoOp | 88.11±2.48 | 88.12±2.73 | +0.01±0.26 |
| CoCoOp | 92.96±0.58 | 93.38±0.92 | +0.42±0.43 |
| IVLP | 92.27±1.15 | 93.34±0.78 | +1.07±0.43 |
| KgCoOp | 93.33±0.01 | 93.67±0.16 | +0.34±0.14 |
| ProGrad | 92.31±0.54 | 92.71±0.70 | +0.40±0.28 |
| MaPLe | 93.54±0.44 | 93.97±0.47 | +0.43±0.20 |
| PromptSRC | 94.92±0.47 | 94.75±0.45 | -0.16±0.31 |
| LoCoOp | 91.96±1.27 | 88.62±3.55 | -3.34±3.08 |

### (n) CIFAR100.

|  | Energy | MLS-E | △ |
|---|---|---|---|
| CoOp | 75.41±0.95 | 78.30±0.46 | +2.89±0.57 |
| CoCoOp | 77.14±1.36 | 81.60±0.63 | +4.45±1.23 |
| IVLP | 86.85±0.73 | 87.94±0.63 | +1.09±0.45 |
| KgCoOp | 78.39±0.44 | 80.69±0.94 | +2.30±1.23 |
| ProGrad | 76.86±0.64 | 80.76±1.26 | +3.90±0.76 |
| MaPLe | 86.01±0.50 | 89.21±0.23 | +3.19±0.31 |
| PromptSRC | 87.67±0.69 | 89.33±0.83 | +1.66±0.18 |
| LoCoOp | 75.70±0.75 | 76.16±2.61 | +0.46±1.86 |

Table 16: Near OOD AUROC (↑) of prompt learning models over 13 datasets using the Energy score and MLS-E with 4-shots.

(a) Average over 13 datasets.

| | Energy | MLS-E | △ |
|---|---|---|---|
| CoOp | 80.96±10.65 | 82.10±10.96 | +1.15±1.67 |
| CoCoOp | 81.06±11.56 | 83.05±11.69 | +1.99±2.22 |
| IVLP | 80.84±12.25 | 84.23±11.72 | +3.39±2.85 |
| KgCoOp | 80.36±11.15 | 83.31±10.95 | +2.95±2.45 |
| ProGrad | 79.95±10.93 | 82.21±10.85 | +2.26±1.91 |
| MaPLe | 80.39±11.25 | 83.76±10.48 | +3.37±3.27 |
| PromptSRC | 84.22±8.81 | 86.37±8.78 | +2.15±1.78 |
| LoCoOp | 77.45±10.27 | 81.35±10.79 | +3.90±5.10 |

(b) ImageNet.

| | Energy | MLS-E | △ |
|---|---|---|---|
| CoOp | 93.77±0.52 | 95.28±0.21 | +1.50±0.71 |
| CoCoOp | 95.25±0.06 | 95.87±0.11 | +0.62±0.10 |
| IVLP | 95.03±0.05 | 95.60±0.17 | +0.57±0.20 |
| KgCoOp | 94.03±0.07 | 94.47±0.07 | +0.43±0.08 |
| ProGrad | 93.72±0.37 | 94.73±0.26 | +1.01±0.22 |
| MaPLe | 93.79±1.60 | 94.00±0.70 | +0.21±0.95 |
| PromptSRC | 94.49±0.11 | 95.67±0.24 | +1.18±0.23 |
| LoCoOp | 92.51±0.31 | 94.90±0.13 | +2.39±0.40 |

(c) Caltech101.

| | Energy | MLS-E | △ |
|---|---|---|---|
| CoOp | 89.67±1.88 | 91.35±2.71 | +1.67±1.55 |
| CoCoOp | 85.37±1.02 | 88.46±0.33 | +3.09±1.23 |
| IVLP | 82.49±1.10 | 91.07±0.79 | +8.57±1.87 |
| KgCoOp | 80.41±0.10 | 88.88±0.74 | +8.47±0.69 |
| ProGrad | 82.12±1.31 | 87.22±1.70 | +5.10±1.99 |
| MaPLe | 83.52±4.07 | 90.61±1.95 | +7.09±4.68 |
| PromptSRC | 82.95±0.45 | 89.72±0.36 | +6.77±0.30 |
| LoCoOp | 71.40±2.18 | 86.42±1.52 | +15.02±0.79 |

(d) OxfordPets.

| | Energy | MLS-E | △ |
|---|---|---|---|
| CoOp | 87.32±2.01 | 89.57±1.80 | +2.26±0.22 |
| CoCoOp | 88.88±1.08 | 92.69±0.75 | +3.81±1.65 |
| IVLP | 88.91±1.64 | 93.05±0.53 | +4.14±1.27 |
| KgCoOp | 89.34±0.08 | 92.86±0.31 | +3.52±0.35 |
| ProGrad | 87.25±1.22 | 89.33±1.55 | +2.08±0.34 |
| MaPLe | 85.70±2.55 | 91.03±1.49 | +5.33±1.64 |
| PromptSRC | 90.31±0.99 | 93.77±0.99 | +3.46±0.32 |
| LoCoOp | 84.77±1.22 | 88.96±1.47 | +4.19±1.08 |

(e) StanfordCars.

| | Energy | MLS-E | △ |
|---|---|---|---|
| CoOp | 91.15±1.35 | 91.46±1.09 | +0.31±0.42 |
| CoCoOp | 93.45±0.67 | 93.80±0.54 | +0.35±0.30 |
| IVLP | 92.10±1.07 | 93.13±1.07 | +1.03±0.06 |
| KgCoOp | 92.92±0.11 | 93.56±0.19 | +0.65±0.16 |
| ProGrad | 92.22±1.02 | 92.77±0.75 | +0.55±0.61 |
| MaPLe | 92.09±1.08 | 93.29±0.71 | +1.20±1.21 |
| PromptSRC | 92.88±0.20 | 94.09±0.33 | +1.22±0.42 |
| LoCoOp | 86.92±1.81 | 91.75±0.44 | +4.83±2.07 |

(f) Flowers102.

| | Energy | MLS-E | △ |
|---|---|---|---|
| CoOp | 91.32±1.85 | 92.79±1.47 | +1.47±1.14 |
| CoCoOp | 86.33±1.79 | 88.59±0.54 | +2.26±1.32 |
| IVLP | 85.48±2.02 | 87.61±1.04 | +2.13±1.12 |
| KgCoOp | 85.19±0.47 | 89.96±0.09 | +4.77±0.38 |
| ProGrad | 87.02±2.01 | 90.27±1.66 | +3.25±1.06 |
| MaPLe | 83.84±2.69 | 86.96±1.59 | +3.12±1.74 |
| PromptSRC | 90.48±0.17 | 92.09±0.82 | +1.61±0.65 |
| LoCoOp | 84.97±0.79 | 87.66±0.73 | +2.69±0.88 |

(g) Food101

| | Energy | MLS-E | △ |
|---|---|---|---|
| CoOp | 86.33±1.54 | 87.58±1.09 | +1.25±0.51 |
| CoCoOp | 90.04±1.32 | 91.66±0.73 | +1.62±0.62 |
| IVLP | 88.91±0.68 | 92.08±0.74 | +3.17±0.71 |
| KgCoOp | 88.93±0.26 | 92.11±0.17 | +3.18±0.28 |
| ProGrad | 86.91±0.76 | 90.44±0.15 | +3.53±0.66 |
| MaPLe | 88.27±1.22 | 91.90±0.72 | +3.63±1.80 |
| PromptSRC | 90.37±0.14 | 91.94±0.40 | +1.57±0.26 |
| LoCoOp | 81.84±2.30 | 90.01±1.67 | +8.17±1.13 |

(h) FGVCAircraft.

| | Energy | MLS-E | △ |
|---|---|---|---|
| CoOp | 59.53±1.10 | 60.54±3.45 | +1.01±3.31 |
| CoCoOp | 59.21±2.05 | 58.83±2.44 | -0.38±4.13 |
| IVLP | 61.99±8.45 | 67.41±11.46 | +5.43±4.79 |
| KgCoOp | 66.06±1.05 | 65.16±1.56 | -0.90±1.01 |
| ProGrad | 57.70±2.93 | 58.64±3.91 | +0.94±1.11 |
| MaPLe | 55.95±1.39 | 64.26±3.44 | +8.31±4.31 |
| PromptSRC | 69.46±2.71 | 70.66±2.79 | +1.19±1.34 |
| LoCoOp | 57.42±3.96 | 60.08±9.93 | +2.66±9.23 |

(i) SUN397.

| | Energy | MLS-E | △ |
|---|---|---|---|
| CoOp | 76.15±0.98 | 76.74±0.56 | +0.59±0.43 |
| CoCoOp | 75.10±0.98 | 77.71±0.31 | +2.61±1.09 |
| IVLP | 76.42±1.31 | 78.87±0.72 | +2.45±1.04 |
| KgCoOp | 74.99±0.39 | 76.35±0.54 | +1.36±0.16 |
| ProGrad | 74.40±1.10 | 76.22±1.36 | +1.81±0.97 |
| MaPLe | 77.45±0.88 | 79.31±0.52 | +1.85±1.06 |
| PromptSRC | 77.43±0.33 | 79.65±0.42 | +2.22±0.71 |
| LoCoOp | 70.75±0.99 | 76.45±0.65 | +5.70±0.61 |

(j) DTD.

| | Energy | MLS-E | △ |
|---|---|---|---|
| CoOp | 67.82±0.29 | 69.37±0.74 | +1.55±0.74 |
| CoCoOp | 62.92±1.15 | 65.30±0.10 | +2.38±1.06 |
| IVLP | 63.93±0.52 | 66.58±1.12 | +2.64±0.82 |
| KgCoOp | 60.68±0.24 | 64.77±0.88 | +4.09±0.65 |
| ProGrad | 61.17±1.47 | 65.56±1.81 | +4.39±1.02 |
| MaPLe | 63.47±0.98 | 65.11±0.45 | +1.64±1.15 |
| PromptSRC | 66.75±1.28 | 69.09±0.80 | +2.34±0.87 |
| LoCoOp | 66.84±1.42 | 68.70±0.99 | +1.86±0.66 |

(k) EuroSAT.

| | Energy | MLS-E | △ |
|---|---|---|---|
| CoOp | 67.49±3.02 | 66.17±3.58 | -1.31±1.75 |
| CoCoOp | 67.69±1.96 | 68.18±2.08 | +0.49±0.72 |
| IVLP | 58.42±8.07 | 64.08±10.84 | +5.66±3.42 |
| KgCoOp | 62.82±0.26 | 67.44±1.00 | +4.62±0.76 |
| ProGrad | 74.08±0.89 | 74.19±2.60 | +0.12±1.74 |
| MaPLe | 68.42±5.44 | 69.97±4.80 | +1.55±0.89 |
| PromptSRC | 77.36±0.86 | 78.11±1.36 | +0.75±0.56 |
| LoCoOp | 69.77±4.99 | 69.54±5.02 | -0.23±0.76 |

(l) UCF101.

| | Energy | MLS-E | △ |
|---|---|---|---|
| CoOp | 80.10±1.53 | 81.59±1.90 | +1.49±1.01 |
| CoCoOp | 79.68±1.26 | 83.47±1.38 | +3.79±2.20 |
| IVLP | 79.58±1.87 | 84.16±1.72 | +4.58±0.93 |
| KgCoOp | 78.52±0.23 | 82.86±0.30 | +4.34±0.37 |
| ProGrad | 79.12±1.21 | 81.29±0.23 | +2.17±1.39 |
| MaPLe | 79.79±0.36 | 85.53±1.24 | +5.74±1.36 |
| PromptSRC | 82.62±0.65 | 85.95±1.04 | +3.33±1.37 |
| LoCoOp | 75.77±1.05 | 81.68±2.47 | +5.91±1.60 |

(m) CIFAR10.

| | Energy | MLS-E | △ |
|---|---|---|---|
| CoOp | 87.12±2.56 | 87.29±2.73 | +0.17±0.24 |
| CoCoOp | 92.67±0.76 | 93.13±0.69 | +0.45±0.17 |
| IVLP | 91.92±1.84 | 92.81±1.31 | +0.89±0.54 |
| KgCoOp | 92.78±0.29 | 93.39±0.39 | +0.61±0.11 |
| ProGrad | 89.36±0.43 | 90.10±1.12 | +0.75±0.76 |
| MaPLe | 87.53±6.17 | 88.19±4.69 | +0.66±1.82 |
| PromptSRC | 92.74±1.69 | 92.86±1.83 | +0.13±0.16 |
| LoCoOp | 90.93±2.17 | 87.70±2.18 | -3.23±1.33 |

(n) CIFAR100.

| | Energy | MLS-E | △ |
|---|---|---|---|
| CoOp | 74.64±1.74 | 77.60±2.69 | +2.96±1.48 |
| CoCoOp | 77.19±1.06 | 81.91±1.30 | +4.72±1.41 |
| IVLP | 85.72±0.61 | 88.55±0.76 | +2.83±0.29 |
| KgCoOp | 77.99±0.45 | 81.21±0.38 | +3.22±0.28 |
| ProGrad | 74.28±0.12 | 77.97±1.46 | +3.69±1.52 |
| MaPLe | 85.28±0.74 | 88.79±0.58 | +3.51±0.33 |
| PromptSRC | 87.04±0.65 | 89.17±1.09 | +2.13±0.94 |
| LoCoOp | 73.01±1.23 | 73.71±2.24 | +0.71±1.11 |

Table 17: Near OOD AUROC (↑) of prompt learning models over 13 datasets using the Energy score and MLS-E with 2-shots.

(a) Average over 13 datasets.

|  | Energy | MLS-E | △ |
|---|---|---|---|
| CoOp | 79.88±10.72 | 81.13±10.73 | +1.25±1.73 |
| CoCoOp | 80.48±11.52 | 82.25±11.63 | +1.77±1.74 |
| IVLP | 79.75±11.35 | 83.11±10.12 | +3.35±2.75 |
| KgCoOp | 80.47±10.84 | 83.35±10.44 | +2.88±2.41 |
| ProGrad | 78.86±11.41 | 81.87±11.07 | +3.02±3.25 |
| MaPLe | 80.20±10.89 | 83.68±10.41 | +3.48±2.64 |
| PromptSRC | 82.99±9.69 | 85.12±9.76 | +2.14±1.75 |
| LoCoOp | 75.13±12.35 | 80.57±10.96 | +5.44±5.58 |

(b) ImageNet.

|  | Energy | MLS-E | △ |
|---|---|---|---|
| CoOp | 94.23±0.46 | 94.56±0.19 | +0.33±0.27 |
| CoCoOp | 94.20±0.38 | 94.73±0.29 | +0.53±0.10 |
| IVLP | 94.80±0.70 | 94.90±0.23 | +0.10±0.55 |
| KgCoOp | 94.01±0.18 | 94.58±0.28 | +0.56±0.10 |
| ProGrad | 93.00±0.30 | 94.86±0.24 | +1.87±0.32 |
| MaPLe | 94.34±0.81 | 94.63±0.23 | +0.29±0.62 |
| PromptSRC | 94.46±0.16 | 95.25±0.18 | +0.79±0.23 |
| LoCoOp | 92.01±0.72 | 94.18±0.17 | +2.17±0.57 |

(c) Caltech101.

|  | Energy | MLS-E | △ |
|---|---|---|---|
| CoOp | 87.05±1.86 | 89.19±1.14 | +2.14±1.37 |
| CoCoOp | 83.71±0.91 | 87.54±1.14 | +3.83±0.28 |
| IVLP | 82.89±2.19 | 87.62±2.42 | +4.73±2.85 |
| KgCoOp | 80.43±0.39 | 88.42±0.39 | +7.99±0.12 |
| ProGrad | 81.49±2.78 | 89.30±2.72 | +7.81±2.94 |
| MaPLe | 84.73±1.85 | 91.47±0.64 | +6.75±1.89 |
| PromptSRC | 83.15±0.36 | 89.57±1.49 | +6.41±1.80 |
| LoCoOp | 67.65±4.62 | 84.35±2.10 | +16.70±4.18 |

(d) OxfordPets.

|  | Energy | MLS-E | △ |
|---|---|---|---|
| CoOp | 85.82±2.98 | 89.09±3.50 | +3.26±0.53 |
| CoCoOp | 86.48±1.08 | 90.84±0.82 | +4.37±0.30 |
| IVLP | 83.94±1.84 | 89.19±1.36 | +5.25±1.88 |
| KgCoOp | 89.06±0.45 | 92.42±0.59 | +3.36±0.30 |
| ProGrad | 88.59±0.72 | 89.84±0.48 | +1.25±1.12 |
| MaPLe | 85.18±1.66 | 88.66±2.71 | +3.48±3.03 |
| PromptSRC | 89.54±0.38 | 92.33±0.80 | +2.79±0.46 |
| LoCoOp | 80.35±3.33 | 88.62±1.51 | +8.27±2.97 |

(e) StanfordCars.

|  | Energy | MLS-E | △ |
|---|---|---|---|
| CoOp | 90.82±0.93 | 91.21±1.08 | +0.39±0.80 |
| CoCoOp | 93.11±0.99 | 93.01±1.08 | -0.10±0.13 |
| IVLP | 90.64±1.66 | 93.14±0.51 | +2.50±1.24 |
| KgCoOp | 92.64±0.07 | 93.24±0.37 | +0.60±0.42 |
| ProGrad | 89.08±1.43 | 91.68±1.05 | +2.60±0.73 |
| MaPLe | 90.52±1.62 | 93.65±0.26 | +3.13±1.81 |
| PromptSRC | 92.57±0.43 | 93.98±0.44 | +1.41±0.59 |
| LoCoOp | 87.21±2.59 | 92.03±1.78 | +4.82±0.97 |

(f) Flowers102.

|  | Energy | MLS-E | △ |
|---|---|---|---|
| CoOp | 89.39±1.03 | 91.05±0.92 | +1.65±0.34 |
| CoCoOp | 86.69±0.68 | 89.06±0.95 | +2.37±1.43 |
| IVLP | 80.13±1.23 | 84.73±0.67 | +4.60±0.56 |
| KgCoOp | 84.67±0.55 | 90.18±0.38 | +5.51±0.23 |
| ProGrad | 86.54±1.70 | 90.38±0.68 | +3.84±1.91 |
| MaPLe | 82.72±0.76 | 87.12±0.21 | +4.40±0.68 |
| PromptSRC | 87.93±0.38 | 90.10±0.45 | +2.17±0.27 |
| LoCoOp | 83.93±1.67 | 86.67±0.85 | +2.74±1.21 |

(g) Food101

|  | Energy | MLS-E | △ |
|---|---|---|---|
| CoOp | 84.00±0.44 | 85.45±0.95 | +1.45±0.68 |
| CoCoOp | 89.85±1.08 | 91.07±1.15 | +1.23±0.39 |
| IVLP | 89.78±0.22 | 90.94±0.57 | +1.16±0.77 |
| KgCoOp | 89.16±0.42 | 91.76±0.44 | +2.60±0.10 |
| ProGrad | 86.64±2.46 | 89.99±0.49 | +3.35±1.99 |
| MaPLe | 85.91±0.46 | 91.15±0.30 | +5.24±0.73 |
| PromptSRC | 90.31±0.51 | 91.53±0.62 | +1.22±0.24 |
| LoCoOp | 81.81±0.50 | 89.49±0.27 | +7.69±0.76 |

(h) FGVCAircraft.

|  | Energy | MLS-E | △ |
|---|---|---|---|
| CoOp | 58.16±0.90 | 60.28±3.37 | +2.12±3.84 |
| CoCoOp | 57.39±1.46 | 58.44±2.68 | +1.05±2.38 |
| IVLP | 61.97±4.29 | 69.19±2.28 | +7.22±2.16 |
| KgCoOp | 68.05±1.85 | 67.95±1.71 | -0.10±0.39 |
| ProGrad | 55.95±1.16 | 59.68±6.81 | +3.73±6.80 |
| MaPLe | 61.81±0.97 | 67.97±3.44 | +6.16±3.07 |
| PromptSRC | 67.39±3.22 | 69.02±3.91 | +1.64±1.99 |
| LoCoOp | 50.10±1.99 | 61.06±6.93 | +10.97±6.16 |

(i) SUN397.

|  | Energy | MLS-E | △ |
|---|---|---|---|
| CoOp | 73.46±0.35 | 74.67±0.70 | +1.21±0.61 |
| CoCoOp | 74.54±0.61 | 76.14±0.22 | +1.60±0.53 |
| IVLP | 76.05±0.70 | 78.27±0.96 | +2.22±0.84 |
| KgCoOp | 74.45±0.16 | 75.87±0.63 | +1.41±0.74 |
| ProGrad | 73.52±0.79 | 74.90±0.47 | +1.37±0.43 |
| MaPLe | 75.84±0.05 | 78.26±0.67 | +2.42±0.71 |
| PromptSRC | 76.71±0.13 | 79.41±0.36 | +2.70±0.36 |
| LoCoOp | 70.28±0.64 | 75.49±0.98 | +5.21±0.38 |

(j) DTD.

|  | Energy | MLS-E | △ |
|---|---|---|---|
| CoOp | 66.18±1.37 | 67.09±1.37 | +0.91±0.55 |
| CoCoOp | 64.06±0.20 | 65.47±1.14 | +1.41±1.11 |
| IVLP | 63.06±1.22 | 66.37±0.81 | +3.31±1.26 |
| KgCoOp | 61.72±0.30 | 65.84±1.29 | +4.12±1.44 |
| ProGrad | 61.17±0.96 | 65.97±0.32 | +4.80±0.67 |
| MaPLe | 62.53±1.74 | 65.49±0.73 | +2.96±1.30 |
| PromptSRC | 66.17±0.12 | 67.04±0.12 | +0.87±0.17 |
| LoCoOp | 62.43±0.38 | 64.85±1.41 | +2.42±1.03 |

(k) EuroSAT.

|  | Energy | MLS-E | △ |
|---|---|---|---|
| CoOp | 69.04±4.15 | 68.32±4.46 | -0.73±0.56 |
| CoCoOp | 68.27±5.42 | 68.24±5.32 | -0.04±0.14 |
| IVLP | 60.75±1.27 | 65.01±3.98 | +4.27±4.75 |
| KgCoOp | 62.98±0.32 | 67.79±1.51 | +4.81±1.81 |
| ProGrad | 67.28±5.69 | 69.20±4.12 | +1.92±2.58 |
| MaPLe | 63.82±4.56 | 67.53±7.31 | +3.71±2.94 |
| PromptSRC | 70.07±0.67 | 71.91±2.06 | +1.84±1.73 |
| LoCoOp | 62.99±2.58 | 67.56±5.05 | +4.57±2.96 |

(l) UCF101.

|  | Energy | MLS-E | △ |
|---|---|---|---|
| CoOp | 81.62±1.38 | 82.97±0.68 | +1.34±0.79 |
| CoCoOp | 78.09±1.09 | 81.44±0.81 | +3.35±0.43 |
| IVLP | 76.70±0.25 | 81.43±0.43 | +4.73±0.57 |
| KgCoOp | 78.47±0.10 | 81.53±1.52 | +3.07±1.45 |
| ProGrad | 79.42±0.61 | 82.78±0.77 | +3.36±0.27 |
| MaPLe | 78.87±0.45 | 82.52±0.99 | +3.64±0.55 |
| PromptSRC | 80.85±0.86 | 83.30±0.17 | +2.45±0.73 |
| LoCoOp | 72.37±3.29 | 80.97±2.80 | +8.60±0.58 |

(m) CIFAR10.

|  | Energy | MLS-E | △ |
|---|---|---|---|
| CoOp | 86.90±0.17 | 86.61±0.59 | -0.29±0.45 |
| CoCoOp | 93.11±0.38 | 93.36±0.41 | +0.24±0.08 |
| IVLP | 92.90±0.41 | 92.94±0.13 | +0.04±0.53 |
| KgCoOp | 93.17±0.07 | 93.54±0.06 | +0.37±0.05 |
| ProGrad | 88.24±0.35 | 87.03±1.63 | -1.21±1.41 |
| MaPLe | 93.02±0.44 | 92.72±0.42 | -0.30±0.83 |
| PromptSRC | 93.88±0.11 | 94.33±0.28 | +0.45±0.19 |
| LoCoOp | 92.70±0.80 | 88.96±1.85 | -3.73±2.64 |

(n) CIFAR100.

|  | Energy | MLS-E | △ |
|---|---|---|---|
| CoOp | 71.71±1.18 | 74.21±0.75 | +2.50±1.84 |
| CoCoOp | 76.71±0.95 | 79.91±2.15 | +3.20±1.41 |
| IVLP | 83.15±1.72 | 86.63±1.02 | +3.48±0.71 |
| KgCoOp | 77.28±0.43 | 80.48±0.31 | +3.20±0.50 |
| ProGrad | 74.20±1.65 | 78.77±1.67 | +4.56±2.70 |
| MaPLe | 83.27±1.28 | 86.61±0.90 | +3.34±2.09 |
| PromptSRC | 85.83±0.39 | 88.86±0.45 | +3.03±0.19 |
| LoCoOp | 72.81±1.34 | 73.16±1.43 | +0.36±0.82 |

Table 18: Near OOD AUROC (↑) of prompt learning models over 13 datasets using the Energy score and MLS-E with 1-shot.

(a) Average over 13 datasets.

|  | Energy | MLS-E | △ |
|---|---|---|---|
| CoOp | 77.68±10.89 | 78.75±11.51 | +1.08±2.34 |
| CoCoOp | 79.42±12.18 | 81.51±11.59 | +2.09±2.95 |
| IVLP | 79.62±10.43 | 82.58±10.29 | +2.96±3.38 |
| KgCoOp | 79.88±11.50 | 82.26±11.74 | +2.38±3.14 |
| ProGrad | 77.86±12.21 | 80.27±11.67 | +2.41±3.09 |
| MaPLe | 78.46±13.27 | 81.19±13.11 | +2.73±5.06 |
| PromptSRC | 81.76±10.60 | 83.23±11.69 | +1.47±3.32 |
| LoCoOp | 75.17±11.73 | 79.38±10.84 | +4.22±6.61 |

(b) ImageNet.

|  | Energy | MLS-E | △ |
|---|---|---|---|
| CoOp | 93.63±0.19 | 93.42±0.82 | -0.21±1.00 |
| CoCoOp | 94.72±0.50 | 94.62±0.52 | -0.10±0.96 |
| IVLP | 94.02±0.58 | 94.86±0.50 | +0.84±0.99 |
| KgCoOp | 93.88±0.11 | 94.41±0.30 | +0.53±0.25 |
| ProGrad | 92.87±0.67 | 94.08±0.47 | +1.21±0.21 |
| MaPLe | 94.00±0.71 | 94.53±0.44 | +0.53±0.39 |
| PromptSRC | 94.17±0.49 | 95.21±0.32 | +1.04±0.17 |
| LoCoOp | 91.88±0.55 | 94.21±0.21 | +2.33±0.43 |

(c) Caltech101.

|  | Energy | MLS-E | △ |
|---|---|---|---|
| CoOp | 82.95±2.81 | 86.02±1.74 | +3.07±1.81 |
| CoCoOp | 81.84±1.15 | 87.99±1.30 | +6.15±2.32 |
| IVLP | 81.47±3.02 | 85.94±2.46 | +4.47±2.40 |
| KgCoOp | 79.90±0.14 | 88.99±0.16 | +9.09±0.30 |
| ProGrad | 78.15±0.45 | 85.03±3.93 | +6.88±3.49 |
| MaPLe | 82.01±2.96 | 88.55±1.13 | +6.53±3.62 |
| PromptSRC | 81.94±0.44 | 89.58±0.54 | +7.65±0.98 |
| LoCoOp | 70.62±2.79 | 84.24±2.65 | +13.62±1.42 |

(d) OxfordPets.

|  | Energy | MLS-E | △ |
|---|---|---|---|
| CoOp | 85.22±1.81 | 89.19±2.02 | +3.97±1.71 |
| CoCoOp | 88.24±1.31 | 90.13±0.36 | +1.89±1.23 |
| IVLP | 87.79±1.80 | 90.55±0.14 | +2.77±1.66 |
| KgCoOp | 88.80±0.21 | 92.32±1.21 | +3.52±1.04 |
| ProGrad | 88.13±2.30 | 90.92±1.65 | +2.79±0.89 |
| MaPLe | 84.55±5.20 | 87.52±2.77 | +2.97±2.43 |
| PromptSRC | 89.65±0.69 | 91.97±0.56 | +2.32±1.25 |
| LoCoOp | 82.36±2.92 | 87.42±2.17 | +5.06±4.41 |

(e) StanfordCars.

|  | Energy | MLS-E | △ |
|---|---|---|---|
| CoOp | 89.17±1.70 | 89.18±1.69 | +0.01±0.54 |
| CoCoOp | 91.99±0.73 | 92.58±0.37 | +0.59±0.76 |
| IVLP | 89.48±0.55 | 92.92±1.53 | +3.43±0.99 |
| KgCoOp | 92.69±0.10 | 93.24±0.17 | +0.55±0.12 |
| ProGrad | 89.62±2.18 | 91.26±0.93 | +1.63±1.28 |
| MaPLe | 91.06±0.93 | 92.08±0.24 | +1.02±0.69 |
| PromptSRC | 92.52±0.47 | 94.18±0.30 | +1.66±0.29 |
| LoCoOp | 88.34±2.38 | 92.03±0.27 | +3.69±2.48 |

(f) Flowers102.

|  | Energy | MLS-E | △ |
|---|---|---|---|
| CoOp | 84.70±2.03 | 87.21±2.66 | +2.51±0.84 |
| CoCoOp | 85.61±0.63 | 88.20±0.53 | +2.59±0.11 |
| IVLP | 80.49±3.74 | 85.33±1.89 | +4.84±2.03 |
| KgCoOp | 84.69±0.36 | 88.76±1.03 | +4.07±1.05 |
| ProGrad | 86.98±0.75 | 88.34±1.07 | +1.36±0.56 |
| MaPLe | 81.17±2.31 | 84.68±3.01 | +3.51±0.71 |
| PromptSRC | 85.61±0.92 | 88.09±1.66 | +2.48±0.75 |
| LoCoOp | 79.39±1.42 | 84.59±2.18 | +5.20±1.08 |

(g) Food101

|  | Energy | MLS-E | △ |
|---|---|---|---|
| CoOp | 85.62±1.78 | 87.58±2.38 | +1.96±0.62 |
| CoCoOp | 89.73±0.89 | 90.73±0.24 | +0.99±0.81 |
| IVLP | 88.76±2.17 | 90.89±0.91 | +2.14±1.39 |
| KgCoOp | 89.35±0.21 | 91.33±0.65 | +1.97±0.70 |
| ProGrad | 87.67±1.79 | 90.64±1.43 | +2.96±1.31 |
| MaPLe | 87.72±0.98 | 90.56±0.84 | +2.84±0.69 |
| PromptSRC | 90.21±0.12 | 91.58±0.12 | +1.37±0.14 |
| LoCoOp | 81.94±3.22 | 88.04±1.50 | +6.10±3.09 |

(h) FGVCAircraft.

|  | Energy | MLS-E | △ |
|---|---|---|---|
| CoOp | 59.73±2.43 | 59.49±5.99 | -0.25±4.21 |
| CoCoOp | 57.01±2.68 | 64.83±3.99 | +7.81±3.17 |
| IVLP | 64.42±1.74 | 70.97±5.02 | +6.54±3.98 |
| KgCoOp | 64.85±1.17 | 65.58±3.35 | +0.74±2.21 |
| ProGrad | 55.12±1.86 | 61.51±2.60 | +6.40±2.98 |
| MaPLe | 52.24±9.93 | 57.00±5.71 | +4.76±14.41 |
| PromptSRC | 64.39±2.64 | 61.96±5.31 | -2.43±6.59 |
| LoCoOp | 51.32±7.65 | 64.50±3.27 | +13.18±8.77 |

(i) SUN397.

|  | Energy | MLS-E | △ |
|---|---|---|---|
| CoOp | 72.41±1.26 | 73.47±2.07 | +1.06±0.86 |
| CoCoOp | 74.11±0.24 | 75.57±0.64 | +1.46±0.40 |
| IVLP | 74.16±0.73 | 77.99±0.88 | +3.83±1.60 |
| KgCoOp | 73.65±0.13 | 75.85±0.15 | +2.20±0.27 |
| ProGrad | 72.12±0.57 | 75.09±0.67 | +2.97±0.19 |
| MaPLe | 74.47±1.59 | 77.94±1.17 | +3.47±0.48 |
| PromptSRC | 75.98±1.06 | 78.35±0.77 | +2.37±0.35 |
| LoCoOp | 69.97±0.93 | 74.18±0.84 | +4.21±0.29 |

(j) DTD.

|  | Energy | MLS-E | △ |
|---|---|---|---|
| CoOp | 63.49±0.91 | 64.01±0.75 | +0.52±0.95 |
| CoCoOp | 62.98±0.54 | 64.51±1.40 | +1.52±0.98 |
| IVLP | 60.59±1.37 | 64.18±1.78 | +3.59±0.89 |
| KgCoOp | 60.76±1.29 | 66.67±0.88 | +5.91±0.46 |
| ProGrad | 60.96±1.56 | 63.81±1.94 | +2.85±0.61 |
| MaPLe | 60.16±1.09 | 63.91±0.52 | +3.75±0.92 |
| PromptSRC | 65.66±1.40 | 67.55±2.69 | +1.89±1.39 |
| LoCoOp | 63.76±0.92 | 66.10±0.60 | +2.34±0.49 |

(k) EuroSAT.

|  | Energy | MLS-E | △ |
|---|---|---|---|
| CoOp | 62.39±4.19 | 61.44±3.94 | -0.96±0.62 |
| CoCoOp | 61.28±3.47 | 59.37±3.06 | -1.91±1.79 |
| IVLP | 66.60±3.49 | 63.96±5.02 | -2.64±4.86 |
| KgCoOp | 61.21±0.83 | 58.57±2.99 | -2.63±2.20 |
| ProGrad | 61.96±1.16 | 61.51±4.02 | -0.45±5.17 |
| MaPLe | 61.26±6.96 | 57.71±8.88 | -3.54±3.44 |
| PromptSRC | 65.55±4.74 | 62.45±2.70 | -3.10±2.07 |
| LoCoOp | 63.35±4.19 | 62.14±6.91 | -1.21±3.25 |

(l) UCF101.

|  | Energy | MLS-E | △ |
|---|---|---|---|
| CoOp | 79.25±1.33 | 80.57±0.71 | +1.32±0.81 |
| CoCoOp | 78.08±1.14 | 81.23±0.28 | +3.15±1.41 |
| IVLP | 77.24±1.68 | 82.06±0.86 | +4.82±2.30 |
| KgCoOp | 78.88±0.97 | 82.21±1.98 | +3.33±1.06 |
| ProGrad | 80.03±1.83 | 81.16±1.73 | +1.14±0.19 |
| MaPLe | 76.88±0.38 | 81.71±0.46 | +4.84±0.53 |
| PromptSRC | 79.62±0.96 | 82.63±1.13 | +3.01±0.21 |
| LoCoOp | 71.54±1.89 | 78.06±2.52 | +6.52±0.63 |

(m) CIFAR10.

|  | Energy | MLS-E | △ |
|---|---|---|---|
| CoOp | 83.06±1.34 | 81.77±0.90 | -1.29±1.60 |
| CoCoOp | 90.85±2.48 | 90.01±2.16 | -0.84±0.59 |
| IVLP | 87.01±6.32 | 86.81±3.85 | -0.20±3.51 |
| KgCoOp | 93.01±0.38 | 91.78±1.72 | -1.23±1.61 |
| ProGrad | 87.62±4.43 | 86.07±4.47 | -1.55±2.03 |
| MaPLe | 92.41±1.38 | 92.43±2.35 | +0.02±1.21 |
| PromptSRC | 93.62±0.39 | 92.59±2.13 | -1.02±1.78 |
| LoCoOp | 89.29±1.17 | 82.42±10.55 | -6.87±9.46 |

(n) CIFAR100.

|  | Energy | MLS-E | △ |
|---|---|---|---|
| CoOp | 68.18±1.23 | 70.43±3.00 | +2.24±2.90 |
| CoCoOp | 76.01±1.44 | 79.86±1.91 | +3.85±0.47 |
| IVLP | 83.00±0.54 | 87.04±0.30 | +4.04±0.72 |
| KgCoOp | 76.77±0.66 | 79.69±1.55 | +2.92±0.91 |
| ProGrad | 70.90±2.52 | 74.08±1.77 | +3.18±1.24 |
| MaPLe | 82.03±2.21 | 86.88±0.88 | +4.85±2.34 |
| PromptSRC | 83.99±0.11 | 85.91±0.89 | +1.92±0.82 |
| LoCoOp | 73.40±1.57 | 74.07±1.22 | +0.67±0.54 |

### A.3.3 MCM SCORE

We show the effectiveness of our method with MCM measured by average AUROC and FPR95 across 13 datasets in Table 19 and Table 20, where $S_{\text{MLS-MCM}} = S_{\text{MCM}} - \beta \cdot S_{\text{Context}}$. Also, we compare the correlation between scores and the Context score using MaxLogit and MCM in Figure 6.

Table 19: Near OOD AUROC (↑) of prompt learning models over 13 datasets using the MCM score and MLS-MCM.

(a) Average over 13 datasets.

|  | MCM | MLS-MCM | △ |
|---|---|---|---|
| CoOp | 79.41 | 81.31 | +1.90 |
| CoCoOp | 79.28 | 82.59 | +3.31 |
| IVLP | 80.95 | 84.48 | +3.53 |
| KgCoOp | 79.82 | 83.31 | +3.49 |
| ProGrad | 79.91 | 81.81 | +1.90 |
| MaPLe | 80.54 | 83.81 | +3.27 |
| PromptSRC | 82.34 | 85.49 | +3.15 |
| LoCoOp | 79.22 | 80.36 | +1.14 |

(b) ImageNet.

|  | MCM | MLS-MCM | △ |
|---|---|---|---|
| CoOp | 93.48 | 94.48 | +0.99 |
| CoCoOp | 94.16 | 94.93 | +0.77 |
| IVLP | 93.68 | 94.18 | +0.50 |
| KgCoOp | 93.84 | 94.30 | +0.46 |
| ProGrad | 93.20 | 94.08 | +0.88 |
| MaPLe | 93.42 | 93.97 | +0.56 |
| PromptSRC | 94.46 | 95.30 | +0.84 |
| LoCoOp | 93.23 | 93.81 | +0.59 |

(c) Caltech101.

|  | MCM | MLS-MCM | △ |
|---|---|---|---|
| CoOp | 92.57 | 92.42 | -0.15 |
| CoCoOp | 90.71 | 90.27 | -0.44 |
| IVLP | 91.35 | 91.56 | +0.21 |
| KgCoOp | 91.08 | 90.83 | -0.26 |
| ProGrad | 91.16 | 91.15 | -0.01 |
| MaPLe | 91.32 | 91.92 | +0.61 |
| PromptSRC | 91.27 | 90.93 | -0.34 |
| LoCoOp | 91.02 | 90.97 | -0.04 |

(d) OxfordPets.

|  | MCM | MLS-MCM | △ |
|---|---|---|---|
| CoOp | 85.75 | 88.59 | +2.84 |
| CoCoOp | 86.85 | 91.41 | +4.56 |
| IVLP | 87.51 | 91.01 | +3.50 |
| KgCoOp | 88.35 | 92.51 | +4.16 |
| ProGrad | 86.86 | 88.12 | +1.26 |
| MaPLe | 86.31 | 90.39 | +4.08 |
| PromptSRC | 88.69 | 93.24 | +4.55 |
| LoCoOp | 86.56 | 87.12 | +0.56 |

(e) StanfordCars.

|  | MCM | MLS-MCM | △ |
|---|---|---|---|
| CoOp | 82.61 | 88.62 | +6.01 |
| CoCoOp | 83.77 | 91.30 | +7.53 |
| IVLP | 84.75 | 92.66 | +7.91 |
| KgCoOp | 83.38 | 92.41 | +9.03 |
| ProGrad | 84.23 | 90.32 | +6.09 |
| MaPLe | 83.80 | 92.13 | +8.33 |
| PromptSRC | 84.64 | 93.85 | +9.21 |
| LoCoOp | 82.83 | 88.23 | +5.40 |

(f) Flowers102.

|  | MCM | MLS-MCM | △ |
|---|---|---|---|
| CoOp | 90.08 | 90.79 | +0.71 |
| CoCoOp | 84.56 | 88.24 | +3.69 |
| IVLP | 87.31 | 89.33 | +2.02 |
| KgCoOp | 87.00 | 91.50 | +4.50 |
| ProGrad | 89.66 | 90.80 | +1.14 |
| MaPLe | 86.07 | 88.84 | +2.77 |
| PromptSRC | 90.61 | 92.96 | +2.35 |
| LoCoOp | 86.98 | 87.05 | +0.07 |

(g) Food101.

|  | MCM | MLS-MCM | △ |
|---|---|---|---|
| CoOp | 85.65 | 87.72 | +2.07 |
| CoCoOp | 89.05 | 91.64 | +2.59 |
| IVLP | 88.71 | 92.07 | +3.36 |
| KgCoOp | 89.41 | 92.30 | +2.89 |
| ProGrad | 88.29 | 90.88 | +2.59 |
| MaPLe | 88.96 | 92.22 | +3.26 |
| PromptSRC | 89.39 | 92.45 | +3.06 |
| LoCoOp | 88.29 | 90.05 | +1.76 |

(h) FGVCAircraft.

|  | MCM | MLS-MCM | △ |
|---|---|---|---|
| CoOp | 41.22 | 52.57 | +11.35 |
| CoCoOp | 37.53 | 57.00 | +19.47 |
| IVLP | 40.70 | 65.75 | +25.05 |
| KgCoOp | 37.83 | 58.34 | +20.51 |
| ProGrad | 40.66 | 50.54 | +9.88 |
| MaPLe | 38.94 | 58.04 | +19.10 |
| PromptSRC | 39.89 | 61.58 | +21.69 |
| LoCoOp | 38.94 | 50.03 | +11.09 |

(i) SUN397.

|  | MCM | MLS-MCM | △ |
|---|---|---|---|
| CoOp | 78.86 | 78.83 | -0.03 |
| CoCoOp | 79.97 | 80.13 | +0.17 |
| IVLP | 80.41 | 80.74 | +0.33 |
| KgCoOp | 80.25 | 80.37 | +0.12 |
| ProGrad | 79.63 | 79.68 | +0.06 |
| MaPLe | 80.21 | 80.72 | +0.52 |
| PromptSRC | 81.71 | 82.03 | +0.32 |
| LoCoOp | 79.75 | 79.72 | -0.03 |

(j) DTD.

|  | MCM | MLS-MCM | △ |
|---|---|---|---|
| CoOp | 69.89 | 70.03 | +0.14 |
| CoCoOp | 66.96 | 67.41 | +0.45 |
| IVLP | 69.26 | 69.24 | -0.02 |
| KgCoOp | 68.88 | 69.11 | +0.23 |
| ProGrad | 68.21 | 68.17 | -0.04 |
| MaPLe | 68.39 | 68.62 | +0.23 |
| PromptSRC | 71.40 | 71.45 | +0.05 |
| LoCoOp | 69.76 | 69.82 | +0.06 |

(k) EuroSAT.

|  | MCM | MLS-MCM | △ |
|---|---|---|---|
| CoOp | 67.49 | 65.33 | -2.16 |
| CoCoOp | 63.63 | 63.24 | -0.39 |
| IVLP | 68.85 | 67.28 | -1.58 |
| KgCoOp | 62.17 | 62.20 | +0.03 |
| ProGrad | 66.93 | 65.91 | -1.02 |
| MaPLe | 69.72 | 67.60 | -2.13 |
| PromptSRC | 72.57 | 68.61 | -3.96 |
| LoCoOp | 63.23 | 61.57 | -1.66 |

(l) UCF101.

|  | MCM | MLS-MCM | △ |
|---|---|---|---|
| CoOp | 84.05 | 84.83 | +0.78 |
| CoCoOp | 83.95 | 84.82 | +0.86 |
| IVLP | 83.46 | 85.01 | +1.55 |
| KgCoOp | 84.55 | 85.22 | +0.67 |
| ProGrad | 85.03 | 85.40 | +0.37 |
| MaPLe | 83.34 | 84.78 | +1.44 |
| PromptSRC | 85.11 | 86.47 | +1.36 |
| LoCoOp | 83.57 | 83.77 | +0.20 |

(m) CIFAR10.

|  | MCM | MLS-MCM | △ |
|---|---|---|---|
| CoOp | 87.12 | 86.88 | -0.24 |
| CoCoOp | 92.69 | 92.76 | +0.08 |
| IVLP | 91.42 | 91.80 | +0.38 |
| KgCoOp | 93.15 | 93.25 | +0.10 |
| ProGrad | 90.09 | 89.84 | -0.25 |
| MaPLe | 92.15 | 92.31 | +0.17 |
| PromptSRC | 94.05 | 93.88 | -0.17 |
| LoCoOp | 91.44 | 87.77 | -3.68 |

(n) CIFAR100.

|  | MCM | MLS-MCM | △ |
|---|---|---|---|
| CoOp | 73.55 | 75.89 | +2.34 |
| CoCoOp | 76.76 | 80.47 | +3.71 |
| IVLP | 84.92 | 87.57 | +2.65 |
| KgCoOp | 77.74 | 80.63 | +2.89 |
| ProGrad | 74.89 | 78.60 | +3.70 |
| MaPLe | 84.42 | 88.00 | +3.58 |
| PromptSRC | 86.67 | 88.70 | +2.02 |
| LoCoOp | 74.22 | 74.77 | +0.55 |

Table 20: Near OOD FPR95 (↓) of prompt learning models over 13 datasets using the MCM score and MLS-MCM.

(a) Average over 13 datasets.

|  | MCM | MLS-MCM | △ |
|---|---|---|---|
| CoOp | 63.70 | 57.48 | -6.21 |
| CoCoOp | 63.76 | 53.62 | -10.14 |
| IVLP | 60.05 | 49.23 | -10.83 |
| KgCoOp | 62.20 | 51.85 | -10.35 |
| ProGrad | 62.89 | 56.22 | -6.67 |
| MaPLe | 60.68 | 49.00 | -11.68 |
| PromptSRC | 55.86 | 46.13 | -9.73 |
| LoCoOp | 63.37 | 60.63 | -2.74 |

(b) ImageNet.

|  | MCM | MLS-MCM | △ |
|---|---|---|---|
| CoOp | 32.74 | 26.85 | -5.89 |
| CoCoOp | 30.51 | 24.63 | -5.89 |
| IVLP | 31.20 | 26.31 | -4.89 |
| KgCoOp | 31.99 | 27.10 | -4.89 |
| ProGrad | 34.16 | 29.29 | -4.88 |
| MaPLe | 32.58 | 27.27 | -5.31 |
| PromptSRC | 28.77 | 22.81 | -5.97 |
| LoCoOp | 35.06 | 31.97 | -3.09 |

(c) Caltech101.

|  | MCM | MLS-MCM | △ |
|---|---|---|---|
| CoOp | 24.16 | 24.43 | +0.27 |
| CoCoOp | 31.48 | 31.77 | +0.28 |
| IVLP | 31.33 | 29.16 | -2.17 |
| KgCoOp | 27.22 | 27.34 | +0.12 |
| ProGrad | 33.36 | 33.33 | -0.04 |
| MaPLe | 32.98 | 27.50 | -5.48 |
| PromptSRC | 26.36 | 25.61 | -0.75 |
| LoCoOp | 30.81 | 30.76 | -0.05 |

(d) OxfordPets.

|  | MCM | MLS-MCM | △ |
|---|---|---|---|
| CoOp | 55.23 | 46.74 | -8.49 |
| CoCoOp | 53.26 | 40.67 | -12.59 |
| IVLP | 50.43 | 43.36 | -7.08 |
| KgCoOp | 51.32 | 36.12 | -15.20 |
| ProGrad | 54.48 | 55.72 | +1.25 |
| MaPLe | 52.24 | 42.94 | -9.30 |
| PromptSRC | 49.69 | 38.07 | -11.62 |
| LoCoOp | 53.91 | 53.14 | -0.77 |

(e) StanfordCars.

|  | MCM | MLS-MCM | △ |
|---|---|---|---|
| CoOp | 64.30 | 41.26 | -23.05 |
| CoCoOp | 63.84 | 34.19 | -29.65 |
| IVLP | 59.38 | 27.69 | -31.69 |
| KgCoOp | 64.69 | 31.72 | -32.97 |
| ProGrad | 62.17 | 37.12 | -25.06 |
| MaPLe | 62.20 | 28.56 | -33.64 |
| PromptSRC | 59.71 | 24.71 | -35.01 |
| LoCoOp | 64.56 | 46.73 | -17.83 |

(f) Flowers102.

|  | MCM | MLS-MCM | △ |
|---|---|---|---|
| CoOp | 45.66 | 42.85 | -2.81 |
| CoCoOp | 62.92 | 52.14 | -10.79 |
| IVLP | 55.70 | 48.98 | -6.72 |
| KgCoOp | 57.90 | 40.86 | -17.04 |
| ProGrad | 46.41 | 42.20 | -4.22 |
| MaPLe | 57.83 | 49.43 | -8.40 |
| PromptSRC | 43.15 | 34.45 | -8.70 |
| LoCoOp | 54.17 | 53.85 | -0.32 |

(g) Food101

|  | MCM | MLS-MCM | △ |
|---|---|---|---|
| CoOp | 64.07 | 53.69 | -10.38 |
| CoCoOp | 55.24 | 39.39 | -15.86 |
| IVLP | 56.66 | 35.87 | -20.79 |
| KgCoOp | 54.18 | 36.14 | -18.04 |
| ProGrad | 58.86 | 43.43 | -15.43 |
| MaPLe | 54.93 | 35.68 | -19.24 |
| PromptSRC | 54.50 | 35.58 | -18.91 |
| LoCoOp | 56.78 | 45.80 | -10.97 |

(h) FGVCAircraft.

|  | MCM | MLS-MCM | △ |
|---|---|---|---|
| CoOp | 96.60 | 87.13 | -9.47 |
| CoCoOp | 97.73 | 79.83 | -17.90 |
| IVLP | 96.33 | 73.45 | -22.89 |
| KgCoOp | 97.50 | 79.50 | -18.00 |
| ProGrad | 96.26 | 86.77 | -9.49 |
| MaPLe | 97.15 | 79.37 | -17.78 |
| PromptSRC | 96.76 | 75.93 | -20.83 |
| LoCoOp | 97.11 | 89.01 | -8.10 |

(i) SUN397.

|  | MCM | MLS-MCM | △ |
|---|---|---|---|
| CoOp | 67.98 | 68.17 | +0.18 |
| CoCoOp | 65.10 | 64.27 | -0.83 |
| IVLP | 65.20 | 63.57 | -1.62 |
| KgCoOp | 64.31 | 63.76 | -0.55 |
| ProGrad | 66.65 | 66.43 | -0.23 |
| MaPLe | 65.25 | 63.02 | -2.24 |
| PromptSRC | 62.12 | 60.91 | -1.21 |
| LoCoOp | 65.51 | 65.60 | +0.09 |

(j) DTD.

|  | MCM | MLS-MCM | △ |
|---|---|---|---|
| CoOp | 87.99 | 86.66 | -1.33 |
| CoCoOp | 90.93 | 89.61 | -1.32 |
| IVLP | 88.19 | 87.42 | -0.77 |
| KgCoOp | 88.74 | 87.37 | -1.37 |
| ProGrad | 89.19 | 88.90 | -0.29 |
| MaPLe | 89.07 | 87.79 | -1.28 |
| PromptSRC | 83.29 | 83.18 | -0.11 |
| LoCoOp | 87.82 | 86.76 | -1.06 |

(k) EuroSAT.

|  | MCM | MLS-MCM | △ |
|---|---|---|---|
| CoOp | 87.62 | 86.12 | -1.49 |
| CoCoOp | 90.61 | 86.67 | -3.94 |
| IVLP | 82.81 | 80.20 | -2.60 |
| KgCoOp | 89.72 | 87.54 | -2.18 |
| ProGrad | 86.46 | 84.96 | -1.50 |
| MaPLe | 77.79 | 74.61 | -3.18 |
| PromptSRC | 78.29 | 82.00 | +3.71 |
| LoCoOp | 90.22 | 87.44 | -2.78 |

(l) UCF101.

|  | MCM | MLS-MCM | △ |
|---|---|---|---|
| CoOp | 61.80 | 54.14 | -7.65 |
| CoCoOp | 63.33 | 50.41 | -12.93 |
| IVLP | 59.62 | 47.81 | -11.81 |
| KgCoOp | 58.62 | 50.47 | -8.14 |
| ProGrad | 59.14 | 51.74 | -7.41 |
| MaPLe | 63.36 | 49.15 | -14.21 |
| PromptSRC | 56.65 | 47.04 | -9.62 |
| LoCoOp | 61.10 | 57.06 | -4.04 |

(m) CIFAR10.

|  | MCM | MLS-MCM | △ |
|---|---|---|---|
| CoOp | 53.53 | 52.71 | -0.83 |
| CoCoOp | 37.02 | 31.98 | -5.04 |
| IVLP | 38.67 | 31.04 | -7.63 |
| KgCoOp | 35.12 | 32.46 | -2.66 |
| ProGrad | 45.00 | 42.88 | -2.12 |
| MaPLe | 36.95 | 29.19 | -7.76 |
| PromptSRC | 27.73 | 23.03 | -4.70 |
| LoCoOp | 37.78 | 54.39 | +16.62 |

(n) CIFAR100.

|  | MCM | MLS-MCM | △ |
|---|---|---|---|
| CoOp | 86.40 | 76.54 | -9.86 |
| CoCoOp | 86.82 | 71.48 | -15.35 |
| IVLP | 65.16 | 45.08 | -20.08 |
| KgCoOp | 87.26 | 73.61 | -13.65 |
| ProGrad | 85.42 | 68.06 | -17.37 |
| MaPLe | 66.44 | 42.43 | -24.00 |
| PromptSRC | 59.12 | 46.33 | -12.79 |
| LoCoOp | 88.94 | 85.63 | -3.31 |

Figure 6: Comparison of MaxLogit (left) or MCM score (right) vs. Context score for near OOD samples with IVLP Khattak et al. (2023a) on EuroSAT Helber et al. (2019). MaxLogit score shows positive correlation with the Context score while MCM score lacks the correlation.

### A.3.4 SCORE COMPARISON

In the main section, average AUROC and FPR95 of different scores are compared in Table 3. Here, we provide the individual's dataset results averaged across 16, 8, 4, 2, and 1-shot with 3 random seeds in Table 21 and Table 22.

Table 21: Near OOD AUROC (↑) of prompt learning models across 13 datasets using the MaxLogit score, the Energy score, MLS-M, MLS-E, and MCM.

(a) Average over 13 datasets.

|  | MaxLogit | MLS-M | Energy | MLS-E | MCM |
|---|---|---|---|---|---|
| CoOp | 80.74 | **81.84** | 80.44 | 81.71 | 79.41 |
| CoCoOp | 81.09 | **82.74** | 80.53 | 82.74 | 79.28 |
| IVLP | 81.12 | 84.34 | 80.49 | **84.40** | 80.95 |
| KgCoOp | 80.84 | 83.12 | 80.14 | **83.23** | 79.82 |
| ProGrad | 79.77 | **82.35** | 78.79 | 81.93 | 79.91 |
| MaPLe | 81.06 | 83.94 | 80.39 | **83.99** | 80.54 |
| PromptSRC | 83.85 | 85.77 | 83.48 | **85.88** | 82.34 |
| LoCoOp | 77.55 | **81.74** | 75.94 | 81.25 | 79.22 |

(b) ImageNet.

|  | MaxLogit | MLS-M | Energy | MLS-E | MCM |
|---|---|---|---|---|---|
| CoOp | 93.78 | 94.66 | 93.73 | **94.83** | 93.48 |
| CoCoOp | 94.85 | 95.14 | 94.76 | **95.30** | 94.16 |
| IVLP | 94.55 | 94.70 | 94.50 | **94.94** | 93.68 |
| KgCoOp | 94.21 | 94.05 | 94.05 | **94.41** | 93.84 |
| ProGrad | 93.62 | 94.67 | 93.46 | **94.79** | 93.20 |
| MaPLe | 94.20 | 94.35 | 94.07 | **94.58** | 93.42 |
| PromptSRC | 94.52 | 95.32 | 94.46 | **95.56** | 94.46 |
| LoCoOp | 93.10 | **94.56** | 92.25 | 94.53 | 93.23 |

(c) Caltech101.

|  | MaxLogit | MLS-M | Energy | MLS-E | MCM |
|---|---|---|---|---|---|
| CoOp | 88.27 | 90.12 | 87.31 | 89.48 | **92.57** |
| CoCoOp | 85.80 | 89.02 | 84.19 | 88.04 | **90.71** |
| IVLP | 85.50 | 90.53 | 84.08 | 89.85 | **91.35** |
| KgCoOp | 83.64 | 90.06 | 80.39 | 88.79 | **91.08** |
| ProGrad | 82.96 | 88.85 | 80.60 | 87.60 | **91.16** |
| MaPLe | 85.91 | **91.53** | 84.52 | 90.95 | 91.32 |
| PromptSRC | 84.94 | 90.56 | 82.59 | 89.70 | **91.27** |
| LoCoOp | 76.08 | 87.75 | 71.45 | 86.09 | **91.02** |

(d) OxfordPets.

|  | MaxLogit | MLS-M | Energy | MLS-E | MCM |
|---|---|---|---|---|---|
| CoOp | 86.22 | **88.73** | 85.91 | 88.62 | 85.75 |
| CoCoOp | 89.58 | **92.28** | 88.93 | 92.19 | 84.56 |
| IVLP | 88.84 | **91.94** | 88.27 | 91.67 | 87.51 |
| KgCoOp | 89.94 | **92.64** | 89.25 | 92.28 | 88.35 |
| ProGrad | 87.82 | **89.60** | 87.54 | 89.51 | 86.86 |
| MaPLe | 87.40 | **91.00** | 86.66 | 90.64 | 86.31 |
| PromptSRC | 90.80 | **93.45** | 90.38 | 93.29 | 88.69 |
| LoCoOp | 84.44 | **89.19** | 82.90 | 88.44 | 86.56 |

(e) StanfordCars.

|  | MaxLogit | MLS-M | Energy | MLS-E | MCM |
|---|---|---|---|---|---|
| CoOp | 91.36 | 91.59 | 91.37 | **91.66** | 82.61 |
| CoCoOp | 92.43 | 92.99 | 92.36 | **93.17** | 83.77 |
| IVLP | 90.43 | 92.98 | 90.45 | **93.24** | 84.75 |
| KgCoOp | 92.77 | 93.25 | 93.43 | **93.43** | 83.38 |
| ProGrad | 91.52 | **92.63** | 91.34 | 92.55 | 84.23 |
| MaPLe | 91.39 | 92.85 | 91.53 | **93.14** | 83.80 |
| PromptSRC | 92.88 | 94.24 | 92.97 | **94.47** | 84.64 |
| LoCoOp | 88.24 | 91.94 | 87.93 | **92.16** | 82.83 |

(f) Flowers102.

|  | MaxLogit | MLS-M | Energy | MLS-E | MCM |
|---|---|---|---|---|---|
| CoOp | 90.83 | **91.99** | 90.12 | 91.51 | 90.08 |
| CoCoOp | 87.93 | **89.41** | 86.76 | 88.89 | 84.56 |
| IVLP | 86.20 | **88.45** | 84.51 | 87.44 | 87.31 |
| KgCoOp | 87.61 | **91.12** | 85.78 | 90.51 | 87.00 |
| ProGrad | 89.27 | **91.41** | 87.51 | 90.37 | 89.66 |
| MaPLe | 86.05 | **88.34** | 84.45 | 87.40 | 86.07 |
| PromptSRC | 91.10 | **92.61** | 89.92 | 91.86 | 90.61 |
| LoCoOp | 86.17 | **88.59** | 83.73 | 87.14 | 86.98 |

(g) Food101

|  | MaxLogit | MLS-M | Energy | MLS-E | MCM |
|---|---|---|---|---|---|
| CoOp | 86.70 | **87.91** | 86.32 | 87.70 | 85.65 |
| CoCoOp | 90.52 | **91.63** | 90.09 | 91.44 | 89.05 |
| IVLP | 89.70 | **91.87** | 89.22 | 91.73 | 88.71 |
| KgCoOp | 89.87 | **92.12** | 89.29 | 91.96 | 89.41 |
| ProGrad | 88.60 | **91.05** | 87.93 | 90.90 | 88.29 |
| MaPLe | 89.10 | **92.00** | 88.43 | 91.84 | 88.96 |
| PromptSRC | 90.94 | **92.11** | 90.59 | 91.97 | 89.39 |
| LoCoOp | 84.87 | **90.12** | 82.51 | 89.45 | 88.29 |

(h) FGVCAircraft.

|  | MaxLogit | MLS-M | Energy | MLS-E | MCM |
|---|---|---|---|---|---|
| CoOp | 55.99 | 56.97 | 58.92 | **59.98** | 41.22 |
| CoCoOp | 52.60 | 55.04 | 57.28 | **61.61** | 37.53 |
| IVLP | 58.47 | 64.16 | 63.66 | **71.01** | 40.70 |
| KgCoOp | 57.82 | 57.46 | 66.07 | **66.68** | 37.83 |
| ProGrad | 53.69 | 55.67 | 57.15 | **59.58** | 40.66 |
| MaPLe | 52.18 | 56.93 | 57.03 | **63.37** | 38.94 |
| PromptSRC | 60.63 | 62.50 | 68.80 | **70.34** | 39.89 |
| LoCoOp | 50.99 | 56.12 | 55.31 | **61.06** | 38.94 |

(i) SUN397.

|  | MaxLogit | MLS-M | Energy | MLS-E | MCM |
|---|---|---|---|---|---|
| CoOp | 75.78 | 76.75 | 75.10 | 76.19 | **78.86** |
| CoCoOp | 76.32 | 78.29 | 75.10 | 77.36 | **79.97** |
| IVLP | 77.13 | 79.60 | 76.17 | 79.03 | **80.41** |
| KgCoOp | 76.45 | 77.91 | 74.80 | 76.54 | **80.25** |
| ProGrad | 75.52 | 77.67 | 74.00 | 76.48 | **79.63** |
| MaPLe | 77.62 | 79.73 | 76.65 | 79.09 | **80.21** |
| PromptSRC | 78.51 | 80.70 | 77.26 | 79.91 | **81.71** |
| LoCoOp | 73.97 | 78.00 | 71.27 | 76.23 | **79.75** |

(j) DTD.

|  | MaxLogit | MLS-M | Energy | MLS-E | MCM |
|---|---|---|---|---|---|
| CoOp | 68.90 | 69.60 | 68.35 | 69.16 | **69.89** |
| CoCoOp | 65.10 | **67.17** | 63.75 | 66.27 | 66.96 |
| IVLP | 64.99 | 67.93 | 63.94 | 67.25 | **69.26** |
| KgCoOp | 63.79 | 68.17 | 61.31 | 66.69 | **68.88** |
| ProGrad | 62.90 | 66.96 | 60.84 | 65.22 | **68.21** |
| MaPLe | 64.80 | 67.79 | 63.54 | 66.92 | **68.39** |
| PromptSRC | 69.09 | 70.38 | 67.56 | 69.27 | **71.40** |
| LoCoOp | 66.63 | 69.05 | 65.29 | 68.07 | **69.76** |

(k) EuroSAT.

|  | MaxLogit | MLS-M | Energy | MLS-E | MCM |
|---|---|---|---|---|---|
| CoOp | **67.94** | 67.83 | 67.54 | 67.49 | 67.49 |
| CoCoOp | **66.87** | 66.76 | 66.12 | 65.99 | 63.63 |
| IVLP | 65.56 | **70.62** | 63.89 | 69.72 | 68.85 |
| KgCoOp | 62.41 | 65.66 | 62.36 | **66.24** | 62.17 |
| ProGrad | 68.96 | **69.71** | 68.19 | 69.10 | 66.93 |
| MaPLe | 71.18 | **72.28** | 70.02 | 71.94 | 69.72 |
| PromptSRC | **75.22** | 74.97 | 74.40 | 74.30 | 72.57 |
| LoCoOp | 66.72 | **67.85** | 66.04 | 67.80 | 63.23 |

(l) UCF101.

|  | MaxLogit | MLS-M | Energy | MLS-E | MCM |
|---|---|---|---|---|---|
| CoOp | 82.17 | 83.65 | 81.31 | 82.96 | **84.05** |
| CoCoOp | 81.32 | **84.02** | 79.65 | 82.92 | 83.95 |
| IVLP | 80.26 | **84.55** | 78.96 | 83.84 | 83.46 |
| KgCoOp | 81.26 | 84.06 | 79.04 | 82.76 | **84.55** |
| ProGrad | 81.21 | 83.71 | 79.39 | 82.30 | **85.03** |
| MaPLe | 80.81 | **84.25** | 79.38 | 83.45 | 83.34 |
| PromptSRC | 83.19 | **85.43** | 81.77 | 84.56 | 85.11 |
| LoCoOp | 76.28 | 82.54 | 73.63 | 81.13 | **83.57** |

(m) CIFAR10.

|  | MaxLogit | MLS-M | Energy | MLS-E | MCM |
|---|---|---|---|---|---|
| CoOp | 84.05 | 85.76 | 82.84 | 84.86 | **87.12** |
| CoCoOp | 89.91 | 92.49 | 88.08 | 91.91 | **92.69** |
| IVLP | 88.17 | 91.10 | 86.40 | 90.40 | **91.42** |
| KgCoOp | 90.26 | 92.77 | 88.09 | 91.93 | **93.15** |
| ProGrad | 83.94 | 89.08 | 81.61 | 87.96 | **90.09** |
| MaPLe | 87.79 | 91.81 | 85.68 | 91.07 | **92.15** |
| PromptSRC | 91.36 | 93.97 | 89.76 | 93.41 | **94.05** |
| LoCoOp | 84.69 | 90.22 | 81.33 | 89.26 | **91.44** |

(n) CIFAR100.

|  | MaxLogit | MLS-M | Energy | MLS-E | MCM |
|---|---|---|---|---|---|
| CoOp | 77.67 | **78.30** | 76.84 | 77.75 | 73.55 |
| CoCoOp | 80.99 | **81.44** | 79.64 | 80.73 | 76.76 |
| IVLP | 84.69 | **88.01** | 82.35 | 87.05 | 84.92 |
| KgCoOp | 80.94 | **81.14** | 78.54 | 79.78 | 77.74 |
| ProGrad | 77.01 | **79.55** | 74.71 | 78.76 | 74.89 |
| MaPLe | 85.35 | **88.35** | 83.17 | 87.49 | 84.42 |
| PromptSRC | 86.82 | **88.78** | 84.76 | 87.83 | 86.67 |
| LoCoOp | 76.03 | **76.67** | 73.54 | 74.83 | 74.22 |

Table 22: Near OOD FPR95 (↓) of prompt learning models across 13 datasets using the MaxLogit score, the Energy score, MLS-M, MLS-E, and MCM.

(a) Average over 13 datasets.

|  | MaxLogit | MLS-M | Energy | MLS-E | MCM |
|---|---|---|---|---|---|
| CoOp | 58.23 | **54.85** | 59.45 | 55.37 | 63.70 |
| CoCoOp | 55.78 | **51.67** | 57.49 | 51.90 | 63.76 |
| IVLP | 55.65 | **48.58** | 57.66 | 49.19 | 60.05 |
| KgCoOp | 57.16 | **51.52** | 59.69 | 51.90 | 62.20 |
| ProGrad | 60.07 | **54.02** | 62.74 | 55.34 | 62.89 |
| MaPLe | 55.58 | **48.36** | 57.82 | 48.76 | 60.68 |
| PromptSRC | 49.65 | **44.60** | 51.86 | 45.15 | 55.86 |
| LoCoOp | 64.89 | **55.76** | 69.07 | 57.80 | 63.37 |

(b) ImageNet.

|  | MaxLogit | MLS-M | Energy | MLS-E | MCM |
|---|---|---|---|---|---|
| CoOp | 31.02 | **26.40** | 33.04 | 26.96 | 32.74 |
| CoCoOp | 26.76 | **23.85** | 29.16 | 24.69 | 30.51 |
| IVLP | 27.23 | **24.54** | 29.40 | 24.78 | 31.20 |
| KgCoOp | 29.84 | **27.45** | 33.09 | 28.35 | 31.99 |
| ProGrad | 32.73 | **27.40** | 35.87 | 28.41 | 34.16 |
| MaPLe | 29.87 | **26.09** | 32.34 | 26.52 | 32.58 |
| PromptSRC | 27.89 | **22.82** | 30.06 | 22.97 | 28.77 |
| LoCoOp | 36.95 | **28.94** | 45.33 | 32.02 | 35.06 |

(c) Caltech101.

|  | MaxLogit | MLS-M | Energy | MLS-E | MCM |
|---|---|---|---|---|---|
| CoOp | 38.42 | 29.90 | 42.21 | 32.07 | **24.16** |
| CoCoOp | 42.22 | 33.02 | 48.28 | 36.22 | **31.48** |
| IVLP | 45.94 | **30.97** | 51.35 | 34.09 | 31.33 |
| KgCoOp | 49.33 | 28.80 | 61.27 | 34.10 | **27.22** |
| ProGrad | 56.38 | 39.35 | 65.31 | 45.11 | **33.36** |
| MaPLe | 44.44 | **28.30** | 49.42 | 31.23 | 32.98 |
| PromptSRC | 43.79 | **25.83** | 52.12 | 28.81 | 26.36 |
| LoCoOp | 69.02 | 37.02 | 80.74 | 44.34 | **30.81** |

(d) OxfordPets.

|  | MaxLogit | MLS-M | Energy | MLS-E | MCM |
|---|---|---|---|---|---|
| CoOp | 51.60 | **45.90** | 52.87 | 46.91 | 55.23 |
| CoCoOp | 42.47 | **35.89** | 45.06 | 37.87 | 53.26 |
| IVLP | 48.28 | **40.88** | 50.67 | 43.75 | 50.43 |
| KgCoOp | 47.06 | **35.52** | 49.07 | 38.01 | 51.32 |
| ProGrad | **50.75** | 51.94 | 51.49 | 52.73 | 54.48 |
| MaPLe | 50.86 | **42.87** | 53.88 | 45.57 | 52.24 |
| PromptSRC | 42.14 | **38.08** | 44.32 | 39.70 | 49.69 |
| LoCoOp | 54.57 | **48.65** | 58.27 | 52.35 | 53.91 |

(e) StanfordCars.

|  | MaxLogit | MLS-M | Energy | MLS-E | MCM |
|---|---|---|---|---|---|
| CoOp | 32.58 | 31.62 | 32.23 | **30.96** | 64.30 |
| CoCoOp | 28.76 | 27.18 | 28.37 | **26.83** | 63.84 |
| IVLP | 33.89 | 26.07 | 33.75 | **25.40** | 59.38 |
| KgCoOp | 29.06 | 27.77 | 28.62 | **27.05** | 64.69 |
| ProGrad | 32.44 | 28.42 | 32.69 | **28.19** | 62.17 |
| MaPLe | 30.75 | 26.05 | 30.20 | **25.29** | 62.20 |
| PromptSRC | 26.27 | 22.88 | 26.11 | **22.29** | 59.11 |
| LoCoOp | 41.93 | 31.83 | 42.86 | **31.34** | 64.56 |

(f) Flowers102.

|  | MaxLogit | MLS-M | Energy | MLS-E | MCM |
|---|---|---|---|---|---|
| CoOp | 40.02 | **36.14** | 42.18 | 37.55 | 45.66 |
| CoCoOp | 50.27 | **45.51** | 53.82 | 46.99 | 62.92 |
| IVLP | 54.62 | **49.00** | 59.10 | 51.31 | 55.70 |
| KgCoOp | 56.18 | **43.15** | 62.44 | 45.23 | 57.90 |
| ProGrad | 44.28 | **38.84** | 49.71 | 43.54 | 46.41 |
| MaPLe | 54.00 | **47.94** | 58.30 | 50.79 | 57.83 |
| PromptSRC | 38.96 | **34.33** | 42.84 | 36.78 | 43.15 |
| LoCoOp | 52.83 | **46.72** | 59.87 | 51.74 | 54.17 |

(g) Food101

|  | MaxLogit | MLS-M | Energy | MLS-E | MCM |
|---|---|---|---|---|---|
| CoOp | 54.00 | **49.48** | 55.28 | 50.05 | 64.07 |
| CoCoOp | 41.85 | **37.57** | 43.18 | 37.95 | 55.24 |
| IVLP | 44.36 | **35.63** | 45.66 | 35.66 | 56.66 |
| KgCoOp | 44.42 | **35.98** | 46.71 | 36.35 | 54.18 |
| ProGrad | 48.58 | 40.02 | 50.78 | **39.99** | 58.86 |
| MaPLe | 45.70 | **35.26** | 47.53 | 35.38 | 54.93 |
| PromptSRC | 39.21 | **34.61** | 40.21 | 34.84 | 54.50 |
| LoCoOp | 58.93 | **42.34** | 65.29 | 44.22 | 56.78 |

(h) FGVCAircraft.

|  | MaxLogit | MLS-M | Energy | MLS-E | MCM |
|---|---|---|---|---|---|
| CoOp | 83.67 | 82.75 | 81.31 | **80.09** | 96.60 |
| CoCoOp | 84.31 | 81.95 | 81.38 | **76.55** | 97.73 |
| IVLP | 79.19 | 75.35 | 75.61 | **69.78** | 96.33 |
| KgCoOp | 79.80 | 79.98 | **72.11** | 72.30 | 97.50 |
| ProGrad | 84.79 | 81.38 | 82.41 | **78.20** | 96.26 |
| MaPLe | 81.98 | 80.73 | 78.68 | **75.01** | 97.15 |
| PromptSRC | 76.62 | 75.53 | 69.47 | **69.00** | 96.76 |
| LoCoOp | 86.36 | 83.58 | 83.64 | **79.53** | 97.11 |

(i) SUN397.

|  | MaxLogit | MLS-M | Energy | MLS-E | MCM |
|---|---|---|---|---|---|
| CoOp | 71.81 | 69.50 | 73.24 | 70.54 | **67.98** |
| CoCoOp | 70.74 | 65.62 | 73.55 | 67.30 | **65.10** |
| IVLP | 69.31 | **63.76** | 71.54 | 65.08 | 65.20 |
| KgCoOp | 70.46 | 67.17 | 74.01 | 70.02 | **64.31** |
| ProGrad | 73.83 | 68.12 | 76.75 | 70.20 | **66.65** |
| MaPLe | 68.72 | **63.34** | 71.20 | 64.64 | 65.25 |
| PromptSRC | 66.85 | **61.55** | 69.99 | 63.56 | 62.12 |
| LoCoOp | 76.47 | 67.24 | 81.12 | 71.02 | **65.51** |

(j) DTD.

|  | MaxLogit | MLS-M | Energy | MLS-E | MCM |
|---|---|---|---|---|---|
| CoOp | 85.34 | **85.16** | 86.00 | 85.21 | 87.99 |
| CoCoOp | 88.48 | **87.50** | 88.92 | 87.58 | 90.93 |
| IVLP | **87.08** | 87.11 | 87.86 | 87.82 | 88.19 |
| KgCoOp | 88.52 | 86.93 | 88.49 | **86.42** | 88.74 |
| ProGrad | 87.78 | **87.55** | 88.90 | 88.31 | 89.19 |
| MaPLe | 88.85 | **86.81** | 89.53 | 87.31 | 89.07 |
| PromptSRC | 84.52 | 83.56 | 86.43 | 84.32 | **83.29** |
| LoCoOp | 84.51 | **83.94** | 85.52 | 85.52 | 87.82 |

(k) EuroSAT.

|  | MaxLogit | MLS-M | Energy | MLS-E | MCM |
|---|---|---|---|---|---|
| CoOp | **85.12** | 85.20 | 85.49 | 85.25 | 87.62 |
| CoCoOp | 82.25 | 82.38 | 82.05 | **81.58** | 90.61 |
| IVLP | 81.16 | **73.84** | 81.29 | 74.34 | 82.81 |
| KgCoOp | 82.08 | 82.30 | 82.60 | **79.15** | 89.72 |
| ProGrad | **81.61** | 82.31 | 82.91 | 82.99 | 86.46 |
| MaPLe | 75.22 | 68.41 | 76.75 | **67.28** | 77.79 |
| PromptSRC | 65.95 | **65.83** | 67.18 | 66.27 | 78.29 |
| LoCoOp | 81.91 | 80.88 | 82.06 | **79.99** | 90.22 |

(l) UCF101.

|  | MaxLogit | MLS-M | Energy | MLS-E | MCM |
|---|---|---|---|---|---|
| CoOp | 55.64 | **50.12** | 56.84 | 51.16 | 61.80 |
| CoCoOp | 57.04 | **50.37** | 58.31 | 50.90 | 63.33 |
| IVLP | 56.96 | **47.71** | 58.72 | 48.52 | 59.62 |
| KgCoOp | 54.77 | **50.16** | 56.41 | 50.89 | 58.62 |
| ProGrad | 57.39 | **48.84** | 59.53 | 50.75 | 59.14 |
| MaPLe | 57.16 | **49.27** | 58.65 | 49.51 | 63.36 |
| PromptSRC | 51.62 | **46.17** | 53.91 | 47.33 | 56.65 |
| LoCoOp | 67.38 | **55.38** | 70.46 | 57.05 | 61.10 |

(m) CIFAR10.

|  | MaxLogit | MLS-M | Energy | MLS-E | MCM |
|---|---|---|---|---|---|
| CoOp | 53.49 | **49.45** | 56.82 | 51.62 | 53.53 |
| CoCoOp | 37.07 | **29.79** | 41.65 | 30.48 | 37.02 |
| IVLP | 39.00 | **30.87** | 43.38 | 32.73 | 38.67 |
| KgCoOp | 38.30 | **30.85** | 45.90 | 33.20 | 35.12 |
| ProGrad | 52.48 | **40.59** | 58.65 | 43.18 | 45.00 |
| MaPLe | 40.75 | **30.00** | 46.10 | 31.57 | 36.95 |
| PromptSRC | 30.54 | **22.74** | 35.64 | 24.13 | 27.73 |
| LoCoOp | 51.96 | **36.97** | 60.82 | 39.74 | 37.78 |

(n) CIFAR100.

|  | MaxLogit | MLS-M | Energy | MLS-E | MCM |
|---|---|---|---|---|---|
| CoOp | 74.33 | 71.48 | 75.30 | **71.41** | 86.40 |
| CoCoOp | 72.96 | 71.09 | 73.60 | **69.70** | 86.82 |
| IVLP | 56.39 | **45.86** | 61.31 | 46.22 | 65.16 |
| KgCoOp | 73.27 | 73.66 | 75.27 | **73.58** | 87.26 |
| ProGrad | 77.89 | **67.49** | 80.70 | 67.79 | 85.42 |
| MaPLe | 54.19 | **43.60** | 59.11 | 43.73 | 66.44 |
| PromptSRC | 51.05 | **45.82** | 55.95 | 47.00 | 59.12 |
| LoCoOp | **80.71** | 81.44 | 81.91 | 82.49 | 88.94 |

A.3.5    FAR OOD DETECTION

We provide additional far OOD detection results of AUROC and FPR in Table 23 and Table 24 respectively.

Table 23: OOD detection AUROC (↑) averaged over 16, 8, 4, 2, and 1 few-shot settings with 3 random seeds.

(a) Places.

|  | MaxLogit | | | Energy | | | MCM |
|---|---|---|---|---|---|---|---|
|  | Original | MLS-M | △ | Original | MLS-E | △ | |
| CoOp | 96.22 | **96.78** | +0.56 | 95.80 | 96.65 | +0.85 | 95.92 |
| CoCoOp | **96.90** | 96.79 | -0.11 | 96.50 | 96.51 | +0.01 | 96.21 |
| IVLP | 95.79 | **96.01** | +0.22 | 95.18 | 95.77 | +0.60 | 95.97 |
| KgCoOp | 96.67 | **96.91** | +0.24 | 96.12 | 96.74 | +0.62 | 96.30 |
| ProGrad | 95.21 | **96.05** | +0.84 | 94.52 | 95.76 | +1.25 | 95.30 |
| MaPLe | **96.61** | 96.52 | -0.09 | 96.00 | 96.11 | +0.11 | 96.26 |
| PromptSRC | 96.54 | **96.59** | +0.05 | 96.00 | 96.20 | +0.21 | 96.40 |
| LoCoOp | 94.67 | **97.10** | +2.43 | 91.59 | 96.45 | +4.86 | 96.57 |

(b) SUN.

|  | MaxLogit | | | Energy | | | MCM |
|---|---|---|---|---|---|---|---|
|  | Original | MLS-M | △ | Original | MLS-E | △ | |
| CoOp | 96.49 | 97.88 | +1.39 | 95.69 | 97.62 | +1.93 | **97.94** |
| CoCoOp | 98.17 | **98.47** | +0.30 | 97.64 | 98.21 | +0.57 | 98.30 |
| IVLP | 97.61 | 98.20 | +0.58 | 96.98 | 97.97 | +0.99 | **98.33** |
| KgCoOp | 97.50 | **98.47** | +0.96 | 96.61 | 98.29 | +1.68 | 98.33 |
| ProGrad | 96.51 | **97.73** | +1.22 | 95.60 | 97.35 | +1.75 | 97.65 |
| MaPLe | 97.91 | **98.54** | +0.63 | 97.13 | 98.21 | +1.07 | 98.42 |
| PromptSRC | 97.63 | 98.11 | +0.48 | 96.88 | 97.64 | +0.76 | **98.44** |
| LoCoOp | 95.38 | **98.94** | +3.56 | 90.79 | 98.20 | +7.40 | 98.80 |

(c) Texture.

|  | MaxLogit | | | Energy | | | MCM |
|---|---|---|---|---|---|---|---|
|  | Original | MLS-M | △ | Original | MLS-E | △ | |
| CoOp | 93.38 | 93.74 | +0.36 | 92.51 | 93.07 | +0.56 | **96.66** |
| CoCoOp | 94.79 | 95.29 | +0.50 | 93.72 | 94.51 | +0.79 | **96.99** |
| IVLP | 93.32 | 95.49 | +2.17 | 91.78 | 94.90 | +3.12 | **97.09** |
| KgCoOp | 93.59 | 96.30 | +2.71 | 92.03 | 95.90 | +3.86 | **96.62** |
| ProGrad | 91.36 | 94.22 | +2.86 | 89.72 | 93.37 | +3.65 | **96.73** |
| MaPLe | 93.36 | 93.89 | +0.52 | 91.73 | 92.64 | +0.91 | **97.50** |
| PromptSRC | 92.69 | 94.52 | +1.83 | 90.94 | 93.52 | +2.58 | **96.81** |
| LoCoOp | 93.18 | 96.16 | +2.98 | 89.80 | 94.82 | +5.02 | **96.96** |

(d) iNaturalist.

|  | MaxLogit | | | Energy | | | MCM |
|---|---|---|---|---|---|---|---|
|  | Original | MLS-M | △ | Original | MLS-E | △ | |
| CoOp | 94.58 | 97.23 | +2.65 | 92.66 | 96.28 | +3.62 | **97.86** |
| CoCoOp | 97.02 | **97.65** | +0.63 | 95.61 | 96.58 | +0.97 | 97.52 |
| IVLP | 95.98 | 97.88 | +1.91 | 93.74 | 96.89 | +3.15 | **98.26** |
| KgCoOp | 97.05 | **98.49** | +1.44 | 95.61 | 98.00 | +2.39 | 97.96 |
| ProGrad | 92.90 | 96.04 | +3.14 | 90.46 | 94.73 | +4.27 | **97.39** |
| MaPLe | 95.80 | 97.60 | +1.80 | 93.43 | 96.25 | +2.82 | **98.32** |
| PromptSRC | 96.37 | 96.76 | +0.38 | 94.41 | 95.01 | +0.60 | **98.44** |
| LoCoOp | 93.41 | **98.46** | +5.05 | 87.11 | 97.08 | +9.97 | 97.95 |

Table 24: OOD detection FPR95 (↓) averaged over 16, 8, 4, 2, and 1 few-shot settings with 3 random seeds.

(a) Places.

| | MaxLogit | | | Energy | | | MCM |
|---|---|---|---|---|---|---|---|
| | Original | MLS-M | △ | Original | MLS-E | △ | |
| CoOp | 17.58 | **13.56** | -4.01 | 20.90 | 14.42 | -6.48 | 16.40 |
| CoCoOp | 13.01 | **12.92** | -0.09 | 15.41 | 14.09 | -1.33 | 14.78 |
| IVLP | 17.82 | **15.32** | -2.50 | 22.14 | 16.67 | -5.47 | 15.41 |
| KgCoOp | 14.43 | **12.46** | -1.97 | 18.32 | 13.33 | -4.99 | 14.34 |
| ProGrad | 22.47 | **16.28** | -6.19 | 28.38 | 17.85 | -10.52 | 18.51 |
| MaPLe | 15.37 | **14.01** | -1.37 | 19.35 | 16.74 | -2.62 | 14.32 |
| PromptSRC | 14.88 | **13.77** | -1.11 | 18.70 | 16.04 | -2.66 | 13.77 |
| LoCoOp | 27.58 | **11.05** | -16.52 | 53.58 | 14.25 | -39.33 | 12.43 |

(b) SUN.

| | MaxLogit | | | Energy | | | MCM |
|---|---|---|---|---|---|---|---|
| | Original | MLS-M | △ | Original | MLS-E | △ | |
| CoOp | 19.44 | **10.06** | -9.38 | 26.66 | 11.60 | -15.06 | 10.24 |
| CoCoOp | 8.37 | **6.39** | -1.98 | 11.64 | 7.18 | -4.46 | 8.15 |
| IVLP | 10.77 | 7.92 | -2.86 | 15.85 | 9.35 | -6.50 | **7.49** |
| KgCoOp | 12.98 | **6.81** | -6.17 | 20.58 | 7.32 | -13.26 | 7.77 |
| ProGrad | 19.14 | **10.37** | -8.77 | 27.60 | 12.33 | -15.28 | 11.37 |
| MaPLe | 9.67 | **5.91** | -3.76 | 15.09 | 7.69 | -7.40 | 7.10 |
| PromptSRC | 11.48 | 8.37 | -3.11 | 17.35 | 11.28 | -6.08 | **6.84** |
| LoCoOp | 29.82 | **3.76** | -26.06 | 64.29 | 7.80 | -56.49 | 4.51 |

(c) Texture.

| | MaxLogit | | | Energy | | | MCM |
|---|---|---|---|---|---|---|---|
| | Original | MLS-M | △ | Original | MLS-E | △ | |
| CoOp | 35.02 | 31.39 | -3.63 | 43.04 | 37.32 | -5.72 | **17.29** |
| CoCoOp | 26.11 | 23.14 | -2.96 | 36.68 | 29.31 | -7.37 | **15.31** |
| IVLP | 33.43 | 20.25 | -13.18 | 43.80 | 24.40 | -19.40 | **13.70** |
| KgCoOp | 33.06 | 18.30 | -14.76 | 46.06 | 22.26 | -23.79 | **17.23** |
| ProGrad | 42.31 | 28.67 | -13.64 | 54.27 | 36.97 | -17.30 | **16.12** |
| MaPLe | 35.20 | 29.43 | -5.77 | 47.83 | 40.47 | -7.36 | **12.49** |
| PromptSRC | 37.51 | 26.90 | -10.61 | 52.59 | 36.40 | -16.19 | **15.64** |
| LoCoOp | 39.95 | 19.69 | -20.25 | 68.07 | 32.19 | -35.87 | **14.72** |

(d) iNaturalist.

| | MaxLogit | | | Energy | | | MCM |
|---|---|---|---|---|---|---|---|
| | Original | MLS-M | △ | Original | MLS-E | △ | |
| CoOp | 32.70 | 13.64 | -19.07 | 48.16 | 20.24 | -27.93 | **9.38** |
| CoCoOp | 14.03 | **9.68** | -4.35 | 24.75 | 16.71 | -8.04 | 12.13 |
| IVLP | 22.51 | 8.85 | -13.66 | 39.50 | 15.17 | -24.33 | **7.28** |
| KgCoOp | 14.76 | **5.76** | -9.00 | 25.50 | 8.60 | -16.90 | 9.32 |
| ProGrad | 43.71 | 21.29 | -22.42 | 58.34 | 32.44 | -25.90 | **13.93** |
| MaPLe | 23.97 | 10.38 | -13.58 | 42.23 | 19.62 | -22.62 | **6.76** |
| PromptSRC | 19.38 | 16.03 | -3.35 | 33.65 | 29.98 | -3.67 | **5.87** |
| LoCoOp | 51.09 | **6.17** | -44.92 | 85.97 | 18.46 | -67.51 | 8.85 |

### A.3.6 COMPARISON WITH ZERO-SHOT CLIP OOD DETECTION

Although our work is specifically designed for few-shot prompt learning CLIP models and is not comparable to zero-shot CLIP OOD detection models, we here provide a comparison with CLIPN (Wang et al., 2023) in Table 25. Following the original setting, we use the maximum softmax probability as an OOD score. Note that MLS cannot be applied to CLIPN as it is a zero-shot model which lacks the context vectors.

Table 25: Near OOD AUROC (↑) and FPR95 (↓) of prompt learning models and CLIPN averaged over 13 datasets.

| | AUROC ↑ | | | FPR95 ↓ | | |
| --- | --- | --- | --- | --- | --- | --- |
| | MLS-M | MLS-E | MCM | MLS-M | MLS-E | MCM |
| CoOp | **81.84** | 81.71 | 79.41 | **54.85** | 55.37 | 63.70 |
| CoCoOp | **82.74** | 82.74 | 79.28 | **51.67** | 51.90 | 63.76 |
| IVLP | 84.34 | **84.40** | 80.95 | **48.58** | 49.19 | 60.05 |
| KgCoOp | 83.12 | **83.23** | 79.82 | **51.52** | 51.90 | 62.20 |
| ProGrad | **82.35** | 81.93 | 79.91 | **54.02** | 55.34 | 62.89 |
| MaPLe | 83.94 | **83.99** | 80.54 | **48.36** | 48.76 | 60.68 |
| PromptSRC | 85.77 | **85.88** | 82.34 | **44.60** | 45.15 | 55.86 |
| LoCoOp | **81.74** | 81.25 | 79.22 | **55.76** | 57.80 | 63.37 |
| CLIPN | N/A | N/A | 79.64 | N/A | N/A | 64.44 |

### A.3.7 IMAGENET PROTOCOL RESULTS

In addition to 13 datasets used in the main experiments, we also provide experimental results on ImageNet Protocol (Palechor et al., 2023) in Table 26. We follow the four-split setting used by Li et al. (2024).

Table 26: OOD AUROC (↑) of 8 prompt learning models averaged over 4 ImageNet protocol datasets using the MaxLogit score, the Energy score, MLS-M, MLS-E, and MCM.

| | MaxLogit | MLS-M | Energy | MLS-E | MCM |
| --- | --- | --- | --- | --- | --- |
| CoOp | 96.46 | **96.74** | 96.43 | 96.72 | 91.11 |
| CoCoOp | 97.42 | **97.69** | 97.34 | 97.65 | 93.77 |
| IVLP | 97.19 | **97.60** | 97.03 | 97.50 | 94.35 |
| KgCoOp | 97.34 | **97.57** | 97.24 | 97.50 | 94.09 |
| ProGrad | 96.84 | **97.20** | 96.73 | 97.12 | 93.18 |
| MaPLe | 97.48 | **97.48** | 97.38 | 97.23 | 94.16 |
| PromptSRC | 97.57 | **97.73** | 97.49 | 97.66 | 94.61 |
| LoCoOp | 96.59 | **97.19** | 96.15 | 96.94 | 93.76 |

### A.3.8 OTHER OOD SCORES

We intentionally excluded OOD scores that require model retraining, architectural modifications, access to OOD samples, or those that are incompatible with fine-tuned prompt learning models. For examples, LogitNorm (Wei et al., 2022) requires training a model with its dedicated training loss to use the score, and relative Mahalanobis distance (RMD) (Ren et al., 2021) is intended to be used with a traditional classifier which has a classifier head. Even if RMD is applied to image features of the CLIP prompt learning models, all prompt learning models output the same RMD score as their image networks are not optimised during fine-tuning. Nonetheless, we provide experimental results of LogitNorm and RMD in Table 27, using 12 datasets excluding ImageNet with 16, 8, 4, and 2-shot settings. For LogitNorm, we trained the prompt learning models with LogitNorm loss substituting the original cross entropy loss. Following the original setting in (Wei et al., 2022), we used the temperature scale of 0.04 and calculated the maximum softmax probability score.

Table 27: Near OOD AUROC (↑) of prompt learning models averaged over 12 datasets with Logit-Norm and RMD scores.

|  | MLS-M | MLS-E | MCM | LogitNorm | RMD |
|---|---|---|---|---|---|
| CoOp | **80.77** | 80.61 | 78.24 | 77.55 | 57.58 |
| CoCoOp | **81.71** | 81.70 | 78.04 | 77.86 | 57.58 |
| IVLP | 83.48 | **83.52** | 79.89 | 79.56 | 57.58 |
| KgCoOp | 82.20 | **82.30** | 78.65 | 77.38 | 57.58 |
| ProGrad | **81.32** | 80.86 | 78.80 | 77.16 | 57.58 |
| MaPLe | 83.07 | **83.11** | 79.47 | 78.39 | 57.58 |
| PromptSRC | 84.97 | **85.08** | 81.33 | 80.28 | 57.58 |
| LoCoOp | **80.67** | 80.14 | 78.05 | 77.52 | 57.58 |

## A.4 ADDITIONAL DEMONSTRATIONS

Figure 7: Comparisons of MaxLogit score and Context score computed by MaPLe (Khattak et al., 2023a) on OxfordPets (Parkhi et al., 2012) (16-shot). Three ID images from the red box are shown, and six OOD images from two green boxes are shown.

In Figure 7, ID and OOd examples with MaxLogit score and Context score are plotted.We also provide additional plots of Figure 3a with all prompt learning models and datasets used using 16-shot setting.

Figure 8: Additional demonstrations of the relationship between MaxLogit score and Context Score using Caltech101 (16-shots) with different prompt learning models.

Figure 9: Additional demonstrations of the relationship between Energy score and Context Score using Caltech101 (16-shots) with different prompt learning models.

Figure 10: Additional demonstrations of the relationship between MaxLogit score and Context Score using ImageNet (16-shots) with different prompt learning models.

Figure 11: Additional demonstrations of the relationship between Energy score and Context Score using ImageNet (16-shots) with different prompt learning models.

Figure 12: Additional demonstrations of the relationship between MaxLogit score and Context Score using OxfordPets (16-shots) with different prompt learning models.

Figure 13: Additional demonstrations of the relationship between Energy score and Context Score using OxfordPets (16-shots) with different prompt learning models.

Figure 14: Additional demonstrations of the relationship between MaxLogit score and Context Score using StanfordCars (16-shots) with different prompt learning models.

Figure 15: Additional demonstrations of the relationship between Energy score and Context Score using StanfordCars (16-shots) with different prompt learning models.

Figure 16: Additional demonstrations of the relationship between MaxLogit score and Context Score using Flowers102 (16-shots) with different prompt learning models.

Figure 17: Additional demonstrations of the relationship between Energy score and Context Score using Flowers102 (16-shots) with different prompt learning models.

Figure 18: Additional demonstrations of the relationship between MaxLogit score and Context Score using Food101 (16-shots) with different prompt learning models.

Figure 19: Additional demonstrations of the relationship between Energy score and Context Score using Food101 (16-shots) with different prompt learning models.

Figure 20: Additional demonstrations of the relationship between MaxLogit score and Context Score using FGVCAircraft (16-shots) with different prompt learning models.

Figure 21: Additional demonstrations of the relationship between Energy score and Context Score using FGVCAircraft (16-shots) with different prompt learning models.

Figure 22: Additional demonstrations of the relationship between MaxLogit score and Context Score using SUN397 (16-shots) with different prompt learning models.

Figure 23: Additional demonstrations of the relationship between Energy score and Context Score using SUN397 (16-shots) with different prompt learning models.

Figure 24: Additional demonstrations of the relationship between MaxLogit score and Context Score using DTD (16-shots) with different prompt learning models.

Figure 25: Additional demonstrations of the relationship between Energy score and Context Score using DTD (16-shots) with different prompt learning models.

Figure 26: Additional demonstrations of the relationship between MaxLogit score and Context Score using EuroSAT (16-shots) with different prompt learning models.

Figure 27: Additional demonstrations of the relationship between Energy score and Context Score using EuroSAT (16-shots) with different prompt learning models.

Figure 28: Additional demonstrations of the relationship between MaxLogit score and Context Score using UCF101 (16-shots) with different prompt learning models.

Figure 29: Additional demonstrations of the relationship between Energy score and Context Score using UCF101 (16-shots) with different prompt learning models.

Figure 30: Additional demonstrations of the relationship between MaxLogit score and Context Score using CIFAR10 (16-shots) with different prompt learning models.

Figure 31: Additional demonstrations of the relationship between Energy score and Context Score using CIFAR10 (16-shots) with different prompt learning models.

Figure 32: Additional demonstrations of the relationship between MaxLogit score and Context Score using CIFAR100 (16-shots) with different prompt learning models.

Figure 33: Additional demonstrations of the relationship between Energy score and Context Score using CIFAR100 (16-shots) with different prompt learning models.