# OpenReview forum: "Enhancing Near OOD Detection in Prompt Learning: Maximum Gains, Minimal Costs"
_ICLR.cc/2025/Conference — Submitted to ICLR 2025_

### Official Review · Reviewer_QkT2 · 2024-10-18

**Soundness:** 2
**Presentation:** 3
**Contribution:** 2
**Rating:** 3
**Confidence:** 4

**Summary:**

The paper introduces a new logit-based score named Marginal Logit Score (MLS) based on existing logit scores and a new complementary score named Context score.

**Strengths:**

1. The proposed method boosts existing prompt learning methods' near OOD detection performance in AUROC by up to 11.67% with minimal computational

2. The method can be easily applied to any prompt learning model without change in architecture or model re-training.

**Weaknesses:**

1. Lack of novelty: The contribution of the paper is limited. All I could discover is a post-hoc OOD score. More description and deeper analyses are needed to certify its usefulness.

2. Incomplete comparison with current methods: The paper mainly compares with few-shot based methods but not OOD-based methods for VLM [1, 2, 3], which should be addressed.

3. Marginal improvements: The performance gains in the main tables are marginal, even when compared with the basic MaxLogit.

4. The writing of the paper should be improved and standardized, including but not limited to the usage of quotation marks in Sec. 2.1 and the usage of superscripts and subscripts in Sec. 3.1, which may confuse readers.

**Questions:**

1. The OOD score is a measure of OOD detection performance. There have been numerous methods for calculating OOD scores in CLIP-based out-of-distribution detection, including, but not limited to, the papers mentioned [1, 2, 3, 4]. The comparison is too narrow, and I suggest that the authors compare with more methods to analyze the effectiveness of the proposed approach.

2. The comparison is not fair. In Table 2, the paper merely compares with prompt-based methods and achieves seemingly significant improvements. However, as shown in Table 3, the improvement becomes marginal when assessed using different metrics and OOD-based methods.

3. The writing may be somewhat confusing. For example, in the near OOD setting, which dataset is the ID dataset? I did my best but could not find any description.

4. Based on the results shown for the given datasets, the effectiveness of the method is not convincing. I suggest that the authors provide more visualizations and additional results to demonstrate its effectiveness and generalization.


[1] Learning Transferable Negative Prompts for Out-of-Distribution Detection.

[2] Negative Label Guided OOD Detection with Pretrained Vision-Language Models.

[3] Enhancing Outlier Knowledge for Few-Shot Out-of-Distribution Detection with Extensible Local Prompts.

[4] CLIPN for Zero-Shot OOD Detection: Teaching CLIP to Say No.

---

> ### Author Response · Authors · 2024-11-18
> **Reponse to Reviewer QkT2**
>
> We would like to clarify the reviewer’s comments.
>
> ### 1. “Lack of novelty: The contribution of the paper is limited. All I could discover is a post-hoc OOD score. More description and deeper analyses are needed to certify its usefulness.”
> ---
> We respectfully believe that it is incorrect and unfair to consider the contribution of our method limited due to its post-hoc nature. One of our key contributions is a simple post-hoc OOD score that improves near-OOD detection by up to 11.67% for any CLIP-based prompt-learning model, without the need for retraining or model modification, and with minimal additional computation. To our knowledge, no previous works have addressed the problem studied in the paper. The usefulness of our method is clearly demonstrated by the comprehensive experiments with detailed analysis. If the reviewer has more specific comments on this, we are happy to address them.
>
> ### 2. “Incomplete comparison with current methods: The paper mainly compares with few-shot based methods but not OOD-based methods for VLM [1, 2, 3], which should be addressed.”
> ---
> We would like to emphasise that our focus is improving near OOD detection performance of few-shot CLIP-based prompt learning models without retraining or making any change in the model. As discussed in L514-527, we intentionally excluded OOD methods that require access to OOD samples during training, retraining of prompt learning models, or that are incompatible with fine-tuned CLIP-based prompt-learning models. In other words, these methods do not fit the purpose. Nevertheless, we also have provided comparisons with CLIPN which is one of references the reviewer pointed out, RMD, and LogitNorm in Appendix A.3.6 and A.3.8.
>
> ### 3. “Marginal improvements: The performance gains in the main tables are marginal, even when compared with the basic MaxLogit.”
> ---
> We could not agree with the reviewer on this comment, as our approach demonstrates consistent improvements across 13 datasets and 8 recent prompt-learning models with a minimal post-hoc process. For example, the average AUROC across datasets for LoCoOp, the state-of-the-art method for far-OOD detection in CLIP-based prompt-learning models, is increased by 4.18%, while its average FPR95 is reduced by 9.13%. We believe that the improvement is significant in the literature.
>
> ### 4. “The writing of the paper should be improved and standardized, including but not limited to the usage of quotation marks in Sec. 2.1 and the usage of superscripts and subscripts in Sec. 3.1, which may confuse readers.”
> ---
> We have changed the quotation marks to the standard quotation marks in the revised version. Regarding Section 3.1, we believe that superscripts and subscripts have been used consistently. However, if the reviewer could point out any specific inconsistencies, we would be happy to address them.
>
> ### 5. “The comparison is not fair. In Table 2, the paper merely compares with prompt-based methods and achieves seemingly significant improvements. However, as shown in Table 3, the improvement becomes marginal when assessed using different metrics and OOD-based methods.”
> ---
> We would like to emphasise once again that our work is specifically tailored for fine-tuned CLIP-based prompt-learning methods and is not applicable to other zero-shot CLIP models. Therefore, we believe that comparisons with 8 recent prompt-learning models are appropriate. Additionally, Table 3 demonstrates that our method consistently outperforms MCM in both AUROC and FPR95. For example, the average FPR95 for MaPLe is reduced by 12.32% when comparing MCM to MLS-M.
>
> ### 6. “The writing may be somewhat confusing. For example, in the near OOD setting, which dataset is the ID dataset? I did my best but could not find any description.”
> ---
> Please refer to L373-377 where we provided details of ID and OOD data settings in our original submission. We used the same base-to-new splits used in prompt learning models (Zhou et al., 2022a; Khattak et al., 2023a; Yao et al., 2023; Zhu et al., 2023; Khattak et al., 2023a;b; Miyai et al., 2023) as near OOD tasks.
>
> ### 7. “Based on the results shown for the given datasets, the effectiveness of the method is not convincing. I suggest that the authors provide more visualizations and additional results to demonstrate its effectiveness and generalization.”
> ---
> In our original submission, we provided extensive empirical evaluations using 8 recent prompt learning models, 13 main datasets for near OOD detection, 5 few-shot settings (16-, 8-, 4-, 2-, and 1-shot), 3 random seeds, 1 additional dataset for near OOD detection, 4 far OOD datasets, and comparisons with zero-shot CLIP OOD detection, LogitNorm, and RMD, provided in the main text and Appendix. We kindly ask the reviewer to specify which visualisations or additional results the reviewer believes are missing.

---

> > ### Author Response · Authors · 2024-11-24
> >
> > Dear Reviewer,
> >
> > We hope our responses answer your questions. If you have more questions or comments, please let us know. We are happy discuss it.
> >
> > Best regards,
> >
> > The Authors

---

> > ### Comment · Reviewer_QkT2 · 2024-11-25
> >
> > According to the authors' response,  I have more doubt about the application range of the proposed method, given the point of view raised by the author that the method is **"specifically tailored for fine-tuned CLIP-based prompt-learning methods and is not applicable to other zero-shot CLIP models"**. From the author's claim, all they need to compare is few-shot prompt learning methods and LoCoOp, which achieves seemingly improvements.
> >
> > Moreover, I believe that I have pointed out the concerns about the experimental settings clearly and the authors just keep emphasizing that the improvements against prompt learning based methods are appropriate.
> >
> > I have listed a number of papers that are relevant to the paper (**or at least from my humble perspective**), and the author reply to them partially.
> >
> > Based on the consideration above, I would keep my score until additional explanation or experimental results that could address my concerns.

---

> > > ### Author Response · Authors · 2024-11-25
> > >
> > > Dear Reviewer,
> > >
> > > Thank you for responding to our rebuttal. We noticed that the reviewer increased their confidence and we appreciate the additional effort on reviewing our paper. Let us clarify the intended scope of our approach.
> > >
> > > ### 1. “Application range”
> > > ---
> > > The first few-shot CLIP-based prompt learning method was proposed in 2022 [a] (the original paper got 1200+ citations) and the idea has been recently shown to be efficient yet effective methods to fine-tune CLIP models without optimising textual and vision encoders. Until recently, there have been 20+ methods in this research line. However, their near-OOD capabilities have been largely overlooked, and we found that existing OOD score functions perform poorly with these models. Our method can plug in to most of them to improve near OOD detection performance with almost zero cost. We have shown significant improvements with our method on many few-shot CLIP-based prompt learning methods. We believe that our method has wide applicability.
> > >
> > > ### 2. “Experimental settings”
> > > ---
> > > If we are understanding the original reviews right, the reviewer’s concerns on our experimental settings were that the reviewer didn’t find the definition of the ID and OOD datasets. We have provided the detailed location of description of the settings in our original paper.
> > >
> > > ### 3. “A number of papers”
> > > ---
> > > We first would like to note that the papers [2,3,4] were included in our original paper and we appreciate the reference [1], which will be included in the paper.
> > >
> > > Our main motivation and intended scope of our approach is to enhance near OOD performance of existing prompt learning models with a new scoring function. Our focus is on improving the existing prompt learning models rather than proposing a new SOTA prompt learning model. As a new scoring function, our main competitors are the existing scoring functions including MaxLogit score and Energy score. In terms of these comparisons, we believe that empirical validations of the improvement for individual prompt learning models justify the effectiveness of our approach.
> > >
> > > Moreover, We didn’t compare with the methods in the papers reviewer mentioned because they are inapplicable to our problem. All of the four methods are independent models with new training methods that enhance OOD detection, while ours is a post-hoc scoring function applied to different base methods without (re)training.
> > >
> > > We hope this clarifies the experimental settings and justifies the applicability of our approach.
> > >
> > > Thank you.
> > >
> > > [a] Zhou, Kaiyang, et al. "Conditional prompt learning for vision-language models." Proceedings of the IEEE/CVF conference on computer vision and pattern recognition. 2022.
> > >
> > > Best regards,
> > >
> > > The Authors

---

> ### Comment · Reviewer_QkT2 · 2024-12-03
>
> I totally understand what the author means by mentioning the definition of the ID and OOD datasets. However, I do not agree with the author's opinion that previous method is inapplicable in existing prompt-based methods. Post-hoc OOD score is algorithm-irrelevant and should not be constrainted within certain methods. I would keep my opinion that comparing with MaxLogit, Energy and MCM is not sufficient enough to demonstrate the usefulness (the most updated algorithm to be 2022)
>
> As for the performance improvement, I do not think the author's explanation is persuasive and would keep my opinion.
>
> Last but not least, I agree with the question about context score raised by 19PE and the counterintuition mitigates the persuasiveness of the paper.

---

### Official Review · Reviewer_19PE · 2024-10-26

**Soundness:** 3
**Presentation:** 3
**Contribution:** 2
**Rating:** 5
**Confidence:** 3

**Summary:**

This paper introduces a novel post-hoc method to improve the near Out-Of-Distribution (OOD) detection capabilities of prompt learning models for vision-language tasks, without requiring architectural changes or retraining. The proposed method, named Marginal Logit Score (MLS), leverages the Context Score computed based on a variant of existing prompts to enhance near OOD detection performance . Experiments on 13 datasets demonstrate that MLS significantly boosts the near OOD detection performance, while maintaining the same classification accuracy.

**Strengths:**

1. The finding is interesting.

2. The experimental results have validated the effectiveness of the proposed method.

**Weaknesses:**

1. The cause of the phenomenon is not clearly explained.

The author find: Compared to ID images, OOD images have a smaller gap between the original score and the Context Score. Based on this phenomenon, the author proposes combining Energy (or MaxLogit) and Context Score to identify OOD. However, the author does not deeply analyze this phenomenon and explain its cause. I speculate that the cause might be because when CLASS is present, CLASS plays the main role in the Prompt, and at this time, regardless of whether it is an ID image or an OOD image, the attention is on CLASS, so the scores are similar. When CLASS is removed, the attention of ID images and OOD images to the remaining vectors in the Prompt will differ, thus producing different scores.

2. The discussion is not thorough enough.

Why combine Energy (or MaxLogit) with Context Score? I see that there is already a significant difference in Context Score between ID images and OOD images. Can't we directly use the Context Score to distinguish between the two types of images?

This work discovers a phenomenon in prompt-based learning and designs an OOD detection algorithm based on this phenomenon, but it does not delve into the cause for the occurrence of this phenomenon, so I am somewhat concerned that this may not be a universal phenomenon. If the authors could provide more analysis on it, the article could be further improved.

**Questions:**

Please refer to the Weaknesses.

---

> ### Author Response · Authors · 2024-11-18
> **Response to Reviewer 19PE (1/2)**
>
> We thank the reviewer for the comments and would like to answer the questions below.
>
> ### 1. “The cause of the phenomenon is not clearly explained.”
> ---
> We agree with the reviewer that when the class name is present in a prompt, logit-based scores are generally higher for ID images than for OOD images, which is a way to reflect the model’s confidence that an image belongs to an ID label. Model confidence is the key to distinguishing between ID and OOD samples.
>
> In this paper, we propose a new way to better reflect the model’s confidence, which is the gap between the logit-based score and the context score namely Marginal Logit Score (MLS). The reason that MLS is better than the original logit-based score is quite intuitive. Recall that the context score measures the distance between the representation of an image and the representation of the prompt without the CLASS label and it contributes to the logit-based score. We believe there are two main factors that lead to high logit-based scores: 1) the model being genuinely confident in its prediction for an ID image, and 2) the model overly paying attention to the context vectors for an OOD image which leads to high logit-based scores. Relying solely on logit-based scores cannot differentiate between these two scenarios and, therefore, fails to accurately reflect the model’s confidence. This limitation explains why the model struggles to distinguish between ID and OOD images when their scores overlap, as illustrated by the shaded region in Figure 3(a).
>
> In this problematic overlapping region, the context score provides a valuable complement to the logit-based scores where the context score tends to be lower for ID images than for OOD images. This occurs because the model recognises ID images as belonging to ID labels, so the context score is relatively lower for ID images than for OOD images as the context score is computed using a prompt without the ID label. Thus, subtracting the context score from the logit-based scores yields a more distinctive scoring function, as illustrated in Figures 3(b) and 3(c), where the shaded overlapping region is notably reduced.
>
> ### 2. “The discussion is not thorough enough.”
> ---
> The context score should be used alongside logit-based scores because, on its own, the context score is not an effective scoring function. We emphasise that the context score is meant to complement the logit-based scores, enhancing them rather than serving as a standalone measure. This can be observed by examining the range of x-values in Figure 3(a), which represents the context score. The ranges of x-values for ID and OOD images completely overlap, making it impossible to set a threshold that effectively separates ID and OOD images using only the context score. The synergy and improved distinguishability arise only when the context score is combined with logit-based scores.
>
> As discussed before, distinguishing between ID and OOD images is about capturing model confidence accurately. Using the context score alone might not be a good choice. We further provide empirical results with two datasets comparing between using the context score alone and using MLS to illustrate our claim. It can be seen that using the context score alone results in a poor performance.
>
> Table R1. Near OOD detection results (AUROC) comparing MLS and Context score with Caltech101 (16-shots).
> \\begin{array}{cccc}  & \\text{MLS-M} & \\text{MLS-E} & \\text{Context} \\\ \\hline \\text{CoOp} & 92.93 & 92.50 & 49.85 \\\ \\text{CoCoOp} & 91.63 & 91.06 & 44.46 \\\ \\text{IVLP} & 94.08 & 93.74 & 35.80 \\\ \\text{KgCoOp} & 90.11 & 88.86 & 38.66 \\\ \\text{ProGrad} & 90.24 & 88.95 & 42.94 \\\ \\text{MaPLe} & 95.00 & 94.76 & 36.55 \\\ \\text{PromptSRC} & 90.49 & 89.66 & 40.79 \\\ \\text{LoCoOp} & 89.43 & 88.03 & 42.02  \\end{array}
>
> Table R2. Near OOD detection results (AUROC) comparing MLS and Context score with OxfordPets (16-shots).
> \\begin{array}{cccc}  & \\text{MLS-M} & \\text{MLS-E} & \\text{Context} \\\ \\hline \\text{CoOp} & 86.80 & 86.66 & 44.12 \\\ \\text{CoCoOp} & 93.63 & 93.56 & 48.31 \\\ \\text{IVLP} & 93.12 & 92.99 & 46.12 \\\ \\text{KgCoOp} & 92.80 & 92.49 & 50.56 \\\ \\text{ProGrad} & 87.83 & 87.60 & 49.43 \\\ \\text{MaPLe} & 94.59 & 94.54 & 45.36 \\\ \\text{PromptSRC} & 94.35 & 94.22 & 45.71 \\\ \\text{LoCoOp} & 89.68 & 88.95 & 42.89 \\end{array}

---

> > ### Author Response · Authors · 2024-11-18
> > **Response to Reviewer 19PE (2/2)**
> >
> > ### 3. “I am somewhat concerned that this may not be a universal phenomenon”
> > ---
> > We have added further evidence of the observed phenomenon across different prompt learning models and datasets in Appendix A.4. Additionally, we would like to highlight that the effectiveness of our approach has been thoroughly validated through extensive empirical evaluations, including 8 recent prompt learning models, 13 main datasets for near OOD detection, 5 few-shot settings (16-, 8-, 4-, 2-, and 1-shot), 3 random seeds, 1 additional dataset for near OOD detection, 4 far OOD datasets, and comparisons with zero-shot CLIP OOD detection, LogitNorm, and RMD, as detailed in the Appendix. We believe this comprehensive validation supports the effectiveness of our approach.

---

> > > ### Author Response · Authors · 2024-11-24
> > >
> > > Dear Reviewer,
> > >
> > > Thanks again for reviewing our paper. We hope our responses answer your questions. If you have more questions or comments, please let us know. We are happy discuss it.
> > >
> > > Best regards,
> > >
> > > The Authors

---

> ### Comment · Reviewer_19PE · 2024-11-27
>
> Thank you for the author's rebuttal. Sorry for the late response.
> While some concerns have been addressed, they do not fully convince me regarding comments #1 and #2. Therefore, I will maintain my original rating.
>
> Regarding #1, the key should be the difference of logits between ID and OOD images rather than "logit-based scores are generally higher for ID images than for OOD images".
> The main challenge of OOD detection is to find the difference between ID and OOD samples.
> Thus, my main question is actually **why ID and OOD samples have similar logit-based scores but different context scores?**.
> However, I think that the authors do not address it well.
> For example, the authors claim "the model recognizes ID images as belonging to ID labels, so the context score is relatively lower for ID images than for OOD images, as the context score is computed using a prompt without the ID label." But why does the change of prompt lead to the change of score? **Generally speaking, prompts adjust features through attention (which is why I mentioned ''the attention on prompts'' in the review). The change in features leads to a change in the score.** The authors haven't clearly explained how the score changes after the prompt is altered, and why there is a difference between ID and OOD samples.
>
> Regarding #2, the reason I believed that using only the context score was sufficient is due to the example in Figure 2; this example is clear. After reading the rebuttal, I still don't understand why the logit-based score must to be combined. Could you provide an example where using only the context score fails, and explain why, similar to Figure 2? In other words, I appreciate the additional experiments and Figure 3 provided by the authors, but **why** is it not good to use only the Context score? **What are the limitation of the Context score?**

---

> > ### Author Response · Authors · 2024-12-03
> >
> > Dear Reviewer,
> >
> > Thanks for replying back to us. We would like to address the reviewer’s comments as below:
> >
> > We agree with the reviewer that the main challenge of OOD detection is to find the difference between ID and OOD samples. Please let us further explain why ID and OOD samples could have similar logit-based scores but different context scores and vice versa. We have newly added Figure 7 in the Appendix where we also show some examples of images. Recall that our task is near OOD detection. In Figure 7, our ID images (red box) are images of dogs of a specific breed and our OOD images (green boxes) can be images of cats or dogs of other breeds.
> >
> > Context score computed only with the context vectors measures the association between the generic, non-class-specific contexts of ID images (e.g, generic dog-related features in Figure 7). This can be interpreted as a domain-specific score. On the other hand, the logit-based scores capture class-specific features and domain-specific features together because they are computed with the prompt of “context vectors+CLASS”. Although both scores contribute to the derivation of the logit score, they capture different perspectives. By subtracting the context score (or domain-specific score) from the logit-based score, we minimise the contribution of domain-specific features to the score because its contribution in near-OOD tasks results in noisy predictions. In near-OOD tasks, ID domain-specific features are often also present in near-OOD images as their distributions are very close to each other. This is, in fact, the limitation of using Context score alone, having overlapping ranges of context scores of ID and near-OOD images. It can be beneficial when it’s used with the logit-base scores, and we further illustrate this in Figure 7.
> >
> > In Figure 7, a perfect model would score all ID images with high logit-based scores and high context score (upper right area), indicating they have high ID domain-specific features and high class-specific features. On the other hand, all OOD images should ideally have low logit-based scores and low context scores (lower left area). For example, the near-OOD images of the bottom left green box are cat images that are easy to distinguish, thus having low class-specific features and low domain-specific features.
> >
> > In reality, it is hard to have a perfect model, and ID and near-OOD samples are spread across the middle area like the red box (ID) and the right green box (near-OOD) next to it. Although they have similar logit-based scores, we can tell the contribution of domain-specific features is different by looking at the different context scores. The model fails to capture the domain-specific features of red-box ID images (low context score), and the logit score is higher than the context score because of the contribution of class-specific features. The green-box near-OOD images on the upper right side have similar logit scores because of a higher contribution of domain-specific features (they are dog images) but a lower contribution of the class-specific features (they are images of dogs of different breeds).
> >
> > In summary, having two distinct contributions from class-specific features and domain-specific features (measured by context score) to the logit-based scores is the reason why some ID and OOD images can have similar logit-based scores with different context scores and vice versa. Also, closeness of ID and near-OOD distributions makes using the context score alone insufficient to distinguish between ID and near-OOD images.

---

> ### Comment · Reviewer_19PE · 2024-12-03
>
> Dear Authors,
>
> Thank you for your response; I feel much clearer now.
>
> After reading your response, my understanding is that the context score corresponds to domain-specific features, while the logit score corresponds to both class-specific and domain-specific features. The core function of subtracting the context score is to eliminate the influence of domain common knowledge when distinguishing between ID and near-OOD samples. Intuitively, this makes sense to me, but I believe the authors may need to restructure the entire paper around this point (e.g., more evidences to support it). Therefore, I keep my score now. I encourage the authors to restructure the paper with this focus and consider submitting it to the next conference.

---

### Official Review · Reviewer_muj8 · 2024-11-03

**Soundness:** 3
**Presentation:** 3
**Contribution:** 2
**Rating:** 6
**Confidence:** 4

**Summary:**

In response to the near OOD dataset and CLIP, this paper designed an OOD score based on logit, referred to as MLS. MLS computes the difference between the logit scores inferred by CLIP and a newly introduced complementary score known as the Context score. This approach effectively enhances the separation between ID samples and near OOD samples, resulting in a significant performance improvement.

**Strengths:**

1.	They introduced an MLS score based on the observation that the logit scores from CLIP, such as max logit, are positively correlated with their proposed context score. This MLS score reduces the overlap between ID data and near ID data compared to the max logit score.

2.	The MLS scoring method proposed in this paper requires no modifications to the model architecture and does not involve retraining. This characteristic makes it efficient and highly adaptable.

**Weaknesses:**

1.	The authors mention that logit score and context score are proportional, but this claim is not sufficiently accurate.
a)	The paper presents the case that Maxlogit is proportional to context score, but it remains unclear whether energy score and other logit-based scores follow this relationship as well. It is recommended that corresponding images be provided in the appendix.
b)	The proof that Maxlogit is proportional to the context score is based on experiments with MaPLe on the Caltech101 dataset, where we observed that ID and OOD data showed approximately linearly separable patterns in the images. It is recommended that the authors validate whether this proportionality and linear separability occur in other datasets and include those images in the appendix.
2.	The paper considers the Base to new dataset as a near OOD dataset. However, it is suggested that the authors also explore more classic near OOD datasets, such as those used in the MCM(Delving into Out-of-Distribution Detection with Vision-Language Representations) paper, where Imagenet10 is used as the ID dataset and Imagenet20 as the OOD dataset. Adding experiments involving these classic near OOD datasets would be beneficial.

**Questions:**

Please refer to the weakness.

---

> ### Author Response · Authors · 2024-11-18
> **Response to Reviewer muj8**
>
> We appreciate the reviewer’s feedback and would like to answer the questions below.
>
> ### 1. More evidence of the relationship between logit-based scores and context score
> ---
> We acknowledge the reviewer’s concern and have added further evidence of the proportional relationship between logit-based scores (MaxLogit and Energy) and the context score, using all eight prompt-learning models trained with 16-shot settings across 13 datasets (see Appendix A.4 for the added images). This pattern is consistently observed across various models and datasets.
>
> ### 2. More near OOD datasets
> ---
> We would like to kindly highlight that we have included additional experimental results using classic near-OOD datasets in Appendix A.3.7. These experiments follow the ImageNet Protocol (Palechor et al., 2023; Li et al., 2024), which closely resembles the ImageNet-10 vs. ImageNet-20 setup, featuring categories like dogs vs. wolves.

---

> > ### Author Response · Authors · 2024-11-24
> >
> > Dear Reviewer,
> >
> > Thanks again for reviewing our paper. We hope our responses answer your questions. If you have more questions or comments, please let us know. We are happy discuss it.
> >
> > Best regards,
> >
> > The Authors

---

### Official Review · Reviewer_KG3M · 2024-11-04

**Soundness:** 3
**Presentation:** 3
**Contribution:** 3
**Rating:** 6
**Confidence:** 4

**Summary:**

This paper explores the few-shot near-OOD identification capabilities of various prompt-based vision-language models (VLMs), such as CLIP. To enhance the identification of near-OOD samples in these models, it introduces a set of post-hoc metrics based on existing logit-based OOD techniques, which can be applied across a wide range of prompt-based VLMs.

**Strengths:**

I believe the paper addresses an important and often overlooked problem. The proposed Marginal Logit Score is simple and appears effective across a wide range of models and datasets, outperforming conventional OOD metrics in near-OOD detection. The metric is intuitively appealing and supported by experimental results.

**Weaknesses:**

-The authors mention that the margin scale is learned from the few-shot ID examples provided. However, there is insufficient analysis of how the number of shots impacts parameter learning and, ultimately, near-OOD detection performance.

- Although far-OOD detection is not the focus of this work, far-OOD samples are important for a model’s overall effectiveness in identifying OOD samples, especially in real-world scenarios. From this perspective, it is crucial to compare the performance of MLE with MCM on datasets or in scenarios where both near- and far-OOD samples are present.

**Questions:**

- I believe the paper would benefit from an ablation study on how the number of shots impacts the learning of the margin scale hyperparameter and, subsequently, the near-OOD detection performance.

- How does MLE compare to MCM on datasets that contain both near- and far-OOD samples?

---

> ### Author Response · Authors · 2024-11-18
> **Response to Reviewer KG3M**
>
> We appreciate your reviews for our work. We would like to clarify and respond each question below:
>
> ### 1. Impact of the number of shots
> ---
> We believe this comment can be addressed in the appendix along with our original submission. As outlined in L452–453, we included Appendix A.3 to present the near-OOD detection performance across various few-shot settings. Tables 7–11 display the results for 16-, 8-, 4-, 2-, and 1-shot scenarios with MLS-M, reporting the mean and standard deviation over three random seeds, while Tables 14–18 correspond to the same few-shot scenarios with MLS-E. These results demonstrate that our method consistently improves near-OOD detection performance across different few-shot settings. This also highlights our method’s effectiveness in learning the margin scale, given that the optimal learning scale cannot be directly obtainable.
>
> ### 2. Far-OOD results
> ---
> This comment can also be answered with our original submission. We provided an in-depth analysis of far-OOD detection performance in L491–512, with results detailed in Appendix A.3.5. For far-OOD detection, logit-based scores perform comparably to MCM, outperforming MCM in half of the evaluations while MCM performs better in the other half. For near-OOD detection, however, our method outperforms MCM in most cases. Consequently, in real-world scenarios where it is unclear whether OOD samples are from far or near OOD distributions, our method can be considered a more reliable method.

---

> > ### Author Response · Authors · 2024-11-24
> >
> > Dear Reviewer,
> >
> > Thanks again for reviewing our paper. We hope our responses answer your questions. If you have more questions or comments, please let us know. We are happy discuss it.
> >
> > Best regards,
> >
> > The Authors

---

> ### Comment · Reviewer_KG3M · 2024-12-02
>
> The reviewers addressed my concerns and I will stick to my original score.

---

### Meta-Review · Area_Chair_kKmE · 2024-12-24

**Metareview:**

In this work, the authors propose a logit-based MLS score and context score to effectively leverage domain-specific and class-specific features, significantly enhancing near-OOD detection performance by up to 11.67% in AUROC through few-shot prompt learning. Extensive experiments conducted on 13 datasets and 8 prompt learning methods demonstrate the effectiveness of the proposed approach. Most reviewers acknowledge the method’s ability to substantially boost near-OOD detection with minimal computational cost, without modifying the model architecture or requiring model retraining. However, the paper received mixed reviews — with two borderline accept, one borderline reject, and one reject — resulting in an average score of 5. The reviewers who assigned negative ratings expressed concerns about the paper's clarity, specifically the need for a more detailed explanation of the underlying reasons behind the performance improvements observed with MLS and context scores. They also suggested the inclusion of additional baseline comparisons. Despite the authors' rebuttal, these concerns persisted. After reviewing the feedback and rebuttal, I agree that the authors should incorporate comparisons with more recent OOD methods, such as NegLabel [1], and further refine the manuscript for resubmission to a future venue.

[1] Jiang, Xue, Feng Liu, Zhen Fang, Hong Chen, Tongliang Liu, Feng Zheng, and Bo Han. "Negative label guided ood detection with pretrained vision-language models." ICLR 2024.

**Additional Comments On Reviewer Discussion:**

Reviewers KG3M and muj8, who assigned borderline accept ratings, did not participate in the discussion. For the remaining two reviewers, the authors addressed the concerns raised by reviewer 19PE by clarifying the underlying mechanisms behind the proposed method and providing additional results on Caltech 101 and OxfordPets. However, both 19PE and QkT2 maintained reservations regarding the paper’s organization and writing quality. Additionally, reviewer QkT2 recommended incorporating more recent OOD methods to strengthen comparisons and further validate the effectiveness and generalizability of the proposed approach. Based on the final discussion, I agree that the authors should restructure and refine the paper to improve clarity and coherence. In addition, by including comparisons with other recent  near OOD methods, this will further enhance its quality for resubmission.

---

### Decision · Program_Chairs · 2025-01-22

Reject